# Towards Understanding Gradient Dynamics
# of the Sliced-Wasserstein Distance via Critical Point Analysis

**Christophe Vauthier** [1]  **Anna Korba** [2]  **Quentin Mérigot** [1]

## Abstract

In this paper, we investigate the properties of the Sliced Wasserstein Distance (SW) when employed as an objective functional. The SW metric has gained significant interest in the optimal transport and machine learning literature, due to its ability to capture intricate geometric properties of probability distributions while remaining computationally tractable, making it a valuable tool for various applications, including generative modeling and domain adaptation. Our study aims to provide a rigorous analysis of the critical points arising from the optimization of the SW objective. By computing explicit perturbations, we establish that stable critical points of SW cannot concentrate on segments. This stability analysis is crucial for understanding the behaviour of optimization algorithms for models trained using the SW objective. Furthermore, we investigate the properties of the SW objective, shedding light on the existence and convergence behavior of critical points. We illustrate our theoretical results through numerical experiments.

## 1. Introduction

An important problem in statistical learning is to approximate an intractable target probability measure $\rho$ on $\mathbb{R}^d$ with a probability measure supported on a finite set of points. Such problems arise in various contexts, such as sampling from Bayesian posterior distributions (Blei et al., 2017; Wibisono, 2018), generative modeling (Bond-Taylor et al., 2021) and training neural networks (Chizat & Bach, 2018; Mei et al., 2018). Recently, a popular framework to address

---
[1]Laboratoire de Mathématiques d'Orsay, Université Paris-Saclay, Gif-sur-Yvette, France [2]Centre de recherche en économie et statistique, ENSAE, Palaiseau, France. Correspondence to: Christophe Vauthier <christophe.vauthier@universite-paris-saclay.fr>, Anna Korba <anna.korba@ensae.fr>, Quentin Mérigot <quentin.merigot@universite-paris-saclay.fr>.

*Proceedings of the 42nd International Conference on Machine Learning*, Vancouver, Canada. PMLR 267, 2025. Copyright 2025 by the author(s).

such tasks has been to consider gradient flows, i.e., optimization dynamics on the space of probability measures, to minimize an objective functional of the form $\mathcal{F}(\mu) := \mathcal{D}(\mu|\rho)$, where $\mathcal{D}$ is a discrepancy (e.g. a distance, or a divergence) between measures. In practice, these can be simulated by considering an initial distribution that is a discrete measure uniformly supported on a set of particles. The particle positions then evolve according to a system of ODEs $\dot{X} = -\nabla F(X)$, which corresponds to the gradient flow of a functional $F : (\mathbb{R}^d)^N \to \mathbb{R}$, where $d$ is the dimension of the space and $N$ the number of particles. Then, a practical scheme is derived by discretizing in time this flow, e.g. with gradient descent. Reversely, gradient descent on particles can be seen as a discretized flow described by this system of ODEs.

Many divergences or distances can be considered as the discrepancy $\mathcal{D}$, each offering different tradeoffs between attractive geometrical properties and computational burden of the associated training dynamics. Generally the objective function is chosen so that the dynamic is tractable given the available information on $\rho$. When the density of $\rho$ is known up to a normalization constant, as often the case in Bayesian inference, standard choices include the Kullback-Leibler divergence (Salim et al., 2020), Kernel Stein Discrepancy (Fisher et al., 2021; Korba et al., 2021) or (eventually weighted) Fisher Divergences (Cai et al., 2024a;b). On the other hand, when samples of the target distribution are available, Integral Probability Metrics (IPM) or Optimal Transport distances are preferred, since they are well-defined for discrete measures. For instance in generative modeling, while original Generative Adversarial Networks are known to optimize a Jensen-Shannon divergence to the distribution of the samples (Goodfellow et al., 2020) and can be understood via the perspective of Wasserstein flows (Yi et al., 2023), a wide range of these metrics have been used for the training of GAN variants, e.g. Wasserstein-1 (Arjovsky et al., 2017), Sinkhorn divergences (Genevay et al., 2018), Maximum Mean Discrepancies (Li et al., 2017) or novel metrics interpolating between IPM and f-divergences (Birrell et al., 2022). Alternatively, recent work directly tackled generative modeling tasks through simulating Wasserstein gradient flows of such discrepancies, e.g. Sliced-Wasserstein distances (Liutkus et al., 2019; Dai & Seljak, 2021; Du

et al., 2023), Energy distances (Hertrich et al., 2024), or f-divergences (Fan et al., 2022; Choi et al., 2024). For all these methods, the choice of the discrepancy objective is crucial for their empirical success[1].

For instance, Wasserstein distances themselves appear to be suitable objectives, in the sense that they preserve the geometry of probability distributions, e.g. when computing barycenters (Rabin et al., 2012). However, for discrete measures, such distances are known to suffer from a large computational cost and poor statistical efficiency (Peyré et al., 2019). To alleviate this issue, several alternatives to the Wasserstein distance were proposed. Among these, the Sliced-Wasserstein distance (SW) (Bonneel et al., 2015) is a computationally attractive proxy. It involves averages of Wasserstein distances in dimension 1 (each of which can be computed in closed-form) with respect to an infinite number of directions. It has gained popularity in machine learning applications, such as computing barycenters of distributions (Bonneel et al., 2015), variational inference (Yi & Liu, 2023) or recently generative modeling (Kolouri et al., 2018; Liutkus et al., 2019; Dai & Seljak, 2021; Du et al., 2023). While its statistical and computational properties have been studied extensively in the literature (Nadjahi et al., 2020; Manole et al., 2022; Nietert et al., 2022), the behavior of its optimization dynamics remain largely unknown. In this paper, we consider the objective functional $\mathcal{F}$ to be a SW distance to a fixed measure $\rho$. We consider a gradient descent scheme on particles, as well as its continuous time and space counterpart, as an optimization scheme pushing particles from a source $\mu$ to approximate the target $\rho$. As this latter optimization problem is non-convex, it is natural to study the critical points that may be encountered during minimization. Our main objective is not only to understand the discretized problem, but also its continuous time and space analog, which motivates us to propose a notion of critical point for the continuous functional $\mathcal{F}$ that is compatible with the critical points for the discretized problem.

We note at this point that there exists many natural notions of critical points for a functional $\mathcal{G}$ defined on the space of probability measures over $\mathbb{R}^d$. A measure $\mu$ is called a critical point of $\mathcal{G}$ if for any curve $(\mu_t)_{t \in [0,1]}$ in the space of measures such that $\mu_0 = \mu$ belonging to a certain family of allowed perturbations, one has

$$\frac{d}{dt}\mathcal{G}(\mu_t)\bigg|_{t=0^+} = 0. \tag{1}$$

Our aim at this point is not to discuss the differentiability assumptions on $\mathcal{G}$, and we will therefore remain at an informal level. Depending on the set of allowed perturbations, we will recover several distinct and arguably interesting notions

of critical points:

- We will call $\mu$ an *Eulerian critical point* if it satisfies (1) for all perturbations of $\mu$ of the form $\mu_t = (1-t)\mu + t\nu$ for $\nu \in \mathcal{P}_2(\mathbb{R}^d)$. This coincides with the standard notion of critical point on the "flat" space $\mathcal{P}_2(\mathbb{R}^d)$ (i.e., not equipped with $W_2$).

- We will call $\mu$ a *Wasserstein critical point* if it satisfies (1) for all $W_2$-geodesics emanating from $\mu$. If $\mu$ is a probability density, we know from Brenier's theorem that geodesics are all curves of the form $\mu_t = ((1-t)\operatorname{Id} + tT)_{\#}\mu$ with $T$ the gradient of a convex function.

- Finally, we will call $\mu$ a *Lagrangian critical point* if it satisfies (1) for all curves of the form $\mu_t = (\operatorname{Id} + t\xi)_{\#}\mu$ for any vector field $\xi \in L^2(\mu, \mathbb{R}^d)^2$.

We now discuss the case where $\mathcal{G} = \mathcal{G}_\rho := \frac{1}{2}W_2^2(\cdot, \rho)$ is the squared Wasserstein distance to a probability density $\rho$ to fix ideas. First, we note that the only Eulerian critical point of this functional is $\rho$, a non-obvious fact, which follows from strong convexity of this $\mathcal{G}_\rho$ (Santambrogio, 2015, Proposition 7.19). However, such critical points are not meaningful when considering continuous time limits of gradient descent schemes (the ODE dynamics obeyed by the particles), as we do in this paper. Second, if $\mu \neq \rho$ and if $(\mu_t)_{t \in [0,1]}$ is the $W_2$-geodesic between $\mu$ and $\rho$, one can verify that $\mathcal{G}_\rho(\mu_t) \leq \mathcal{G}_\rho(\mu) - ct$ for some $c > 0$, thus implying that $\mu$ is not critical. Therefore, the only Wasserstein critical point of $\mathcal{G}_\rho$ is, again, $\mu = \rho$. In this case, every Wasserstein critical point is therefore also a Lagrangian critical point. The converse holds when $\mu$ is absolutely continuous, because one can take $\xi = T - \operatorname{Id}$, but not in general. As explained in (Mérigot et al., 2021) and studied in detail in (Sarrazin, 2022, Chapter 4), the functional $\mathcal{G}_\rho$ admits *many* Lagrangian critical points. First and foremost, any local or global minimizer of $X = (x_1, \ldots, x_N) \in (\mathbb{R}^d)^N \mapsto \mathcal{G}_\rho(\frac{1}{N}\sum_i \delta_{x_i})$ induces a Lagrangian critical point $\mu_X = \frac{1}{N}\sum_i \delta_{x_i}$ (showing the practical relevance of this notion), but moreover any $W_2$-limit of Lagrangian critical points are Lagrangian critical (with the caveat that the definition of Lagrangian critical points in (Sarrazin, 2022) is restricted to compactly supported measures and continuous perturbations $\xi$). This notion of critical point translates a difficulty that comes from the discretization, but that persists in the continuous limit.

**Contributions and outline.** Regarding the theoretical guarantees of optimization schemes applied to SW, a natural question is the following: given a sequence of discrete measures $(\mu_N)$ supported on $N$ atoms, and constructed using a first-order algorithm applied on a SW objective, can we expect this sequence to converge to the target measure

---

[1]Note though that GANs use a parametric setting, that is, we optimize $\theta \to \mathcal{D}(\mu_\theta, \rho)$ where $\theta$ is a parameter vector for a neural network.

[2]$L^2(\mu, \mathbb{R}^d) = \{f : \mathbb{R}^d \to \mathbb{R}^d, \int \|f(x)\|^2 d\mu(x) < \infty\}$.

$\rho$ as $N \to \infty$? This question is difficult because of the non-convexity of the discretized SW objective. However, we could hope that the non-convexity becomes milder as $N \to +\infty$, in the spirit of (Chizat & Bach, 2018; Mérigot et al., 2021).

Our paper is a first step towards answering this question and is organized as follows. In Section 2, we introduce the necessary background on optimal transport and Sliced-Wassertein distances. In Section 3, we discuss properties of gradient descent of the functional $\mathcal{F}$ over discrete measures and of its critical points, showing in particular that trajectories of gradient descent avoid the non-differentiability locus of $F$. In Section 4, we give an explicit characterization of Lagrangian critical points of the SW objective $\mathcal{F} = \frac{1}{2} \mathrm{SW}_2^2(\cdot, \rho)$, and we prove that our notion of critical points passes to weak limits under mild assumptions. This implies that the limit of discrete critical points (e.g., obtained numerically), is a Lagrangian critical point. In Section 5 we construct explicit examples of Lagrangian critical points of $\mathcal{F}$ supported on lower-dimensional subsets of $\mathbb{R}^d$. This shows in particular that there exists "bad" Lagrangian critical points points of the SW objective which are distinct from the target $\rho$. A natural question is then whether these "bad" Lagrangian critical points can actually occur as the limit of discrete measures obtained by an optimization algorithm. Since we expect that gradient descent will converge to stable critical points (Panageas et al., 2019), it is tempting to rule out these bad critical points by showing that they are unstable. We establish in Section 5 in dimension $d = 2$ that any Lagrangian critical point that contains a segment must be unstable. Since our proof relies on delicate explicit computations, the extension to lower dimensional critical points in higher dimension is left as future work. Finally Section 6 presents illustrations of our theoretical results on numerical experiments.

## 2. Background

**Measures and optimal transport**  We first give some background on optimal transport distances. We denote $\mathcal{P}(\mathbb{R}^d)$ the set of probability measures on $\mathbb{R}^d$ and $\mathcal{P}_p(\mathbb{R}^d)$ the set of probability measures with finite $p$th moment ($p \geq 1$). The $d$-dimensional Lebesgue and $k$-dimensional Hausdorff measures are denoted respectively by $\mathcal{L}^d$ and $\mathcal{H}^k$. In our setting, a probability density $\rho$ on $\mathbb{R}^d$ is a probability measure which is absolutely continuous with respect to the Lebesgue measure; for simplicity we will often use the same notation for $\rho$ and its density. Given a measurable map $T$ from $\mathbb{R}^d$ to itself and $\mu \in \mathcal{P}(\mathbb{R}^d)$, $T_{\#}\mu$ denotes the pushforward measure of $\mu$ by $T$. The Wasserstein distance of order $p$ between any probability measures $\mu, \nu$ in $\mathcal{P}_p(\mathbb{R}^d)$

is defined as

$$\mathrm{W}_p^p(\mu, \nu) = \inf_{\pi \in \Pi(\mu, \nu)} \int_{\mathbb{R}^d \times \mathbb{R}^d} \|x - y\|^p \mathrm{d}\pi(x, y), \quad (2)$$

where $\| \cdot \|$ denotes the Euclidean norm, and $\Pi(\mu, \nu)$ is the set of probability measures on $\mathbb{R}^d \times \mathbb{R}^d$ with marginals $\mu$ and $\nu$.

**1D optimal transport**  Consider probability measures $\mu, \nu \in \mathcal{P}_p(\mathbb{R})$, and let $F_\mu^{-1}$ and $F_\nu^{-1}$ be their quantile functions, i.e. $F_\mu^{-1}(t) = \inf\{s \in \mathbb{R} \mid F_\mu(s) \geq t\}$ where $F_\mu$ is the cumulative distribution function (cdf). By (Rachev & Rüschendorf, 1998, Theorem 3.1.2.(a)), the 1D Wasserstein distance is the $L^p$ distance between the quantile functions,

$$\mathrm{W}_p^p(\mu, \nu) = \int_0^1 |F_\mu^{-1}(t) - F_\nu^{-1}(t)|^p \mathrm{d}t. \quad (3)$$

If $X = (x_1, \dots, x_N) \subseteq \mathbb{R}^N$ is a finite set in $\mathbb{R}$, $\mu_X = \frac{1}{N} \sum_i \delta_{x_i}$ is the associated empirical measure, and $\sigma_X$ is a permutation such that $i \mapsto x_{\sigma_X(i)}$ is non-decreasing (similarly, we define $Y, \mu_Y, \sigma_Y$), Equation (3) becomes more explicit:

$$\mathrm{W}_p^p(\mu_X, \mu_Y) = \frac{1}{N} \sum_{i=1}^N |x_{\sigma_X(i)} - y_{\sigma_Y(i)}|^p, \quad (4)$$

showing the complexity of 1D optimal transport is the same as sorting, i.e. $O(N \log N)$. However, in dimension higher than one, there is no explicit expression for $\mathrm{W}_p^p(\mu, \nu)$ and despite the progress made in the last decade, the computational cost remains superlinear in the number of atoms (Peyré et al., 2019).

**Sliced-Wasserstein distance**  The Sliced-Wasserstein (SW) distance (Rabin et al., 2012) defines an alternative metric by leveraging the computational efficiency of $\mathrm{W}_p^p$ for univariate distributions. For $\theta \in \mathbb{S}^d$, $P_\theta : \mathbb{R}^d \to \mathbb{R}$ denotes the linear form $x \mapsto \langle \theta | x \rangle$. Then, the SW distance of order $p$ between $\mu, \nu \in \mathcal{P}_p(\mathbb{R}^d)$ is

$$\mathrm{SW}_p^p(\mu, \nu) = \int_{\mathbb{S}^{d-1}} \mathrm{W}_p^p(P_{\theta\sharp}\mu, P_{\theta\sharp}\nu) d\theta, \quad (5)$$

where $\mathbb{S}^{d-1}$ is the $(d-1)$-dimensional unit sphere and $d\theta$ is the uniform distribution on $\mathbb{S}^{d-1}$. Since $P_{\theta\sharp}\mu$, $P_{\theta\sharp}\nu$ are univariate distributions, the Wasserstein distances in (5) are conveniently computed using (3). The sliced-Wasserstein distance $\mathrm{SW}_p$ is always smaller than the original Wasserstein distance (Bonnotte, 2013, Proposition 5.1.3), and is even bi-Hölder equivalent to this distance on the subset $\mathcal{P}(B(0, R)) \subseteq \mathcal{P}_p(\mathbb{R}^d)$. The computational and statistical aspects of sliced-Wasserstein distances are by now well studied, we refer to (Nadjahi et al., 2020) and references therein.

# 3. Discrete Sliced-Wasserstein distance dynamics

Before investigating the convergence of the gradient flow of the Sliced-Wasserstein distance to its critical points and the characterization of the latter, we first study in this section the optimization of the Sliced-Wasserstein distance in practice, where the optimized (source) measure is discrete. Our first subsection studies the differentiability properties of the Sliced-Wasserstein objective when the first argument is a discrete measure, while the second provides a descent lemma for this objective. Finally, we show quantitatively that for a suitable stepsize, gradient descent does not collapse particles and is thus well-defined for all times.

**Differentiability of the SW functional.** We consider a target probability density $\rho \in \mathcal{P}_p(\mathbb{R}^d)$, and we define the function

$$F : X = (X_1, ..., X_N) \in (\mathbb{R}^d)^N \mapsto \frac{1}{p} \mathrm{SW}_p^p(\mu_X, \rho), \quad (6)$$

where $\mu_X = \frac{1}{N} \sum_{i=1}^N \delta_{X_i}$ is the uniform empirical measure associated to the set of points $X$. As $\rho$ has finite $p$-moment, $F(X) < +\infty$ for every point cloud $X$. As seen in Section 2, the SW distance involves sorting the projections of $X$ over directions. However, the sorting operation, seen as a function of $\mathbb{R}^N$ to $\mathbb{R}^N$, is piecewise linear and non-differentiable when two of the coordinates agree. We may therefore expect our functional $F$ to be non-differentiable at any point cloud $X$ which belongs to the generalized diagonal $\Delta_N := \{(X_1, ..., X_N) \in (\mathbb{R}^d)^N \mid \exists i \neq j, X_i = X_j\}$. The next proposition shows differentiability of $F$ on the complement of this generalized diagonal.

As usual, we denote $\mathfrak{S}_N$ the group of permutations of $\{1, ..., N\}$. We will use the notation $V_{\theta,i}$ for the $i$-th Power cell associated to $P_{\theta \#}\rho$, i.e.

$$V_{\theta,i} = F_{P_{\theta \#}\rho}^{-1}\left(\left[\frac{i}{N}, \frac{i+1}{N}\right]\right). \quad (7)$$

Moreover, given a point cloud $X = (X_1, \ldots, X_N) \in (\mathbb{R}^d)^N$, we denote $\sigma_{X,\theta} \in \mathfrak{S}_N$ a permutation such that the map $i \in \{1, \ldots, N\} \mapsto \langle X_{\sigma_{X,\theta}(i)} | \theta \rangle$ is non-decreasing.

**Proposition 3.1.** *If $p \geq 2$, then $F$ is differentiable at any point cloud $X = (X_1, \ldots, X_N) \in (\mathbb{R}^d)^N$ which does not belong to the generalized diagonal $\Delta_N$. The gradient of $F$ is continuous on $(\mathbb{R}^d)^N \setminus \Delta_N$, and its component with respect to the $i$-th vector $X_i$ is then*

$$\nabla_{X_i} F(X) = \int_{\mathbb{S}^{d-1}} \int_{V_{\theta, \sigma_{X,\theta}^{-1}(i)}} \mathrm{sgn}(\langle X_i | \theta \rangle - x)$$
$$\times |\langle X_i | \theta \rangle - x|^{p-1} \theta dP_{\theta \#}\rho(x) d\theta, \quad (8)$$

In the particular case where $p = 2$, this expression can be further simplified by introducing the barycenters of the Power cells $V_{\theta,i}$, i.e. $b_{\theta,i} = N \int_{V_{\theta,i}} x dP_{\theta \#}\rho(x)$:

$$\nabla_{X_i} F(X) = \frac{1}{N}\left(\frac{1}{d}X_i - \int_{\mathbb{S}^{d-1}} b_{\theta,\sigma_{X,\theta}^{-1}(i)} \theta d\theta\right). \quad (9)$$

The proof of Proposition 3.1 is deferred to Appendix B.1. This proposition is valid in the semi-discrete setting, where the source measure is finitely supported and $\rho$ has a density, while similar results in the literature tackle different settings, e.g. fully-discrete (Tanguy et al., 2024a) or where both measures are densities (Manole et al., 2022).

**Descent lemma.** While our previous result provides a general formula for gradients of SW distances of order $p \geq 2$, we focus on the particular case $p = 2$ where the computations are the most simple. We then have the following "descent lemma", which gives guarantees that a gradient step decreases the loss, for the gradient descent on $F$,

**Proposition 3.2.** *For every $X \in (\mathbb{R}^d)^N \setminus \Delta_N$ and every $\lambda > 0$, denoting $Y := X - \lambda \nabla F(X)$, we have*

$$F(Y) - F(X) \leq -\lambda\left(1 - \frac{\lambda}{2Nd}\right)\|\nabla F(X)\|^2 \quad (10)$$

The proof of Proposition 3.2 is provided in Appendix B.2 and relies on the semiconcavity of $F$. This proposition implies that if $X$ is not a critical point of $F$ and if the step-size $\lambda$ belongs to $(0, 2Nd)$, one gradient descent step from $X$ strictly decreases the value of $F$. In particular, the r.h.s. of the inequality (10) is minimal for a step-size $\lambda = Nd$, and we may expect the convergence speed of the gradient descent to be the fastest for step-sizes around this value. Considering the expression of $\nabla F(X)$ given by (9), one iteration of the gradient descent with such a step writes:

$$X_i^{k+1} \leftarrow X_i^k - Nd\nabla_i F(X^k) = d\int_{\mathbb{S}^{d-1}} b_{\theta,\sigma_{X^k,\theta}^{-1}(i)} \theta d\theta. \quad (11)$$

Interestingly, choosing a step of $Nd$ for the $\mathrm{SW}_2^2$ objective is reminiscent of the results obtained by (Mérigot et al., 2021). They study a variant of Lloyd's algorithm, which optimizes $X \mapsto \mathrm{W}_2^2(\mu_X, \rho)$ by assigning to $X^{k+1}$ the barycenters of the Power cells (also referred to as Laguerre cells) associated to $X^k$, and which was proven, under certain conditions, to approximate $\rho$ closely after a single step (see Theorem 3 and Corollary 4 in (Mérigot et al., 2021)).

Another consequence of Proposition 3.2 is that the sum of squared gradients of $F$ at $X^k$ is bounded. Indeed, for $\lambda = Nd$, we have

$$\|\nabla F(X^k)\|^2 \leq \frac{2}{Nd}(F(X^k) - F(X^{k+1})), \quad (12)$$

which implies that any converging subsequence of $(X^k)$ converges to a critical point $X^*$ of the energy. The convergence of the whole sequence $(X^k)$ to a critical point is open in general. It can be proven if one assumes that that the energy level $F^{-1}(F(X^*))$ only contains a finite number of critical points, as in (Bourne et al., 2020, Appendix), but this hypothesis cannot be checked in practice. (Portales et al., 2024) proves convergence of the whole sequence of iterates of Lloyd-type algorithms in several settings, but they acknowledge that their techniques do not extend to the case of $\mathcal{F} = \frac{1}{2} \text{SW}_2^2(\cdot, \rho)$ when $\rho$ is a probability density.

**Well-behavedness of gradient descent** In the gradient descent scheme described above, it is a priori possible that the iterates will get close to the generalized diagonal $\Delta_N$. This is a problem, as $F$ is only known to be differentiable on $(\mathbb{R}^d)^N \setminus \Delta_N$. The following property ensures that, if the densities of the projections of $\rho$ are bounded, the iterates will remain away from $\Delta_N$.

**Proposition 3.3.** *Let $X \in (\mathbb{R}^d)^N \setminus \Delta_N$ and $\lambda > 0$, and define $Y := X - \lambda \nabla F(X)$. Then, if $\lambda \in (0, Nd)$, we have $Y \notin \Delta_N$.*
*Furthermore, if there exists $\beta > 0$ which bounds from above the density of $P_{\theta\#}\rho$ for every $\theta \in \mathbb{S}^{d-1}$, then there exists $C = C(d)$ such that for every $i \neq j$, if $\|X_i - X_j\| < \frac{dC}{N\beta}$, then $\|Y_i - Y_j\| > \|X_i - X_j\|$. In particular, if $X$ is a critical point of $F$, then*

$$\min_{i \neq j} \|X_i - X_j\| \geq \frac{dC}{N\beta} \qquad (13)$$

The proof of Proposition 3.3 is provided in Appendix B.3. Interestingly, the proof strategy we use also implies that the continuous flow $\dot{X} = -\nabla F(X)$ is defined for all times when initialized from a point cloud $X(0)$ not in $\Delta_N$, as discussed in the same appendix.

*Remark* 3.4. Note that Proposition 3.1, Proposition 3.2, and the first part of Proposition 3.3 actually admit straightforward extensions, with the same statements, to the case where $\rho$ is only assumed to have no atoms (note that this includes for instance densities supported on a lower dimensional manifold of $\mathbb{R}^d$ (which are not absolutely continuous w.r.t. Lebesgue in $\mathbb{R}^d$ but are without atoms). Indeed, it turns out that for such $\rho$, for almost every $\theta \in \mathbb{S}^{d-1}$, its projection $P_{\theta\#}\rho$ has no atoms, and we can thus define the Power cells $V_{\theta,i}$ and the barycenters $b_{\theta,i}$, which requires minimal changes in the proof. For further discussion, we refer to Appendix B.4, which also examines extensions of these results to more general target measures $\rho \in \mathcal{P}_2(\mathbb{R}^d)$.

# 4. Characterization of critical points

The goal of this this section is to derive a rigorous characterization of Lagrangian critical points of the SW objective

$\mathcal{F} = \frac{1}{2} \text{SW}_2^2(\cdot, \rho)$. Unlike in the previous section, where we worked in the semi-discrete setting (i.e. with $\mu$ discrete and $\rho$ a density), our framework will hold for general $\mu, \rho \in \mathcal{P}_2(\mathbb{R}^d)$.

## 4.1. Barycentric characterization

As in the introduction, we first define Lagrangian critical points using derivatives of $\mathcal{F}$ along perturbations of the measure.

**Definition 4.1.** A measure $\mu \in \mathcal{P}_2(\mathbb{R}^d)$ is a *Lagrangian critical point* for $\text{SW}_2^2(\cdot, \rho)$ if for every $\xi \in L^2(\mu, \mathbb{R}^d)$,

$$\left. \frac{d}{dt} \text{SW}_2^2((\text{Id} + t\xi)_{\#}\mu, \rho) \right|_{t=0^+} = 0. \qquad (14)$$

The right derivative is always well-defined, since a convex function always has left and right directional derivatives, as will be justified in Proposition 4.7(a).

As Definition 4.1 is difficult to verify in practice, we will now define a second notion of Lagrangian criticality, which we will prove to be equivalent to the first under mild assumptions on $\mu$, and which will be very similar in spirit to the concept of Lagrangian critical measures for the standard Wasserstein distance developed in (Sarrazin, 2022).

We assume that $\mu \in \mathcal{P}_2(\mathbb{R}^d)$ is fixed, and for every direction $\theta$, we denote $\gamma_\theta$ the unique 1D optimal transport plan between $\mu_\theta = P_{\theta\#}\mu$ and $\rho_\theta = P_{\theta\#}\rho$. We finally consider the barycentric projection $\bar{\gamma}_\theta$ of this transport plan (Ambrosio et al., 2005, Definition 5.4.2), which we can define using conditional expectations:

$$\bar{\gamma}_\theta : \mathbb{R} \to \mathbb{R}, \ u \mapsto \mathbb{E}_{(U,V) \sim \gamma_\theta}[V \mid U = u]. \qquad (15)$$

We are now ready to state our second definition of Lagrangian critical points.

**Definition 4.2.** A measure $\mu \in \mathcal{P}_2(\mathbb{R}^d)$ is a *barycentric Lagrangian critical point* for $\text{SW}_2^2(\cdot, \rho)$ if $v_\mu = 0$ $\mu$-a.e., where $v_\mu$ is the vector field defined by

$$v_\mu : x \mapsto \frac{1}{d}x - \int_{\mathbb{S}^{d-1}} \bar{\gamma}_\theta(\langle x|\theta\rangle)\theta d\theta. \qquad (16)$$

Note that this integral is well-defined by the selection result (Villani, 2008, Corollary 5.22).

*Remark* 4.3. This notion of barycentric Lagrangian critical point appears in (Li & Moosmueller, 2025) (although it is not explicitly named), where it plays a role in the study of the convergence of stochastic iterative approximation schemes for the Sliced-Wasserstein distance. Indeed, Assumption (A3) therein rewrites in our framework as "$\eta$ is a barycentric Lagrangian critical point for $\text{SW}_2^2(\cdot, \mu)$" (see also (Li & Moosmueller, 2025, Remark 8)).

Our two notions of Lagrangian critical points are compatible with the notion of critical points of the discretized problem defined in the previous section, as stated in the following Proposition.

**Proposition 4.4.** *Assume that $\rho$ is a probability density and let $X \in (\mathbb{R}^d)^N \setminus \Delta_N$. Then $\nabla F(X) = 0$ if and only if $\mu_X$ is a Lagrangian critical point for $\mathrm{SW}_2^2(\cdot, \rho)$ if and only if $\mu_X$ is a barycentric Lagrangian critical point for $\mathrm{SW}_2^2(\cdot, \rho)$.*

The proof of Proposition 4.4 is deferred to Appendix B.5. A natural (non trivial) follow-up question is then whether the limit of a sequence of discrete critical points $\mu_N = \frac{1}{N} \sum_{i=1}^N \delta_{X_i}$ (e.g. obtained numerically) is also a critical point (as defined either in Definition 4.1 or in Definition 4.2). The following theorem provides an answer to this question.

**Theorem 4.5** (Limits of critical points are critical). *Assume that $\rho$ is without atoms and supported on a compact $\Omega \subseteq \mathbb{R}^d$. If a sequence $(\mu_N)_{N \geq 1}$ of barycentric Lagrangian critical points for $\mathrm{SW}_2^2(\cdot, \rho)$ converges weakly to an atomles measure $\mu$, then $\mu$ is barycentric Lagrangian critical for $\mathrm{SW}_2^2(\cdot, \rho)$.*

The proof of Theorem 4.5 can be found in Appendix B.8. Crucially, it relies on the study of the intricate relationship between the two definitions of Langrangian critical points we have defined. This study is detailed in the next section.

### 4.2. Technical tools for Theorem 4.5

We have already shown in Proposition 4.4 that the two notions of critical points agree for discrete measures. Here, we discuss why Definition 4.2 is also natural in a more general setting, such as those of Wasserstein gradient flows, i.e., curves $(\mu_t)_{t>0}$ of steepest descent with respect to the Wasserstein-2 ($\mathrm{W}_2$) metric of the objective $\mathcal{F}$. Indeed, by (Bonnotte, 2013, Section 5.7.1), when $\rho$ is absolutely continuous, the absolutely continuous stationary points $\mu$ of the gradient flow dynamics of $\mathcal{F}$ are characterized by

$$\int_{\mathbb{S}^{d-1}} \varphi_\theta'(\langle x|\theta \rangle)\theta d\theta = 0, \quad \mu - \text{a.e. } x \in \mathbb{R}^d \quad (17)$$

where $\varphi_\theta$ is the Kantorovitch potential from $\mu_\theta$ to $\rho_\theta$ for the cost $c(s,t) = \frac{1}{2}(s-t)^2$. But since we have $\varphi_\theta' = \mathrm{Id} - T_\theta$ where $T_\theta$ is the unique optimal transport map from $\mu_\theta$ to $\rho_\theta$ (Santambrogio, 2015, Section 1.3.1), and $\bar{\gamma}_\theta = T_\theta$ (as $\gamma_\theta = (\mathrm{Id}, T_\theta)_\# \mu_\theta$), we see that (17) rewrites as $v_\mu = 0$ $\mu$-ae, and thus an absolutely continuous measure $\mu$ is a stationary point of the Wasserstein gradient flow of $\mathcal{F}$ iff it is a barycentric Lagrangian critical point. Furthermore, (Bonnotte, 2013, Lemma 5.7.2) immediately rewrites as the following result:

**Proposition 4.6.** *(Bonnotte) If $\mu, \rho \in \mathcal{P}(B(0, R))$ are absolutely continuous and both have a strictly positive density*

on $B(0, R)$, then $\mu = \rho$ if and only if $\mu$ is barycentric Lagrangian critical for $\mathrm{SW}_2^2(\cdot, \rho)$

Now, we will see that Definition 4.1 and 4.2 coincide if $\mu, \rho$ are compactly supported and without atoms. For $\mu \in \mathcal{P}(\mathbb{R}^d)$, we denote $\| \cdot \|_{L^2(\mu)}$ and $\langle \cdot, \cdot \rangle_{L^2(\mu)}$ the norm and the inner product on $L^2(\mu, \mathbb{R}^d)$.

**Proposition 4.7.** *Let $\mu \in \mathcal{P}_2(\mathbb{R}^d)$, then :*

*(a) The function $F_\mu : L^2(\mu, \mathbb{R}^d) \mapsto \mathbb{R}$ defined as follows is convex:*

$$F_\mu : \xi \mapsto \frac{1}{d}\|\xi\|_{L^2(\mu)}^2 - \mathrm{SW}_2^2((\mathrm{Id} + \xi)_\# \mu, \rho) \quad (18)$$

*(b) The vector field $v_\mu$ belongs to $L^2(\mu, \mathbb{R}^d)$. Furthermore, $-2v_\mu$ belongs to the subdifferential of $F_\mu$ at 0, that is, for every $\xi \in L^2(\mu, \mathbb{R}^d)$,*

$$F_\mu(0) - 2\langle v_\mu | \xi \rangle_{L^2(\mu)} \leq F_\mu(\xi) \quad (19)$$

*(c) If $\mu$ and $\rho$ have compact support and are without atoms, then for every vector field $\xi \in L^2(\mu, \mathbb{R}^d)$, the function $\varphi(t) = \mathrm{SW}_2^2((\mathrm{Id} + t\xi)_\# \mu, \rho)$ is differentiable at $t = 0$, with*

$$\varphi'(0) = 2\langle v_\mu | \xi \rangle_{L^2(\mu)} \quad (20)$$

**Corollary 4.8.** *If $\mu$ is a Lagrangian critical point for $\mathrm{SW}_2^2(\cdot, \rho)$, then it is also a barycentric Lagrangian critical point for $\mathrm{SW}_2^2(\cdot, \rho)$. If furthermore $\mu$ and $\rho$ have compact support and are without atoms, then the converse statement is also true.*

The proof of Proposition 4.7 and Corollary 4.8 can be found in Appendix B.6 and Appendix B.7 respectively. Our Proposition 4.7(c) extends the result (Bonnotte, 2013, 5.1.7. Proposition) on the differentiability of SW. In particular, Bonnotte's results holds under the strong assumption that $\mu, \rho$ are absolutely continuous, whereas Proposition 4.7(c) makes the much milder assumption that they are without atoms.[3]

## 5. Lower-dimensional critical points: existence and instability

### 5.1. Leveraging symmetry to find critical points

Now that we have characterized Lagrangian critical points, it is natural to ask ourselves whether there can exist Lagrangian critical measures $\mu$ different than the target distribution $\rho$. An effective approach to construct such critical points is to look for measures that are supported on a symmetry axis of a well-chosen measure $\rho$. Our next result provides several examples.

---

[3]For instance, distributions on lower dimensional manifolds do not have a density with respect to the Lebesgue measure but can be without atoms.

**Proposition 5.1.** *The following are barycentric Lagrangian critical points :*

(a) *In dimension $d = 2$, the measure $\mu = \frac{\pi}{8}\mathcal{H}^1_{|[-\frac{4}{\pi},\frac{4}{\pi}]}$ is a barycentric Lagrangian critical point for the measure $\rho$ with density $\rho(x) = \frac{1}{2\pi}\frac{1}{\sqrt{1-|x|^2}}\mathbf{1}_{B(0,1)}(x)$, which we will hereafter call the (two-dimensional) sliced-uniform measure.*

(b) *In dimension $d > 1$, the measure $\mu$ defined by $\mu := (\mathrm{Id}, 0_{d-1})_{\#}\mu_0$ with $\mu_0 = \mathcal{N}(0, \alpha_d^2)$ is a barycentric Lagrangian critical point for the standard Gaussian $\rho = \mathcal{N}(0, I_d)$, where $\alpha_d$ is defined by $\alpha_d = d\int_{\mathbb{S}^{d-1}}|\langle\theta|e_1\rangle|d\theta$ and $(e_1, ..., e_d)$ is the canonical basis of $\mathbb{R}^d$.*

We refer to $\rho$ in Proposition 5.1(a) as the sliced-uniform measure, as for every $\theta \in \mathbb{S}^{d-1}$, its projection $P_{\theta\#}\rho$ is the normalized restriction of the Lebesgue measure to $[-1, 1]$. Proposition 5.1(a) provides an example of target measure $\rho$ on a disk in $d = 2$ that is symmetric with respect to any line, and which admits in this case a critical point supported on a segment, hence of strictly lower dimension. Proposition 5.1(b) provides a similar result for isotropic Gaussians. The proof of Proposition 5.1 is deferred to Appendix B.9.

We now discuss informally why we expect to find critical points of this type. Assume that there exists a subspace $H$ of $\mathbb{R}^d$ such that the target $\rho$ is symmetric with respect to $H$, i.e. $S_{H\#}\rho = \rho$ where $S_H$ is the reflection at $H$. Then, if $\mathrm{spt}(\mu) \subseteq H$, then for every $\theta \in \mathbb{S}^{d-1}$, we have $\rho_{S_H(\theta)} = \rho_\theta$ and $\mu_{S_H(\theta)} = \mu_\theta$, thus $T_\theta = T_{S_H(\theta)}$. Thus, for every $x \in \mathrm{spt}(\mu) \subseteq H$, we have by straightforward computations [4]:

$$v_\mu(x) = \frac{x}{d} - \int_{\mathbb{S}^{d-1}} T_\theta(\langle\theta|x\rangle)P_H(\theta)d\theta \in H, \quad (21)$$

where $P_H$ is the projection on $H$. This means that the iterates of the gradient descent $\mu \leftarrow (\mathrm{Id} - \gamma v_\mu)_{\#}\mu$ will remain supported on $H$. Therefore, taking the limit of the trajectory (for an infinite number of iterations) should be a critical point of $\mathcal{F}$, still supported on $H$.

### 5.2. Some explicit unstable critical points

Previously, we highlighted critical points that are supported on a subset of $\mathbb{R}^d$, for a target distribution that is full-dimensional. This is problematic because our gradient algorithm may be stuck at these critical points, which are typically at a high level in the energy landscape. We now investigate their stability, as gradient descent is unlikely to get stuck at unstable critical points, with the aim of showing that such points do not appear in practice.

We will focus on a particular case of unstable behavior. We will restrict ourselves to the case $d = 2$, and we will show that when the target measure $\rho$ is absolutely continuous, measures $\mu$ that contain a part supported on a segment are not stable for $\mathrm{SW}_2^2$ when perturbed in a certain way.

**Proposition 5.2.** *Let $\rho \in \mathcal{P}_2(\mathbb{R}^2)$ be an absolutely continuous measure, such that the densities of its projections $\rho_\theta$ are uniformly bounded from above by $b > 0$. Let $\mu \in \mathcal{P}_2(\mathbb{R}^2)$ be any measure such that there exists a segment $S \subseteq \mathbb{R}^2$ and $a > 0$ such that $a\mathcal{H}^1_{|S} \leq \mu$. Then, if $\mu^t$ is the perturbation*

$$\mu^t := \frac{1}{2}(\tau_{-t\vec{n}\#}\mu + \tau_{t\vec{n}\#}\mu) \quad (22)$$

*where $\tau_{\vec{a}}$ is the translation by $\vec{a} \in \mathbb{R}^2$ and $\vec{n} \in \mathbb{S}^1$ is orthogonal to $S$, then the perturbation $\mu_t$ is unstable for $\mathrm{SW}_2^2(\cdot, \rho)$: that is, for any $C > 0$, there exists a neighborhood $(-\varepsilon, \varepsilon)$ of $t = 0$ in which*

$$\mathrm{SW}_2^2(\mu^t, \rho) \leq \mathrm{SW}_2^2(\mu, \rho) - Ct^2. \quad (23)$$

The proof of Proposition 5.2 is deferred to Appendix B.10. Our Proposition 5.2 proves that critical points as described therein, are highly unstable. Indeed, we do not have a Taylor expansion $\mathrm{SW}_2^2(\mu^t, \rho) = \mathrm{SW}_2^2(\mu, \rho) + at + \frac{1}{2}bt^2 + o(t^2)$ with $a = 0$ and $b < 0$. Instead, the inequality $\mathrm{SW}_2^2(\mu^t, \rho) \leq \mathrm{SW}_2^2(\mu, \rho) - Ct^2$ is true for *any* $C > 0$ provided that $t$ is close enough to 0. In particular, this implies that $\mathrm{SW}_2^2(\mu^t, \rho)$ is not twice differentiable at $t = 0$. Hence, while the SW flow may exhibit critical points that are not global minimizers, they may be unstable in general. Our result proves this in the case where the target contains a segment.

On the other hand, the perturbation $\mu^t$ used in Proposition 5.2 is not of the form $(\mathrm{Id} + t\xi)_{\#}\mu$, and thus does not fit in our previously defined framework of Lagrangian critical points. However, this result suggests that by taking a $L^2$ vector field $\xi$ which alternates rapidly between $\vec{n}$ and $-\vec{n}$ on the segment $S$, then $\mathrm{SW}_2^2((\mathrm{Id} + t\xi)_{\#}\mu, \rho)$ will also have a local maximum at $t = 0$ (see for example the numerical experiments in Figure 1 below).

Note that the proof of Proposition 5.2 makes heavy use of the properties of the segment, among which that the existence of a relatively simple closed form of the quantile functions of the projections are available. In general, it is difficult to describe how the quantile functions of the projections behave when considering general measures and perturbations.

## 6. Experiments

This section presents the results of our experiments, designed to examine the extent to which the theoretical find-

---

[4]$v_\mu(x) = \frac{x}{d} - \int \frac{T_\theta(\langle\theta|x\rangle)\theta + T_{S_H(\theta)}(\langle S_H(\theta)|x\rangle)S_H(\theta)}{2}d\theta = \frac{x}{d} - \int T_\theta(\langle\theta|x\rangle)\frac{\theta + S_H(\theta)}{2}d\theta$ as $x \in H$.

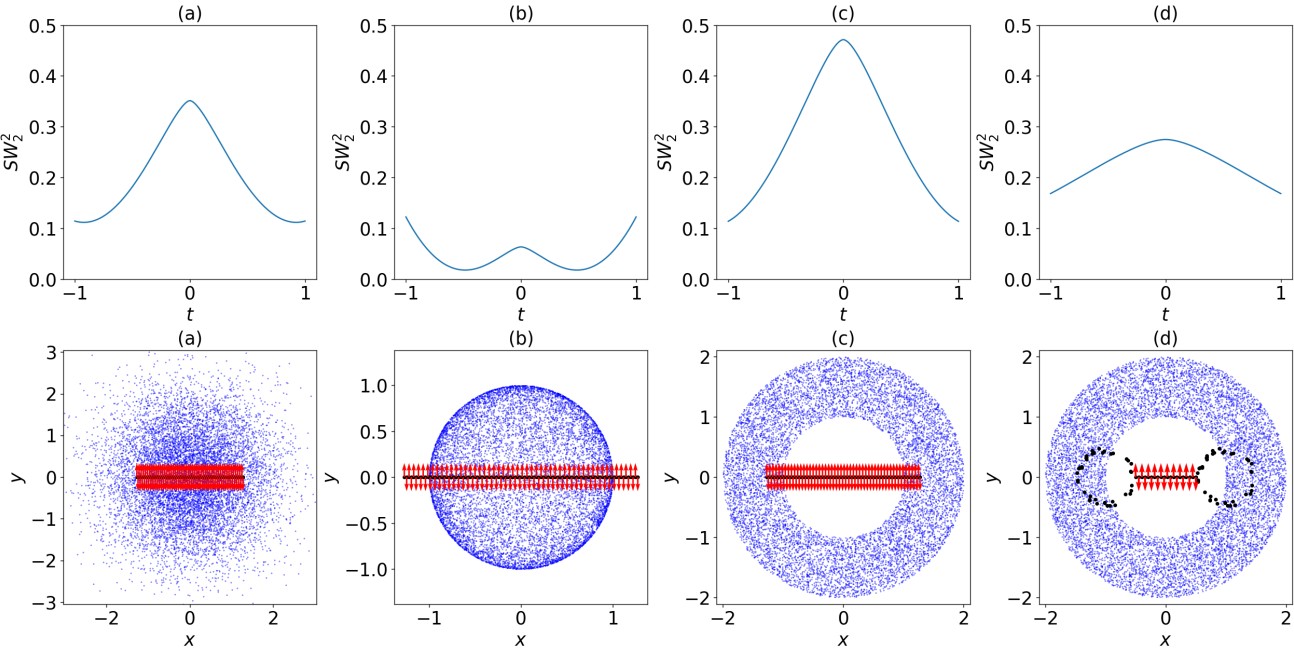

*Figure 1.* Instability of measures containing an horizontal segment. On the top line are plotted the value $\mathrm{SW}_2^2(\mu^t, \rho)$ for different measures $\mu$, $\rho$ and perturbations $\xi$. On the bottom line are depictions of the different $\mu$ (black points), $\rho$ (approximated by the blue points) and $\xi$ (red arrows). Columns (a) and (b): $\mu$ is a point cloud of $N = 100$ points uniformly distributed on the segment $[-4/\pi, 4/\pi] \times \{0\}$, $\xi$ alternates between $e_2$ and $-e_2$, and $\rho$ is the normal (a) and sliced-uniform distribution (see Proposition 5.1) (b). Column (c): Same $\mu$ and $\xi$, and this time $\rho$ is the uniform measure on the shell $C(0, 1, 2)$. Column (d) : $\rho$ is again the shell, and $\mu$ is a point cloud with a "dumbbell-like" shape, whose central segment is perturbed similarly as in (a),(b),(c).

ings from the previous sections hold in practice[5].

In the experiments, $F(X)$ is approximated by taking the average of 1D Wasserstein distances over $L = 100$ directions, and by approximating $\rho$ with a point cloud $Y$ containing $M = 10000$ points.

**Instability of critical points.** First, we considered a point cloud $X = (X_1, ..., X_N)$ with $X_i = -\frac{4}{\pi} + \frac{8}{\pi} \frac{i-1}{N-1}$, with $N = 100$, that approximates the measure $\mu = \frac{\pi}{8} \mathcal{H}^1_{|[-\frac{4}{\pi}, \frac{4}{\pi}]}$ that was studied in Section 5. We considered a perturbation $\xi$ that alternates between $e_2$ and $-e_2$ and we plotted $t \mapsto F(X + t\xi) = \mathrm{SW}_2^2(\mu_X^t, \rho)$ in Figure 1 for different choices of $\rho$. We see that the numerical results are consistent with our theoretical findings: indeed, we have a local maximum for all three considered target measures. Furthermore, when $X$ is a point cloud with a more complex shape but which includes an horizontal segment, we still observe an instability by perturbing the segment and leaving the other points of the point cloud unchanged. Moreover, while the perturbation considered in Proposition 5.2 is not induced by a vector field $\xi$, those in these experiments are, and they do

exhibit an instability. This suggests that, if we approximate the perturbation in Proposition 5.2 closely enough with a vector field that alternates between $\vec{n}$ and $-\vec{n}$, we could obtain a unstable perturbation of the form $(\mathrm{Id} + t\xi)_\# \mu$, which would fit in our framework of Lagrangian critical points.

**Gradient descent.** We also investigated the convergence speed of the gradient descent for $\mathrm{SW}_2^2$ for different choices of step-sizes, as shown in Figure 2. We observe that choosing step-sizes close to $\lambda = dN$ (here $d = 2$), as justified in Section 3 does indeed yield a important decrease of the loss at the first few iterations, while lower step-sizes result in slower convergence of the descent, and step-sizes larger than $2dN$ (the threshold above which Proposition 3.2 stops applying) result in divergence of the descent.

## 7. Conclusion

In this work, we have studied critical points of SW objectives with respect to a probability measure $\rho$, by leveraging the notion of Lagrangian critical points in the space of measures. We provided a detailed analysis of the critical points of a flow associated with a non-convex objective distance, in contrast with most of the literature that primarily deals

---

[5]Code available at https://github.com/cvauthier/Critical-Points-of-Sliced-Wasserstein

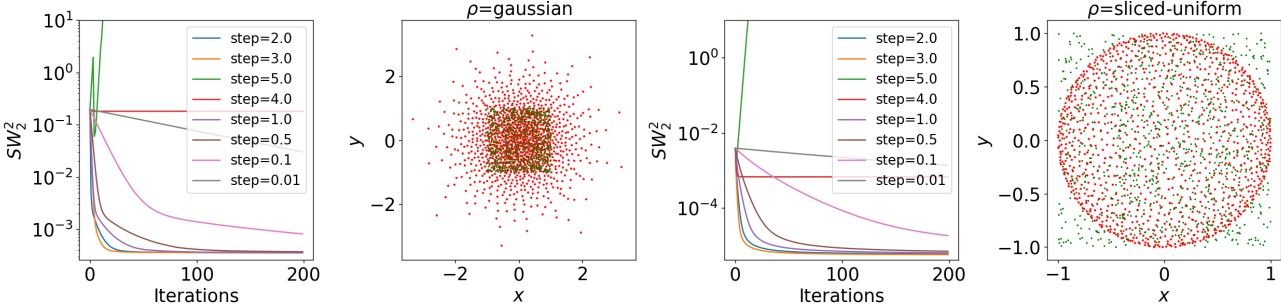

*Figure 2.* Gradient descent of $\mathrm{SW}_2^2$. On a point cloud of $N = 1000$ points for different choices of step-size and $\rho$. Left : convergence speed of gradient descent, where $\rho$ is the normal distribution, for different step-sizes (given in multiples of $N$ in the legend). Center left : Initial point cloud (in green), sampled uniformly in $[-1, 1]^2$, and final point cloud (in red) after 200 iterations with step-size $\lambda = 2N$. Center right and right : same as respectively the left and center left images, but with $\rho$ the sliced-uniform measure (see Proposition 5.1).

with convex ones or that uses functional inequalities.

One limitation of our study is that, while we have defined our framework of Lagrangian critical points for all measures $\mu, \nu \in \mathcal{P}_2(\mathbb{R}^d)$, most of our results require the target $\rho$ to be without atoms (in Section 3, $\rho$ is assumed to be a density, but, as pointed out in Remark 3.4, most of its results can be extended to $\rho$ without atoms). This can limit the applicability of our results to machine learning applications where one often has to work with discrete targets $\rho$. However, our assumptions are sufficient to allow us to tackle many types of singular measures which arise in machine learning and generative modeling, such as densities supported on a lower dimensional manifold of $\mathbb{R}^d$ (which are not absolutely continuous but are without atoms). Furthermore, the fact that our numerical experiments, in which the target measures were discretized, exhibit the behaviors of convergence and instability that our theoretical analysis highlighted, suggests that our results should still be relevant in the cases where the target measure is approximated by a discrete measure. Another limitation is that our main instability result, Proposition 5.2, only holds in dimension $d = 2$ and involves a perturbation which is technically outside our framework of Lagrangian critical points. Generalizing this result to higher dimensions or exhibiting more general unstable Lagrangian critical points could be an avenue for future work.

Finally, many important open questions about critical points of SW remain. First, is it possible to prove that any Wasserstein or Lagrangian critical point $\mu$ of $\mathcal{F} = \frac{1}{2} \mathrm{SW}_2^2(\cdot, \rho)$ which is absolutely continuous must be equal to $\rho$ ? Theorem 4.1 in (Cozzi & Santambrogio, 2024) gives a (very) partial answer to this question: it implies in particular that if $\rho$ is a standard Gaussian and if $\mu$ has finite entropy, then $\mu = \rho$. Second, can we get a better understanding of stable critical points? There exists finitely supported stable critical

points (e.g. the global minimizers of the discretized energy) and we have shown in Proposition 5.2 that stable critical points cannot contain a segment. More generally, one could hope to show that any stable critical point $\mu$ of $\mathcal{F}$ which is atomless must be equal to $\rho$. Third, we note that there exists other proxies of the Wasserstein-$p$ distances based on 1-dimensional projections, such as Max-sliced Wasserstein (Deshpande et al., 2019), SW distances with respect to other probability measures on the unit sphere (Nguyen & Ho, 2024; Rowland et al., 2019; Mahey et al., 2024). Extending our study to these variants of SW is the topic of future research.

## Acknowlegements

QM and AK acknowledge the support of the Agence nationale de la recherche, through the PEPR PDE-AI project (ANR-23-PEIA-0004). CV acknowledges the support of Région Île-de-France through the DIM AI4IDF project.

## Impact Statement

This paper presents work whose goal is to advance the field of Machine Learning. There are many potential societal consequences of our work, none which we feel must be specifically highlighted here.

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

# A. Some useful results

## A.1. Projections of measures without atoms

In this subsection, we prove an useful lemma on measures without atoms. If $\mu$ is a measure on $\mathbb{R}^d$, we say that $\mu$ is *with atomless projections*, which we abbreviate WAP, if its projection $\mu_\theta$ is without atoms for almost every $\theta \in \mathbb{S}^{d-1}$. It is straightforward that if $\mu$ is WAP, then it is without atoms. It turns out that for finite measures, the converse is also true :

**Proposition A.1.** *Let $\mu$ be a finite measure on $\mathbb{R}^d$, then $\mu$ is atomless if and only if it is WAP.*

*Proof.* We have already seen that if $\mu$ has atoms, then it can't be WAP.
Now, for every $k \in \{0, \ldots, d-1\}$, let $AG_k(\mathbb{R}^d)$ be the $k$-th affine Grassmannian of $\mathbb{R}^d$, that is the set of affine subspaces of $\mathbb{R}^d$ of dimension $k$, and for every $k \in \{0, \ldots, d-1\}$ and measure $\mu$ on $\mathbb{R}^d$, we note

$$A_{k,\mu} = \{V \in AG_k(\mathbb{R}^d) \mid \mu(V) > 0\} \tag{24}$$

(in particular, $A_{0,\mu}$ is the set of atoms of $\mu$). Let $\mu$ be a fixed finite measure on $\mathbb{R}^d$ without atoms. We construct by induction a sequence of finite measures $\mu_0 = \mu, \mu_1, \ldots, \mu_{d-1}$ such that for every $k$, $AG_{k,\mu_k} = \emptyset$, and if $k > 0$, then $\mu_k$ is WAP $\Rightarrow \mu_{k-1}$ is WAP. Our first term $\mu_0 = \mu$ satisfies by assumption $A_{0,\mu_0} = \emptyset$. Now assume that we have built $\mu_0, \ldots, \mu_{k-1}$.
If $V_1, \ldots, V_l \in A_{k,\mu_{k-1}}$ are distinct, then

$$\mu_{k-1}(V_1 \cup \ldots \cup V_l) = \sum_{i=1}^{l} \mu_{k-1}(V_i) \tag{25}$$

as the intersection of any subset of these has null $\mu_{k-1}$-measure since $A_{k-1,\mu_{k-1}} = \emptyset$. In particular, the family $(\mu_{k-1}(V))_{V \in A_{k,\mu_{k-1}}}$ is summable, with sum $\leq 1$, and $A_{k,\mu_{k-1}}$ is at most countable. Define

$$\mu_k := \mu_{k-1} - \mu_{k-1|\bigcup A_{k,\mu_{k-1}}} \tag{26}$$

Then, by construction, $A_{k,\mu_k} = \emptyset$. Now, let $\theta \in \mathbb{S}^{d-1}$ be such that $(\mu_{k-1})_\theta$ has an atom : there exists $u \in \mathbb{R}$ such that $(\mu_{k-1})_\theta(\{u\}) > 0$. Assume that $(\mu_k)_\theta(\{u\}) = 0$, then this implies that there exists $V \in A_{k,\mu_{k-1}}$ such that $(\mu_{k-1|V})_\theta(\{u\}) > 0$, that is $\mu_{k-1}(V \cap P_\theta^{-1}(u)) > 0$. Since $A_{k-1,\mu_{k-1}} = \emptyset$, this implies that $V \cap P_\theta^{-1}(u)$ is an affine subspace of dimension $k$, that is $V \subseteq P_\theta^{-1}(u)$, and $\theta \in V^\perp$. This argument thus proves

$$\{\theta \in \mathbb{S}^{d-1} \mid (\mu_{k-1})_\theta \text{ has an atom}\} \subseteq \{\theta \in \mathbb{S}^{d-1} \mid (\mu_k)_\theta \text{ has an atom}\} \cup \{\theta \in \mathbb{S}^{d-1} \mid \exists V \in A_{k,\mu_{k-1}}, \theta \in V^\perp\} \tag{27}$$

Since the second set in the RHS is of null measure (as an at most countable union of sets of null measure), this inclusion implies that if $\mu_k$ is WAP, then $\mu_{k-1}$ is also WAP. This finishes our induction.
Now, we have built our sequence $\mu_0, \ldots, \mu_{d-1}$. But $A_{d-1,\mu_{d-1}} = \emptyset$ implies that $\mu_{d-1}$ is WAP (and that in fact $(\mu_{d-1})_\theta$ is without atoms for *every* $\theta$). Thus, all the measures of the sequence are WAP, and in particular $\mu_0 = \mu$ is WAP. $\qquad\square$

## A.2. Disintegration of measures

We state here the so-called *disintegration theorem*, which we will need in the proofs of our results. Let $X, Y$ be two separable metric spaces. We say that a family $(\mu_x)_{x \in X}$ of probability measures on $\mathcal{P}(Y)$ is a *Borel family of measures* if for every Borel set $B \subset Y$, the map $x \in X \mapsto \mu_x(B)$ is Borel measurable. We say that a separable metric space $X$ is a *Radon space* if for every $\mu \in \mathcal{P}(X)$, every $\varepsilon > 0$ and every Borel set $B \subseteq X$, there exists a compact set $K_\varepsilon \subseteq X$ such that $K_\varepsilon \subseteq B$ and $\mu(B \setminus K_\varepsilon) \leq \varepsilon$. In particular, it is known that Polish spaces (i.e. complete metric separable spaces) are Radon spaces (see (Ambrosio et al., 2005, Section 5.1)), so $\mathbb{R}^d$ is a Radon space.

**Theorem A.2.** *Let $X, Y$ be two Radon separable metric spaces, $\mu \in \mathcal{P}(X)$ and $\pi : X \mapsto Y$ be a measurable map. Let $\nu := \pi_{\#}\mu \in \mathcal{P}(Y)$. Then there exists a Borel family of measures $(\mu_y)_{y \in Y} \subseteq \mathcal{P}(X)$, which is $\nu$-a.e. uniquely defined, such that*

$$\mu_y(X \setminus \pi^{-1}(y)) = 0 \quad \text{for } \nu\text{-a.e. } y \in Y \tag{28}$$

*and, for every measurable map $f : X \mapsto [0, \infty]$,*

$$\int_X f(x)d\mu(x) = \int_Y \int_{\pi^{-1}(y)} f(x)d\mu_y(x)d\nu(y) \tag{29}$$

The statement of this theorem is taken from (Ambrosio et al., 2005, Theorem 5.3.1). In the case where $X$ is of the form $X = Z \times Y$ and $\pi$ is the projection on the second coordinate, we may identify each $\pi^{-1}(y)$ with $Z$, and the theorem reformulates as : there exists a Borel family of measures $(\mu_y)_{y \in Y} \subseteq \mathcal{P}(Z)$, which is $\nu$-a.e. uniquely defined, such that for every measurable map $f : Z \times Y \mapsto [0, \infty]$, $\int_{Z \times Y} f(z, y) d\mu(z, y) = \int_Y \int_Z f(z, y) d\mu_y(z) d\nu(y)$.

## B. Proofs

### B.1. Proof of Proposition 3.1

First, consider a probability density $\rho \in \mathcal{P}_p(\mathbb{R})$, with cumulative distribution function $F_\rho : \mathbb{R} \mapsto [0; 1]$. Let $\mu_X = \frac{1}{N} \sum_{i=1}^N \delta_{x_i}$ the uniform empirical measure associated to $X = (x_1, ..., x_N) \in \mathbb{R}^N$. For every $i \in \{1, ..., N\}$, we define $V_i = F_\rho^{-1}([\frac{i-1}{N}; \frac{i}{N}])$ the $i$-th Power cell associated to $\rho$. Then the properties of one-dimensional optimal transport imply that, for every $X = (x_1, ..., x_N) \in \mathbb{R}^N$ with $x_{\sigma(1)} \leq ... \leq x_{\sigma(N)}$, $\sigma \in \mathfrak{S}_N$, we have

$$G(X) := \frac{1}{p} W_p^p(\mu_X, \rho) = \frac{1}{p} \sum_{i=1}^N \int_{V_i} |x_{\sigma(i)} - x|^p d\rho(x) = \frac{1}{p} \sum_{i=1}^N \int_{V_{\sigma^{-1}(i)}} |x_i - x|^p d\rho(x). \tag{30}$$

We can then easily see that when $p > 1$, $G$ is $C^1$ on the complement of the generalized diagonal $\Delta_N = \{(x_1, ..., x_N) \in \mathbb{R}^N \mid \exists i \neq j, x_i = x_j\}$, and its partial derivatives are given by

$$\partial_i G(x_1, ..., x_N) = \int_{V_{\sigma^{-1}(i)}} \text{sgn}(x_i - x)|x_i - x|^{p-1} d\rho(x), \tag{31}$$

where $\sigma \in \mathfrak{S}_N$ is such that $x_{\sigma(1)} < ... < x_{\sigma(N)}$. In the particular case where $p = 2$, the partial derivatives take the simpler form

$$\partial_i G(x_1, ..., x_N) = \int_{V_{\sigma^{-1}(i)}} (x_i - x) d\rho(x) = \frac{1}{N}(x_i - b_{\sigma^{-1}(i)}) \tag{32}$$

with $b_i = N \int_{V_i} x d\rho(x)$ the barycenter of the $i$-th Power cell $V_i$.

With these considerations on one-dimensional measures in mind, we can now move on to prove Proposition 3.1. For this, we will need the following lemma.

**Lemma B.1.** *If $p \geq 2$, $\rho \in \mathcal{P}_p(\mathbb{R})$ is a probability density and $X = (x_1, ..., x_N) \in \Delta_N$ with $x_{\sigma(1)} < ... < x_{\sigma(N)}$, $\sigma \in \mathfrak{S}_N$, and $H = (h_1, ..., h_N) \in \mathbb{R}^N$ is a perturbation such that $X + H$ has the same ordering $\sigma$ as $X$, then writing $R_1 G(X, H) = G(X + H) - G(X) - \langle \nabla G(X) | H \rangle$ we have*

$$|R_1 G(X, H)| \leq 2^{p-2}(p-1) \sum_{i=1}^N |h_i|^p + |h_i|^2 \int |x_i - x|^{p-2} d\rho(x) \tag{33}$$

*(this is a finite quantity since $\rho$ has finite order $p$ moments).*

*Proof.* Consider the function $f(x) = |x|^p$. Since $p \geq 2$, we see that $f$ is $C^2$ and that $f'(x) = px|x|^{p-2}$, $f''(x) = p(p-1)|x|^{p-2}$. As a consequence, applying Taylor's theorem, for every $x, h \in \mathbb{R}$,

$$f(x + h) - f(x) - f'(x)h = \int_x^{x+h} f''(t)(x - t) dt \tag{34}$$

$$|f(x + h) - f(x) - f'(x)h| \leq \int_x^{x+h} |f''(t)(x - t)| dt \tag{35}$$

$$\leq \int_x^{x+h} p(p-1) \max(|x|, |x + h|)^{p-2} |h| dt \tag{36}$$

$$\leq p(p-1)|h|^2 (|x| + |h|)^{p-2} \tag{37}$$

$$\leq 2^{p-2} p(p-1)|h|^2 (|x|^{p-2} + |h|^{p-2}) \tag{38}$$

$$\leq 2^{p-2} p(p-1)(|h|^p + |h|^2 |x|^{p-2}) \tag{39}$$

Therefore, since $X + H$ and $X$ have the same ordering $\sigma$,

$$R_1 G(X, H) = \frac{1}{p} \sum_{i=1}^{N} \int_{V_{\sigma^{-1}(i)}} (|x_i + h_i - x|^p - |x_i - x|^p - p\,\text{sgn}(x_i - x)|x_i - x|^{p-1}) d\rho(x) \tag{40}$$

$$|R_1 G(X, H)| \leq \frac{1}{p} \sum_{i=1}^{N} \int_{V_{\sigma^{-1}(i)}} 2^{p-2} p(p-1)(|h_i|^p + |h_i|^2 |x_i - x|^{p-2}) d\rho(x) \tag{41}$$

$$\leq 2^{p-2}(p-1) \sum_{i=1}^{N} \int_{V_{\sigma^{-1}(i)}} |h_i|^p + |h_i|^2 |x_i - x|^{p-2} d\rho(x) \tag{42}$$

$$\leq 2^{p-2}(p-1) \sum_{i=1}^{N} |h_i|^p + |h_i|^2 \int |x_i - x|^{p-2} d\rho(x) \tag{43}$$

$\square$

Now we can prove Proposition 3.1.

*Proof (Proposition 3.1).* First, let's introduce the following definitions : for every $\epsilon > 0$ let

$$\Theta_\epsilon := \{\theta \in \mathbb{S}^{d-1} \mid \exists i \neq j, |\langle X_i - X_j | \theta \rangle| \leq \epsilon\} \tag{44}$$

and for every $\theta \in \mathbb{S}^{d-1}$ define the function $G_\theta : X \in \mathbb{R}^N \mapsto \frac{1}{p} W_p^p(\mu_X, P_{\theta\#}\rho)$ For every point cloud $X \in (\mathbb{R}^d)^N$ and every $\theta \in \mathbb{S}^{d-1}$, let $\sigma_{\theta,X} \in \mathfrak{S}_N$ be a (not necessarily unique) permutation such that $\langle X_{\sigma_{\theta,X}(1)} | \theta \rangle \leq \dots \leq \langle X_{\sigma_{\theta,X}(N)} | \theta \rangle$, and let

$$\tilde{\nabla}_{X_i} F(X) := \int_{\mathbb{S}^{d-1}} \int_{V_{\theta, \sigma_{\theta,X}^{-1}(i)}} \text{sgn}(\langle X_i | \theta \rangle - x) |\langle X_i | \theta \rangle - x|^{p-1} \theta dP_{\theta\#}\rho(x) d\theta \tag{45}$$

We want to prove that if $X \notin \Delta_N$, $F$ is differentiable at $X$ and $\nabla F(X) = \tilde{\nabla} F(X)$.

Let $\epsilon > 0$ be fixed. We see that if $\|H\| \leq \epsilon$, then for every $\theta \notin \Theta_{2\epsilon}$, $\sigma_{\theta,X+H} = \sigma_{\theta,X}$. Furthermore we know that there exists $C_0 = C_0(X) > 0$ such that

$$\mathcal{U}_{\mathbb{S}^{d-1}}(\Theta_\epsilon) \leq C_0 \epsilon \tag{46}$$

where $\mathcal{U}_{\mathbb{S}^{d-1}}$ is the uniform distribution (i.e. the normalized volume measure) on $\mathbb{S}^{d-1}$. We now consider a perturbation $H$ such that $\|H\| \leq \epsilon/2$. We have

$$F(X + H) - F(X) - \langle \tilde{\nabla} F(X) | H \rangle = A(H) + B(H) + C(H) \tag{47}$$

with

$$A(H) = \int_{\Theta_\epsilon^c} (G_\theta(P_\theta(X + H)) - G_\theta(P_\theta(X)) - \langle P_\theta(H) | \nabla G_\theta(P_\theta(X)) \rangle) d\theta \tag{48}$$

$$B(H) = \int_{\Theta_\epsilon} (G_\theta(P_\theta(X + H)) - G_\theta(P_\theta(X))) d\theta \tag{49}$$

$$C(H) = -\int_{\Theta_\epsilon} \langle P_\theta(H) | \nabla G_\theta(P_\theta(X)) \rangle d\theta \tag{50}$$

When $\theta \in \Theta_\epsilon^c$, we have $\sigma_{\theta,X+H} = \sigma_{\theta,X}$ and we can apply lemma B.1 to $G_\theta$ to obtain that

$$|G_\theta(P_\theta(X + H)) - G_\theta(P_\theta(X)) - \langle P_\theta(H) | \nabla G_\theta(P_\theta(X)) \rangle| \leq C\|H\|^2 \tag{51}$$

with a constant $C$ that is uniform on $\theta$ and depends only on $X$, $\rho$, $\epsilon$ and $p$ (indeed, the moments of $P_{\theta\#}\rho$ are bounded by those of $\rho$). Therefore we deduce that

$$A(H) = o(\|H\|) \tag{52}$$

Now, notice that

$$|\partial_i G_\theta(P_\theta(X))| \leq \int_{V_{\theta,\sigma_{\theta,X}^{-1}(i)}} |\langle X_i|\theta\rangle - x|^{p-1} dP_{\theta\#}\rho(x) \tag{53}$$

so

$$\sum_{i=1}^N |\partial_i G_\theta(P_\theta(X))| \leq \sum_{i=1}^N \int_{V_{\theta,\sigma_{\theta,X}^{-1}(i)}} |\langle X_i|\theta\rangle - x|^{p-1} dP_{\theta\#}\rho(x) = \mathrm{W}_{p-1}^{p-1}(\mu_{P_\theta(X)}, P_{\theta\#}\rho) \leq \mathrm{W}_{p-1}^{p-1}(\mu_X, \rho) \tag{54}$$

therefore we deduce that

$$|C(H)| \leq C_0\epsilon\|H\|\mathrm{W}_{p-1}^{p-1}(\mu_X, \rho) \tag{55}$$

Finally, for a generic $\theta$, using the mean value inequality on $f(x) = x^p$ with $f'(x) = px^{p-1}$, and using the shorthand notations $W_p(X) = W_p(\mu_{P_\theta(X)}, P_{\theta\#}\rho)$, we have

$$|G_\theta(P_\theta(X+H)) - G_\theta(P_\theta(X))| = |W_p(X+H)^p - W_p(X)^p| \tag{56}$$

$$\leq |W_p(X+H) - W_p(X)| \sup_{[W_p(X), W_p(X+H)]} |f'| \tag{57}$$

$$\leq p|W_p(X+H) - W_p(X)| \max(W_p(X), W_p(X+H))^{p-1} \tag{58}$$

Now, by the triangle inequality, we have

$$|W_p(X+H) - W_p(X)| \leq W_p(P_{\theta\#}\mu_{X+H}, P_{\theta\#}\mu_X) \leq W_p(\mu_{X+H}, \mu_X) \leq \|H\| \tag{59}$$

And similarly $W_p(X) \leq W_p(\mu_X, \rho)$ and

$$W_p(X+H) \leq W_p(X) + W_p(P_{\theta\#}\mu_{X+H}, P_{\theta\#}\mu_X) \leq W_p(X) + \|H\| \leq W_p(\mu_X, \rho) + \epsilon \tag{60}$$

Therefore, we have

$$|G_\theta(P_\theta(X+H)) - G_\theta(P_\theta(X))| \leq C\|H\| \tag{61}$$

with a constant $C$ which is uniform in $\theta$ and depends only on $p$, $\epsilon$ and $\mathrm{W}(\mu_X, \rho)$. Therefore

$$|B(H)| \leq C_0 C\epsilon\|H\| \tag{62}$$

Thus, we have proven that

$$F(X+H) - F(X) - \langle\tilde{\nabla}F(X)|H\rangle = o(\|H\|) \tag{63}$$

which shows that $\nabla F(X) = \tilde{\nabla}F(X)$. To show the continuity of $\nabla F$, let $X^k \in (\mathbb{R}^d)^N$ be a sequence converging to $X$, with $X^k, X \notin \Delta_N$. Recall that for every $i \in \{1, \ldots, N\}$, we have

$$\nabla_{X_i} F(X^k) = \int_{\mathbb{S}^{d-1}} \int_{V_{\theta,\sigma_{\theta,X^k}^{-1}(i)}} \mathrm{sgn}(\langle X_i^k|\theta\rangle - x)|\langle X_i^k|\theta\rangle - x|^{p-1}\theta dP_{\theta\#}\rho(x)d\theta \tag{64}$$

Let $\theta \in \mathbb{S}^{d-1}$ be such that $\langle X_i|\theta\rangle \neq \langle X_j|\theta\rangle$ for every $i \neq j$, then there exists $k_0$ such that for every $k \geq k_0$, $\sigma_{\theta,X^k} = \sigma_{\theta,X}$. In particular, this implies that for every $i$, $\sigma_{\theta,X^k}^{-1}(i) = \sigma_{\theta,X}^{-1}(i)$, and thus

$$\int_{V_{\theta,\sigma_{\theta,X^k}^{-1}(i)}} \mathrm{sgn}(\langle X_i^k|\theta\rangle - x)|\langle X_i^k|\theta\rangle - x|^{p-1}\theta dP_{\theta\#}\rho(x) = \int_{V_{\theta,\sigma_{\theta,X}^{-1}(i)}} \mathrm{sgn}(\langle X_i^k|\theta\rangle - x)|\langle X_i^k|\theta\rangle - x|^{p-1}\theta dP_{\theta\#}\rho(x)$$

$$\tag{65}$$

$$\xrightarrow[k\to\infty]{} \int_{V_{\theta,\sigma_{\theta,X}^{-1}(i)}} \mathrm{sgn}(\langle X_i|\theta\rangle - x)|\langle X_i|\theta\rangle - x|^{p-1}\theta dP_{\theta\#}\rho(x)$$

$$\tag{66}$$

where the limit is obtained by dominated convergence, using the fact that the sequence $X^k$ is bounded and that $P_{\theta\#}\rho$ has finite moments of order $p-1$. Moreover, since for every $k$ and $i$, we have

$$\left| \int_{V_{\theta,\sigma^{-1}_{\theta,X^k}(i)}} \mathrm{sgn}(\langle X_i^k|\theta\rangle - x)|\langle X_i^k|\theta\rangle - x|^{p-1}\theta dP_{\theta\#}\rho(x) \right| \leq \int |\langle X_i^k|\theta\rangle - x|^{p-1}dP_{\theta\#}\rho(x) \tag{67}$$

$$\leq 2^{p-1}(|\langle X_i^k|\theta\rangle|^{p-1} + \int |x|^{p-1}dP_{\theta\#}\rho(x)) \tag{68}$$

$$\leq 2^{p-1}(|X_i^k|^{p-1} + \int |x|^{p-1}d\rho(x)) \tag{69}$$

and since the sequence $X^k$ is bounded and $\rho$ has finite moments of order $p-1$, this implies by dominated convergence that $\lim_{k\to\infty} \nabla_{X_i} F(X^k) = \nabla_{X_i} F(X)$ for every $i$. This proves the continuity of $\nabla F$. Finally, in the case $p=2$, the expression of $\nabla_{X_i} F$ simplifies as

$$\nabla_{X_i} F(X) = \int_{\mathbb{S}^{d-1}} \int_{V_{\theta,\sigma^{-1}_{\theta,X}(i)}} \mathrm{sgn}(\langle X_i|\theta\rangle - x)|\langle X_i|\theta\rangle - x|\theta dP_{\theta\#}\rho(x)d\theta \tag{70}$$

$$= \int_{\mathbb{S}^{d-1}} \int_{V_{\theta,\sigma^{-1}_{\theta,X}(i)}} (\langle X_i|\theta\rangle - x)\theta dP_{\theta\#}\rho(x)d\theta \tag{71}$$

$$= \int_{\mathbb{S}^{d-1}} \frac{1}{N}\langle X_i|\theta\rangle\theta d\theta - \int_{\mathbb{S}^{d-1}} \int_{V_{\theta,\sigma^{-1}_{\theta,X}(i)}} x dP_{\theta\#}\rho(x)\theta d\theta \tag{72}$$

$$= \frac{1}{N}\left(\frac{1}{d}X_i - \int_{\mathbb{S}^{d-1}} b_{\theta,\sigma^{-1}_{\theta,X}(i)}\theta d\theta\right) \tag{73}$$

where we used the definition of the $b_{\theta,i}$ in the last line. $\qquad\square$

As a side note, remark that $F$ is actually twice differentiable almost everywhere, as a consequence of the following semi-concavity property for $F$:

**Proposition B.2.** $F$ is $\frac{1}{Nd}$-semiconcave (i.e. $F - \frac{1}{2Nd}\|\cdot\|^2$ is concave).

*Proof.* Indeed, $F(X) - \frac{1}{2Nd}\|X\|^2 = \int_{\mathbb{S}^{d-1}} \frac{1}{2}\mathrm{W}_2^2(\mu_{P_\theta(X)}, P_{\theta\#}\rho) - \frac{1}{2N}\|P_\theta(X)\|^2 d\theta$ for every $X \in \mathbb{R}^{d\times N}$, and we use the fact that the projection $P_\theta$ is linear and that $Y \in \mathbb{R}^N \mapsto \frac{1}{2}\mathrm{W}_2^2(\mu_Y, P_{\theta\#}\rho)$ is $\frac{1}{N}$-semiconcave (see for example Proposition 1, (Mérigot et al., 2021)) $\qquad\square$

### B.2. Proof of Proposition 3.2

To prove the descent lemma Proposition 3.2, we first need to prove that $F$ is smooth.

**Proposition B.3.** *For every $X, Y \in (\mathbb{R}^d)^N \setminus \Delta_N$, we have*

$$F(Y) \leq F(X) + \langle \nabla F(X)|Y - X\rangle + \frac{1}{2Nd}\|X - Y\|^2 \tag{74}$$

*Proof.* First, let $\theta \in \mathbb{S}^{d-1}$ be fixed, such that $\langle X_i|\theta\rangle \neq \langle X_j|\theta\rangle$ for every $i \neq j$. Then, since the map which sends $V_{\theta,i}$ to

$\langle Y_{\sigma_{X,\theta}(i)}|\theta\rangle$ is a (not necessarily optimal) transport map from $\rho_\theta$ to $\mu_{P_\theta(Y)}$, we have

$$W_2^2(\mu_{P_\theta(Y)}, \rho_\theta) \leq \sum_{i=1}^N \int_{V_{\theta,\sigma_{X,\theta}^{-1}(i)}} |\langle Y_i|\theta\rangle - x|^2 d\rho_\theta(x) \tag{75}$$

$$\leq \sum_{i=1}^N \int_{V_{\theta,\sigma_{X,\theta}^{-1}(i)}} |\langle Y_i|\theta\rangle - \langle X_i|\theta\rangle + \langle X_i|\theta\rangle - x|^2 d\rho_\theta(x) \tag{76}$$

$$\leq \frac{1}{N} \sum_{i=1}^N \langle Y_i - X_i|\theta\rangle^2 + \sum_{i=1}^N \int_{V_{\theta,\sigma_{X,\theta}^{-1}(i)}} 2\langle Y_i - X_i|\theta\rangle(\langle X_i|\theta\rangle - x)d\rho_\theta(x) + W_2^2(\mu_{P_\theta(X)}, \rho_\theta) \tag{77}$$

$$\leq \frac{1}{N} \sum_{i=1}^N \langle Y_i - X_i|\theta\rangle^2 + \sum_{i=1}^N \frac{2}{N}\langle Y_i - X_i|\theta\rangle(\langle X_i|\theta\rangle - b_{\theta,\sigma_{X,\theta}^{-1}(i)}) + W_2^2(\mu_{P_\theta(X)}, \rho_\theta) \tag{78}$$

Integrating over the sphere we have

$$\mathrm{SW}_2^2(\mu_Y, \rho) \leq \frac{1}{N} \sum_{i=1}^N \int_{\mathbb{S}^{d-1}} \langle Y_i - X_i|\theta\rangle^2 d\theta + \frac{2}{N} \sum_{i=1}^N \int_{\mathbb{S}^{d-1}} \langle Y_i - X_i|\theta\rangle(\langle X_i|\theta\rangle - b_{\theta,\sigma_{X,\theta}^{-1}(i)})d\theta + \mathrm{SW}_2^2(\mu_X, \rho) \tag{79}$$

$$\leq \frac{1}{Nd} \sum_{i=1}^N \|Y_i - X_i\|^2 + \sum_{i=1}^N \left\langle Y_i - X_i \mid \frac{2}{N} \int_{\mathbb{S}^{d-1}} (\langle X_i|\theta\rangle - b_{\theta,\sigma_{X,\theta}^{-1}(i)})\theta d\theta \right\rangle + \mathrm{SW}_2^2(\mu_X, \rho) \tag{80}$$

In the RHS of the last inequality, we recognize the expression of the gradient of $F$ which we recall is $\nabla_{X_i} F = \frac{1}{N} \int_{\mathbb{S}^{d-1}} (\langle X_i|\theta\rangle - b_{\theta,\sigma_{X,\theta}^{-1}(i)})\theta d\theta$. Therefore, substituting it gives the intended result

$$F(Y) \leq \frac{1}{2Nd}\|X - Y\|^2 + \langle Y - X|\nabla F(X)\rangle + F(X). \tag{81}$$
$\square$

Now, we can prove Proposition 3.2. Equation (10) is obtained directly from Equation (74) by taking $Y := X - \lambda \nabla F(X)$.

### B.3. Proof of Proposition 3.3

We will first need to prove the following lemmas :

**Lemma B.4.** *Let $\rho \in \mathcal{P}([a,b])$ be an absolutely continuous probability measure, with density (which we will also denote $\rho$) bounded from above by $\beta > 0$. Then the barycenter $x_0 = \int_a^b x d\rho(x)$ of $\rho$ satisfies $|x_0 - a|, |x_0 - b| \geq \frac{1}{2\beta}$.*

*Proof.* Since $\rho \leq \beta$, integrating $\rho$ on $[a,b]$, we note that $\frac{1}{\beta} \leq b - a$. Let $\rho_0 \in \mathcal{P}([a,b])$ be the probability with density $\beta$ on $[a, a + 1/\beta]$ and $0$ on $[a + 1/\beta, b]$. Its cumulative distribution function is thus

$$F_{\rho_0}(x) = \begin{cases} \beta(x-a) & \text{if } x \in [a, a+1/\beta] \\ 1 & \text{if } x \geq a + \frac{1}{\beta} \end{cases} \tag{82}$$

and, since $\rho \leq \beta$, we have $F_\rho \leq F_{\rho_0}$ on $[a,b]$. Thus, the quantile functions of $\rho, \rho_0$ satisfy $F_\rho^{-1} \geq F_{\rho_0}^{-1}$ (this follows directly from their definition), and we have

$$x_0 - a = \int_a^b (x-a)d\rho(x) = \int_0^1 (F_\rho^{-1}(x) - a)dx \tag{83}$$

$$\geq \int_0^1 (F_{\rho_0}^{-1}(x) - a)dx = \int_a^b (x-a)d\rho_0(x) = \int_a^{a+\frac{1}{\beta}} \beta(x-a)dx = \frac{1}{2\beta} \tag{84}$$

where we used the fact that $\mu = F_{\mu\#}^{-1} \mathcal{L}_{[0,1]}^1$ for any probability measure $\mu$ on the real line (see (Santambrogio, 2015, Proposition 2.2)). Similarly, we can show that $b - x_0 \geq \frac{1}{2\beta}$. $\square$

**Lemma B.5.** *For every $X \in (\mathbb{R}^d)^N \setminus \Delta_N$, we have for every $i \neq j$,*

$$N\langle \nabla_{X_i} F(X) - \nabla_{X_j} F(X) | X_i - X_j \rangle \leq \frac{1}{d} \|X_i - X_j\|^2 \tag{85}$$

*If we further assume that there exists $\beta > 0$ bounding from above the density of $\rho_\theta$ for every $\theta \in \mathbb{S}^{d-1}$, then there exists $C = C(d)$ such that for every $i \neq j$,*

$$N\langle \nabla_{X_i} F(X) - \nabla_{X_j} F(X) | X_i - X_j \rangle \leq \frac{1}{d} \|X_i - X_j\|^2 - \frac{C}{N\beta} \|X_i - X_j\| \tag{86}$$

*Proof.* Using the notations of Proposition 3.1 and Equation (9), we have

$$N\langle \nabla_{X_i} F(X) - \nabla_{X_j} F(X) | X_i - X_j \rangle = \frac{1}{d} \|X_i - X_j\|^2 - \int_{\mathbb{S}^{d-1}} (b_{\theta, \sigma_{X,\theta}^{-1}(i)} - b_{\theta, \sigma_{X,\theta}^{-1}(j)}) \langle \theta | X_i - X_j \rangle d\theta \tag{87}$$

By symmetry, we have in fact

$$N\langle \nabla_{X_i} F(X) - \nabla_{X_j} F(X) | X_i - X_j \rangle = \frac{1}{d} \|X_i - X_j\|^2 - 2 \int_{\{\langle \theta | X_i - X_j \rangle > 0\}} (b_{\theta, \sigma_{X,\theta}^{-1}(i)} - b_{\theta, \sigma_{X,\theta}^{-1}(j)}) \langle \theta | X_i - X_j \rangle d\theta \tag{88}$$

Indeed, for every $\theta \in \mathbb{S}^{d-1}$, we can check that we have $\sigma_{X,-\theta}^{-1}(k) = N + 1 - \sigma_{X,\theta}^{-1}(k)$ and $b_{-\theta, k} = b_{\theta, N+1-k}$ for every $k = 1, \ldots, N$. However, if $\theta \in \mathbb{S}^{d-1}$ is such that $\langle \theta | X_i - X_j \rangle > 0$, then we have $\sigma_{X,\theta}^{-1}(i) > \sigma_{X,\theta}^{-1}(j)$, and thus $b_{\theta, \sigma_{X,\theta}^{-1}(i)} - b_{\theta, \sigma_{X,\theta}^{-1}(j)} \geq 0$. Therefore the integrand in the right-hand side of (88) is nonnegative, and we deduce from this

$$N\langle \nabla_{X_i} F(X) - \nabla_{X_j} F(X) | X_i - X_j \rangle \leq \frac{1}{d} \|X_i - X_j\|^2 \tag{89}$$

This proves (85). Now, if we assume that there exists $\beta > 0$ such that $\rho_\theta \leq \beta$ for every $\theta \in \mathbb{S}^{d-1}$, then we have

$$b_{\theta, \sigma_{X,\theta}^{-1}(i)} - b_{\theta, \sigma_{X,\theta}^{-1}(j)} \geq \frac{1}{N\beta} \tag{90}$$

Indeed, for every $k = 1, \ldots, N$, the distance separating the barycenter $b_{\theta, k}$ from the boundary of its corresponding Power cell $V_{\theta, k}$ is at least $\frac{1}{2\beta N}$, which we see by applying Lemma B.4 to the probability measure $N\rho_{\theta | V_{\theta, k}}$. In particular, since $\langle \theta | X_i - X_j \rangle$ is also positive, we have

$$\langle \theta | X_i - X_j \rangle (b_{\theta, \sigma_{X,\theta}^{-1}(i)} - b_{\theta, \sigma_{X,\theta}^{-1}(j)}) \geq \frac{1}{N\beta} \langle \theta | X_i - X_j \rangle \tag{91}$$

Injecting this into Equation (88), we obtain the inequality

$$N\langle \nabla_{X_i} F(X) - \nabla_{X_j} F(X) | X_i - X_j \rangle \leq \frac{1}{d} \|X_i - X_j\|^2 - 2 \int_{\{\langle \theta | X_i - X_j \rangle > 0\}} \frac{1}{N\beta} \langle \theta | X_i - X_j \rangle d\theta \tag{92}$$

$$\leq \frac{1}{d} \|X_i - X_j\|^2 - \frac{2}{N\beta} \|X_i - X_j\| \int_{\{\langle \theta | \theta_0 \rangle > 0\}} \langle \theta | \theta_0 \rangle d\theta \tag{93}$$

$$\leq \frac{1}{d} \|X_i - X_j\|^2 - \frac{C}{N\beta} \|X_i - X_j\| \tag{94}$$

where $\theta_0 := \frac{X_i - X_j}{\|X_i - X_j\|}$, and where $C := 2 \int_{\{\langle \theta | \theta_0 \rangle > 0\}} \langle \theta | \theta_0 \rangle d\theta > 0$. Note that, by symmetry, $C$ does not depend on $\theta_0 \in \mathbb{S}^{d-1}$ and depends only on $d$. This proves (86). $\qquad\square$

We can now prove the proposition.

*Proof (Proposition 3.3).* If $i \neq j$, then we have

$$\langle Y_i - Y_j | X_i - X_j \rangle = \langle X_i - X_j - \lambda(\nabla_{X_i} F(X) - \nabla_{X_j} F(X)) | X_i - X_j \rangle \tag{95}$$

$$= \|X_i - X_j\|^2 - \lambda \langle \nabla_{X_i} F(X) - \nabla_{X_j} F(X) | X_i - X_j \rangle \tag{96}$$

$$\geq \|X_i - X_j\|^2 - \frac{\lambda}{N}\left(\frac{1}{d}\|X_i - X_j\|^2\right) = \left(1 - \frac{\lambda}{Nd}\right)\|X_i - X_j\|^2 \tag{97}$$

where we used (85) from Lemma B.5 to obtain the last line. In particular, if $\lambda \in (0, Nd)$, then $\langle Y_i - Y_j | X_i - X_j \rangle > 0$ and thus $Y_i \neq Y_j$, for every $i \neq j$. Therefore, $Y \notin \Delta_N$. This proves the first part of the proposition.

Now, assuming that there exists $\beta > 0$ such that the density of $\rho_\theta$ is bounded from above by $\beta$ for every $\theta \in \mathbb{S}^{d-1}$, we then have for every $i \neq j$,

$$\|Y_i - Y_j\|^2 = \|(X_i - X_j) - \lambda(\nabla_{X_i} F(X) - \nabla_{X_j} F(X))\|^2 \tag{98}$$

$$= \|X_i - X_j\|^2 - 2\lambda\langle \nabla_{X_i} F(X) - \nabla_{X_j} F(X) | X_i - X_j \rangle + \lambda^2 \|\nabla_{X_i} F(X) - \nabla_{X_j} F(X)\|^2 \tag{99}$$

$$\geq \|X_i - X_j\|^2 - 2\lambda\langle \nabla_{X_i} F(X) - \nabla_{X_j} F(X) | X_i - X_j \rangle \tag{100}$$

$$\geq \|X_i - X_j\|^2 - 2\frac{\lambda}{N}\left(\frac{1}{d}\|X_i - X_j\|^2 - \frac{C}{N\beta}\|X_i - X_j\|\right) \tag{101}$$

where we used (86) from Lemma B.5 in the last line. Thus, we have proved

$$\|Y_i - Y_j\|^2 \geq \|X_i - X_j\|^2 + 2\frac{\lambda}{N}\|X_i - X_j\|\left(\frac{C}{N\beta} - \frac{1}{d}\|X_i - X_j\|\right) \tag{102}$$

Now :

- If $\|X_i - X_j\| \leq \frac{dC}{N\beta}$, we have directly $\|Y_i - Y_j\| > \|X_i - X_j\|$ from Equation (102).

- If $X$ is a critical point, we have $\nabla F(X) = 0$ and thus $Y = X$. Therefore, Equation (102) yields

$$0 \geq 2\frac{\lambda}{N}\|X_i - X_j\|\left(\frac{C}{N\beta} - \frac{1}{d}\|X_i - X_j\|\right) \tag{103}$$

which implies

$$\frac{1}{d}\|X_i - X_j\| \geq \frac{C}{N\beta} \tag{104}$$

$\square$

As a side note, observe that if we consider the continuous time gradient flow

$$\begin{cases} X(t = 0) = X_0 & \text{with } X_0 \in (\mathbb{R}^d)^N \setminus \Delta_N \\ \dot{X}(t) = -\nabla F(X(t)) & \text{for } t > 0 \end{cases} \tag{105}$$

then Lemma B.5 implies that for every $t > 0$ at which the flow is well-defined, for every $i \neq j$,

$$\frac{d}{dt}\frac{1}{2}\|X_i - X_j\|^2 = -\langle \nabla_{X_i} F(X) - \nabla_{X_j} F(X) | X_i - X_j \rangle \tag{106}$$

$$\geq -\frac{1}{Nd}\|X_i - X_j\|^2 + \frac{C}{N^2\beta}\|X_i - X_j\| \tag{107}$$

$$\geq \frac{\|X_i - X_j\|}{N}\left(\frac{C}{N\beta} - \frac{1}{d}\|X_i - X_j\|\right) \tag{108}$$

where we used (86) to obtain the second line, and, in particular,

$$\frac{d}{dt}\|X_i - X_j\|^2 > 0 \tag{109}$$

whenever $\|X_i - X_j\| \leq \frac{dC}{N\beta}$. This implies that :

- If $\|X_i - X_j\| \geq \frac{dC}{N\beta}$ at $t = 0$, then this inequality must stay true at every $t > 0$.

- If $\|X_i - X_j\| \leq \frac{dC}{N\beta}$ at $t = 0$, then $\|X_i - X_j\|$ increases until it is greater or equal than $\frac{dC}{N\beta}$, and does not become lower than this threshold afterwards.

Thus, we see that the continuous time gradient flow is also well-behaved, in that it will tend to stay far away from the generalized diagonal $\Delta_N$.

### B.4. Extensions of the results of Section 3

If, instead of assuming that $\rho$ is absolutely continuous with respect to the Lebesgue measure, we simply assume that $\rho$ is without atoms, then, by Proposition A.1, the projection $\rho_\theta$ is without atoms for every $\theta \in \mathbb{S}^{d-1} \setminus N_\rho$ where $N_\rho \subseteq \mathbb{S}^{d-1}$ is some set of directions of measure zero. In particular, for every $\theta \notin N_\rho$, the cumulative distribution function $F_{\rho_\theta}$ is continuous, and we can define the Power cells $V_{\theta,i} := F_{\rho_\theta}^{-1}\left(\left[\frac{i-1}{N}, \frac{i}{N}\right]\right)$ and the barycenters $b_{\theta,i} := N \int_{V_{\theta,i}} x d\rho_\theta(x)$ for every $i \in \{1, \ldots, N\}$. Then Proposition 3.1, Proposition 3.2 and the first part of Proposition 3.3 extend to $\rho$, with the exact same statement. Indeed, their proofs as stated in Appendix B.1, Appendix B.2 and Appendix B.3 work exactly the same (with the difference that we only consider directions $\theta$ that are not in $N_\rho$).

In fact, we can find extensions of Proposition 3.1 and Proposition 3.2 when we only assume that $\rho \in \mathcal{P}_2(\mathbb{R}^d)$. The difficulty is that, since the projections $\rho_\theta$ can no longer be assumed to be without atoms, we can't characterize the optimal transport between $\rho_\theta$ and $P_{\theta\#}\mu_X$ in terms of Power cells. For this, we first recall the following well-known results on optimal transport in 1D :

**Theorem B.6.** *Let $\mu, \nu \in \mathcal{P}_2(\mathbb{R})$, then the so-called monotone transport plan between $\mu$ and $\nu$, given by $\gamma_{mon} := (F_\mu^{-1}, F_\nu^{-1})_\# \mathcal{L}_{|[0,1]}^1$, is the unique optimal transport plan between $\mu$ and $\nu$ for any cost of the form $c(x,y) = h(x-y)$ with $h : \mathbb{R} \mapsto [0, \infty)$ strictly convex. Furthermore, $\gamma_{mon}$ is the unique transport plan $\gamma \in \Pi(\mu, \nu)$ which satisfies*

$$\forall (x,y), (x',y') \in \mathrm{spt}(\gamma), x < x' \Rightarrow y \leq y' \tag{110}$$

We refer to (Santambrogio, 2015, Chapter 2) for the detailed statement and proof of these results. In the special case where one of the two mesures is a point cloud, we then have the following lemma :

**Lemma B.7.** *Let $\rho \in \mathcal{P}_2(\mathbb{R})$ and $N > 0$. Then there exists an unique family of probability measures $\rho_1, \ldots, \rho_N \in \mathcal{P}_2(\mathbb{R})$ such that :*

- $\rho = \frac{1}{N} \sum_{i=1}^N \rho_i$

- *For every $i < j$, $x_i \in \mathrm{spt}(\rho_i)$ and $x_j \in \mathrm{spt}(\rho_j)$, we have $x_i \leq x_j$*

- *For every $X = (x_1, \ldots, x_N) \in \mathbb{R}^N$ with $x_1 < \ldots < x_N$, the unique optimal transport plan between $\mu_X$ and $\rho$ (for any cost of the form $c(x,y) = h(x-y)$ with $h : \mathbb{R} \mapsto [0, \infty)$ strictly convex) is given by*

$$\gamma = \frac{1}{N} \sum_{i=1}^N \delta_{x_i} \otimes \rho_i \tag{111}$$

*Proof.* First, we fix a point cloud $X = (x_1, \ldots, x_n) \in \mathbb{R}^N$ such that $x_1 < \ldots < x_N$, and we let $\gamma_X$ be the unique optimal transport plan between $\mu_X$ and $\rho$. By the disintegration theorem Theorem A.2, there exists an unique family $\rho_{X,1}, \ldots, \rho_{X,N} \in \mathcal{P}_2(\mathbb{R})$ such that $\gamma_X = \frac{1}{N} \sum_{i=1}^N \delta_{x_i} \otimes \rho_{X,i}$. In particular, we have $\rho = \pi_{2\#}\gamma_X = \frac{1}{N} \sum_{i=1}^N \rho_{X,i}$. Furthermore, if $i < j$, $y_i \in \mathrm{spt}(\rho_{X,i})$ and $y_j \in \mathrm{spt}(\rho_{X,j})$, then we have $(x_i, y_i), (x_j, y_j) \in \mathrm{spt}(\gamma_X)$. and Theorem B.6 directly implies that $y_i \leq y_j$. Now, all that is left to do is to show that the family $(\rho_{X,i})_i$ does not actually depend on $X$. This is the case since, if $X' = (x_1', \ldots, x_N') \in \mathbb{R}^N$ is another point cloud with $x_1' < \ldots < x_N'$, we see that the transport plan $\gamma = \frac{1}{N} \sum_{i=1}^N \delta_{x_i'} \otimes \rho_{X,i}$ also satisfies (110), so that $\gamma = \gamma_{X'}$ by Theorem B.6. But then we must have $\rho_{X,i} = \rho_{X',i}$ for every $i$ by unicity of the family $(\rho_{X',i})_i$. This finishes the proof. $\square$

*Remark* B.8. In the case where $\rho \in \mathcal{P}_2(\mathbb{R})$ has no atoms, it is not difficult to see that $\rho_i = N\rho_{|V_i}$ where $V_i$ is the $i$-th Power cell $V_i := F_\rho^{-1}([(i-1)/N, i/N])$.

Then, fixing $\rho \in \mathcal{P}_2(\mathbb{R}^d)$, for every $\theta \in \mathbb{S}^{d-1}$, we denote $\rho_{\theta,1}, \ldots, \rho_{\theta,N} \in \mathcal{P}_2(\mathbb{R})$ the measures given by applying Lemma B.7 to $\rho_\theta$, and we define $b_{\theta,i} := \int x d\rho_{\theta,i}(x)$ the corresponding barycenters (by Remark B.8, if $\rho$ is without atoms, this is indeed the barycenter of $\rho_\theta$ on the Power cell $V_{\theta,i}$, and there is no conflict of notation). We then have the following extensions of Proposition 3.1, Proposition 3.2 and Proposition 3.3 :

**Proposition B.9.** *If $p \geq 2$, then $F : X \in (\mathbb{R}^d)^N \mapsto \frac{1}{p} \mathrm{SW}_p^p(\mu_X, \rho)$ is differentiable at any point cloud $X = (X_1, \ldots, X_N) \in (\mathbb{R}^d)^N$ which does not belong to the generalized diagonal $\Delta_N$. The gradient of $F$ is continuous on $(\mathbb{R}^d)^N \setminus \Delta_N$ and the expression of its component with respect to the $i$-th vector $X_i$ is then*

$$\nabla_{X_i} F(X) = \frac{1}{N} \int_{\mathbb{S}^{d-1}} \int \mathrm{sgn}(\langle X_i|\theta\rangle - x)|\langle X_i|\theta\rangle - x|^{p-1}\theta d\rho_{\theta,\sigma_{X,\theta}^{-1}(i)}(x)d\theta, \tag{112}$$

*In the particular case where $p = 2$, this expression can be further simplified :*

$$\nabla_{X_i} F(X) = \frac{1}{N}\left(\frac{1}{d}X_i - \int_{\mathbb{S}^{d-1}} b_{\theta,\sigma_{X,\theta}^{-1}(i)}\theta d\theta\right) \tag{113}$$

*Still in the case $p = 2$, for every $X \in (\mathbb{R}^d)^N \setminus \Delta_N$ and every $\lambda > 0$, denoting $Y := X - \lambda \nabla F(X)$, we have*

$$F(Y) - F(X) \leq -\lambda\left(1 - \frac{\lambda}{2Nd}\right)\|\nabla F(X)\|^2 \tag{114}$$

*and, provided $\lambda \in (0, Nd)$, we have $Y \notin \Delta_N$.*

*Proof.* The optimal transport plan between $P_{\theta\#}\mu_X$ and $\rho_\theta$ is given by

$$\gamma_\theta = \frac{1}{N}\sum_{i=1}^N \delta_{\langle X_i|\theta\rangle} \otimes \rho_{\theta,\sigma_{X,\theta}^{-1}(i)} \tag{115}$$

We then prove that $F$ is differentiable at $X$ with the given expression, and that $\nabla F$ is continuous, the same way as in the proof of Proposition 3.1 (in Appendix B.1), where we replace every integration on a Power cell (of the form $\int_{V_{\theta,i}} \ldots d\rho_\theta(x)$) by an integration on $\rho_{\theta,i}$ (of the form $\frac{1}{N}\int \ldots d\rho_{\theta,i}(x)$).
Similarly, we prove the upper bound on $F(Y) - F(X)$ the same way as in the proof of Proposition 3.2 (in Appendix B.2), by noting that a (not necessarily optimal) transport plan between $P_{\theta\#}\mu_Y$ and $\rho_\theta$ is given by

$$\tilde{\gamma}_\theta = \frac{1}{N}\sum_{i=1}^N \delta_{\langle Y_i|\theta\rangle} \otimes \rho_{\theta,\sigma_{X,\theta}^{-1}(i)} \tag{116}$$

Finally, if $\lambda \in (0, Nd)$, we prove that $Y \notin \Delta_N$ the same way as in the proof of Proposition 3.3 (in Appendix B.3), as we will still have $N\langle \nabla_{X_i} F(X) - \nabla_{X_j} F(X)|X_i - X_j\rangle \leq \frac{1}{d}\|X_i - X_j\|^2$ by the same reasoning. $\qquad\square$

Furthermore, for a fixed $N > 0$, provided $\rho \in \mathcal{P}_2(\mathbb{R}^d)$ satisfies a technical assumptions on its barycenters $b_{\theta,i}$, then we have the following extension of Proposition 3.3 :

**Proposition B.10.** *Assume that there exists $m > 0$ and $\Theta \subseteq \mathbb{S}^{d-1}$ with $\mathcal{U}_{\mathbb{S}^{d-1}}(\Theta) > 0$, such that for every $\theta \in \Theta$ and $i \in \{1, \ldots, N-1\}$, $b_{\theta,i+1} - b_{\theta,i} \geq m$. Then there exists some constant $C = C(\Theta) > 0$ such that for every $X \in (\mathbb{R}^d)^N$ and $\lambda > 0$, setting $Y := X - \lambda \nabla F(X)$, for every $i \neq j$, if $\|X_i - X_j\| < dCm$, then $\|Y_i - Y_j\| > \|X_i - X_j\|$. In particular, if $X$ is a critical point of $F$, then*

$$\min_{i \neq j}\|X_i - X_j\| \geq dCm \tag{117}$$

*Proof.* By the same argument as in the proof of Proposition 3.3 in Appendix B.3, we have

$$\|Y_i - Y_j\|^2 \geq \|X_i - X_j\|^2 - 2\lambda\langle \nabla_{X_i} F(X) - \nabla_{X_j} F(X)|X_i - X_j\rangle \tag{118}$$

with

$$N\langle \nabla_{X_i} F(X) - \nabla_{X_j} F(X)|X_i - X_j\rangle = \frac{1}{d}\|X_i - X_j\|^2 - 2\int_{\{\langle\theta|\theta_0\rangle>0\}} (b_{\theta,\sigma_{X,\theta}^{-1}(i)} - b_{\theta,\sigma_{X,\theta}^{-1}(j)})\langle\theta|X_i - X_j\rangle d\theta \tag{119}$$

and $\theta_0 := \frac{X_i - X_j}{\|X_i - X_j\|}$. Indeed, we can again check that we have $\sigma_{X,-\theta}^{-1}(k) = N + 1 - \sigma_{X,\theta}^{-1}(k)$ and $b_{-\theta,k} = b_{\theta,N+1-k}$ for every $k = 1, \ldots, N$. In particular, we may assume that $\Theta = -\Theta$. Now, if $\theta \in \mathbb{S}^{d-1}$ is such that $\langle \theta | \theta_0 \rangle \geq 0$, we again have $(b_{\theta,\sigma_{X,\theta}^{-1}(i)} - b_{\theta,\sigma_{X,\theta}^{-1}(j)}) \langle \theta | X_i - X_j \rangle > 0$, and if furthermore $\theta \in \Theta$, we have $(b_{\theta,\sigma_{X,\theta}^{-1}(i)} - b_{\theta,\sigma_{X,\theta}^{-1}(j)}) \langle \theta | X_i - X_j \rangle \geq m \langle \theta | X_i - X_j \rangle$, so that

$$\int_{\{\langle\theta|\theta_0\rangle>0\}} (b_{\theta,\sigma_{X,\theta}^{-1}(i)} - b_{\theta,\sigma_{X,\theta}^{-1}(j)}) \langle \theta|X_i - X_j \rangle d\theta \geq \int_{\Theta\cap\{\langle\theta|\theta_0\rangle>0\}} m\langle\theta|X_i - X_j\rangle d\theta \tag{120}$$

$$\geq m\|X_i - X_j\| \int_{\Theta\cap\{\langle\theta|\theta_0\rangle>0\}} \langle\theta|\theta_0\rangle d\theta \tag{121}$$

Now, let $\alpha \in (0, 1]$ be the unique value such that

$$\mathcal{U}_{\mathbb{S}^{d-1}}(\{\alpha > \langle\theta|\theta_0\rangle > 0\}) = \mathcal{U}_{\mathbb{S}^{d-1}}(\Theta \cap \{\langle\theta|\theta_0\rangle > 0\}) = \frac{1}{2}\mathcal{U}_{\mathbb{S}^{d-1}}(\Theta) \tag{122}$$

(by symmetry, $\alpha$ does not depend on $\theta_0$, and only depends on $\mathcal{U}_{\mathbb{S}^{d-1}}(\Theta)$). Then, we have

$$\int_{\Theta\cap\{\langle\theta|\theta_0\rangle>0\}} \langle\theta|\theta_0\rangle d\theta = \int_{\Theta\cap\{\langle\theta|\theta_0\rangle\geq\alpha\}} \langle\theta|\theta_0\rangle d\theta + \int_{\Theta\cap\{\alpha>\langle\theta|\theta_0\rangle>0\}} \langle\theta|\theta_0\rangle d\theta \tag{123}$$

$$\geq \alpha\mathcal{U}_{\mathbb{S}^{d-1}}(\Theta \cap \{\langle\theta|\theta_0\rangle \geq \alpha\}) + \int_{\Theta\cap\{\alpha>\langle\theta|\theta_0\rangle>0\}} \langle\theta|\theta_0\rangle d\theta \tag{124}$$

$$\geq \alpha\mathcal{U}_{\mathbb{S}^{d-1}}(\Theta^c \cap \{\alpha > \langle\theta|\theta_0\rangle > 0\}) + \int_{\Theta\cap\{\alpha>\langle\theta|\theta_0\rangle>0\}} \langle\theta|\theta_0\rangle d\theta \tag{125}$$

$$\geq \int_{\{\alpha>\langle\theta|\theta_0\rangle>0\}} \langle\theta|\theta_0\rangle d\theta \tag{126}$$

where the third line is obtained by noticing that (122) implies that $\mathcal{U}_{\mathbb{S}^{d-1}}(\Theta \cap \{\langle\theta|\theta_0\rangle \geq \alpha\}) = \mathcal{U}_{\mathbb{S}^{d-1}}(\Theta^c \cap \{\alpha > \langle\theta|\theta_0\rangle > 0\})$. Thus, denoting $C = C(\Theta) := 2 \int_{\{\alpha>\langle\theta|\theta_0\rangle>0\}} \langle\theta|\theta_0\rangle d\theta$, we have, combining (118), (119), (121) and (126),

$$\|Y_i - Y_j\|^2 \geq \|X_i - X_j\|^2 - \frac{2\lambda}{N}\left(\frac{1}{d}\|X_i - X_j\|^2 - Cm\|X_i - X_j\|\right) \tag{127}$$

from which we deduce $\|Y_i - Y_j\| > \|X_i - X_j\|$ if $\|X_i - X_j\| < Cmd$. In particular, if $X$ is a critical point of $F$, then $\nabla F(X) = 0$ and $Y = X$, so we must have $\|X_i - X_j\| \geq Cmd$. $\qquad\square$

*Remark* B.11. In particular, if there exists $\beta > 0$ and $\Theta \subseteq \mathbb{S}^{d-1}$ with $\mathcal{U}_{\mathbb{S}^{d-1}}(\Theta) > 0$ such that for every $\theta \in \Theta$, $\rho_\theta$ is absolutely continuous with a density bounded from above by $\beta$, then, by the same reasoning as in the proof of Proposition 3.3 in Appendix B.3, for every $\theta \in \Theta$ and $i = 1, \ldots, N - 1$, we have $b_{\theta,i+1} - b_{\theta,i} \geq \frac{1}{N\beta}$. Thus $\rho$ satisfies the assumption of Proposition B.10 for every $N > 0$ with $\Theta$ and $m := \frac{1}{\beta N}$. Therefore, we only need to have an upper bound on the densities of the $\rho_\theta$ for a non negligible set of directions $\theta$ (instead of all of them) for the gradient descent to be well-behaved (i.e. to guarantee that the iterates do not get too close to the generalized diagonal and are repelled by it).

## B.5. Proof of Proposition 4.4

First, it will be helpful to introduce the following family of transport plans between the projected measures : for a given $\theta \in \mathbb{S}^{d-1}$, we use Theorem A.2 to disintegrate $\mu$ and $\rho$ with respect to $P_\theta$ to get families of probabilities $(\mu_{\theta,u})_{u\in\mathbb{R}}$ and $(\rho_{\theta,v})_{v\in\mathbb{R}}$ such that $\mathrm{spt}(\mu_{\theta,u}) \subseteq P_\theta^{-1}(u)$, $\mathrm{spt}(\rho_{\theta,v}) \subseteq P_\theta^{-1}(s)$ and for every test function $\varphi \in C^0(\Omega)$, $\int \varphi(x)d\mu(x) = \int\int \varphi(x)d\mu_{\theta,u}(x)d\mu_\theta(u)$ and $\int \varphi(y)d\rho(y) = \int\int \varphi(y)d\rho_{\theta,v}(y)d\rho_\theta(v)$. We then define $\hat{\gamma}_\theta$ as the probability measure whose integral over a test function $\varphi(x,y) \in C^0(\Omega \times \Omega)$ is

$$\int \varphi(x,y)d\hat{\gamma}_\theta(x,y) = \int\int\int \varphi(x,y)d\mu_{\theta,u}(x)d\rho_{\theta,v}(y)d\gamma_\theta(u,v). \tag{128}$$

We can see then that $\hat{\gamma}_\theta$ is a transport plan (not necessarily optimal) between $\mu$ and $\nu$ and that $(P_\theta, P_\theta)_{\#}\hat{\gamma}_\theta = \gamma_\theta$ (in other words, $\hat{\gamma}_\theta$ is optimal for the cost function $(x,y) \mapsto \langle y - x|\theta\rangle^2$). We also disintegrate $\gamma_\theta$ with respect to the first

variable, giving a family of probabilities $(\gamma_{\theta,u})_{u \in \mathbb{R}}$ such that for every test function $\varphi \in C^0(\mathbb{R} \times \mathbb{R})$, $\int \varphi(u,v)d\gamma_\theta(u,v) = \int \int \varphi(u,v)d\gamma_{\theta,u}(v)d\mu_\theta(u)$. Notice that these give an alternative definition of $\bar{\gamma}_\theta$: indeed $\bar{\gamma}_\theta(u) = \int vd\gamma_{\theta,u}(v)$.

We can now proceed to the proof of Proposition 4.4.

*Proof (Proposition 4.4).* First, if $\xi \in L^2(\mu_X, \mathbb{R}^d)$, then, defining $H \in (\mathbb{R}^d)^N$ by $H_i := \xi(X_i)$ for every $i = 1, \ldots, N$, we have for every $t > 0$ (small enough so that $X + tH \notin \Delta_N$),

$$F(X + tH) = \frac{1}{2}\operatorname{SW}_2^2(\mu_{X+tH}, \rho) = \frac{1}{2}\operatorname{SW}_2^2((\operatorname{Id} + t\xi)_{\#}\mu_X, \rho) \tag{129}$$

from which we deduce, by taking the right derivative at $t = 0$,

$$\langle \nabla F(X)|H \rangle = \frac{d}{dt}\operatorname{SW}_2^2((\operatorname{Id} + t\xi)_{\#}\mu, \rho)\Big|_{t=0^+} \tag{130}$$

In particular, we immediately see from Definition 4.1 that $\nabla F(X) = 0$ if and only if $\mu_X$ is a Lagrangian critical point.

Second, the condition $v_{\mu_X} = 0$ $\mu_X$-a.e. from Definition 4.2 writes as

$$\frac{1}{d}X_i - \int_{\mathbb{S}^{d-1}} \bar{\gamma}_\theta(\langle X_i|\theta \rangle)\theta d\theta = 0, \quad i \in \{1, \ldots, N\} \tag{131}$$

Fix $\theta \in \mathbb{S}^{d-1}$ such that the $\langle X_1|\theta \rangle, \ldots, \langle X_N|\theta \rangle$ are distinct. Using the notations from Appendix B.4, we know that $\gamma_\theta = \frac{1}{N}\sum_{i=1}^{N} \delta_{\langle X_i|\theta \rangle} \otimes \rho_{\theta, \sigma_{X,\theta}^{-1}(i)}$ where $\rho_{\theta,1}, \ldots, \rho_{\theta,N}$ is the family given by applying Lemma B.7 to $\rho_\theta$. In particular, we deduce that for every $i$, $\gamma_{\theta,\langle X_i|\theta \rangle} = \rho_{\theta,\sigma_{X,\theta}^{-1}(i)}$, and thus

$$\bar{\gamma}_\theta(\langle X_i|\theta \rangle) = \int_{\mathbb{R}} vd\gamma_{\theta,\langle \theta|X_i \rangle}(v) = \int vd\rho_{\theta,\sigma_{X,\theta}^{-1}(i)}(v) = b_{\theta,\sigma_{X,\theta}^{-1}(i)} \tag{132}$$

and, using (113), (131) rewrites as

$$N\nabla_{X_i}F(X) = 0, \quad i \in \{1, \ldots, N\} \tag{133}$$

Thus, $\nabla F(X) = 0$ iff $\mu_X$ is a barycentric Lagrangian critical point. $\square$

*Remark* B.12. Notice that this proof works in fact for general $\rho \in \mathcal{P}_2(\mathbb{R}^d)$ (with the expression of gradient $\nabla F$ given in Proposition B.9).

## B.6. Proof of Proposition 4.7

First, we prove Proposition 4.7(a). Let $\xi_0, \xi_1 \in L^2(\mu, \mathbb{R}^d)$, we denote $S^t = \operatorname{Id} + (1-t)\xi_0 + t\xi_1$ and $\mu^t = S_{\#}^t\mu$. For any fixed $t \in [0,1]$, $\gamma := ((P_\theta, P_\theta) \circ (S_0, S_1))_{\#}\mu$ is a transport plan between $\mu_\theta^0$ and $\mu_\theta^1$ such that

$$\mu_\theta^t = ((1-t)\pi_1 + t\pi_2)_{\#}\gamma. \tag{134}$$

where $\pi_i$ is the projection on the $i$-th coordinate. Furthermore, by Proposition 7.3.1 of (Ambrosio et al., 2005), there exists a plan $\eta \in \mathcal{P}(\mathbb{R} \times \mathbb{R} \times \mathbb{R})$ such that $(\pi_1, \pi_2)_{\#}\eta = \gamma$ and $((1-t)\pi_1 + t\pi_2, \pi_3)_{\#}\eta$ is an optimal transport plan between $\mu_\theta^t$ and $\rho_\theta$. Then, according to Theorem 7.3.2 of (Ambrosio et al., 2005), asserting the semi-concavity of the squared Wasserstein distance, we have

$$\operatorname{W}_2^2(\mu_\theta^t, \rho_\theta) \geq (1-t)\operatorname{W}_2^2(\mu_\theta^0, \rho_\theta) + t\operatorname{W}_2^2(\mu_\theta^0, \rho_\theta) - t(1-t)\operatorname{W}_\eta^2(\mu_\theta^0, \mu_\theta^1), \tag{135}$$

where $W_\eta$ is defined in (7.3.2) of (Ambrosio et al., 2005) by

$$W_\eta^2(((1-t)\pi_i + t\pi_j)_{\#}\eta, \pi_{k\#}\eta) := \int_{\mathbb{R} \times \mathbb{R} \times \mathbb{R}} |(1-t)x_i + tx_j - x_k|^2 d\eta(x_i, x_j, x_k) \tag{136}$$

for every $i, j, k \in \{1, 2, 3\}$ and $t \in [0, 1]$. In this case, we have

$$W_\eta^2(\mu_\theta^0, \mu_\theta^1) = \int_{\mathbb{R}^3} (x_1 - x_2)^2 d\eta(x_1, x_2, x_3) = \int_{\mathbb{R}^2} (x - y)^2 d\gamma(x, y) \tag{137}$$

$$= \int_{\mathbb{R}^2} \langle x - y | \theta \rangle^2 d(S_0, S_1)_{\#}\mu(x, y) = \int \langle \xi_0(x) - \xi_1(x) | \theta \rangle^2 d\mu(x) \tag{138}$$

(we take $i = 0, j = 2, k = 1$ and $t = 0$ in (136)). Integrating the inequality (135) over $\theta \in \mathbb{S}^{d-1}$, we get

$$\mathrm{SW}_2^2(\mu^t, \rho) \geq (1 - t)\,\mathrm{SW}_2^2(\mu^0, \rho) + t\,\mathrm{SW}_2^2(\mu^1, \rho) - t(1 - t) \int \int_{\mathbb{S}^{d-1}} \langle \xi_1(x) - \xi_0(x) | \theta \rangle^2 d\theta d\mu(x) \tag{139}$$

$$\geq (1 - t)\,\mathrm{SW}_2^2(\mu^0, \rho) + t\,\mathrm{SW}_2^2(\mu^1, \rho) - \frac{1}{d}t(1 - t)\|\xi_1 - \xi_0\|_{L^2(\mu)}^2 \tag{140}$$

This rewrites as

$$F_\mu((1 - t)\xi_0 + t\xi_1) \leq (1 - t)F_\mu(\xi_0) + tF_\mu(\xi_1) \tag{141}$$

which proves the convexity of $F_\mu$.

Now, we prove Proposition 4.7(b). First, we show that $v_\mu \in L^2(\mu, \mathbb{R}^d)$. This is the case because $\mathrm{Id} \in L^2(\mu, \mathbb{R}^d)$ as $\mu \in \mathcal{P}_2(\mathbb{R}^d)$, and

$$\int_{\mathbb{R}^d} \left| \int_{\mathbb{S}^{d-1}} \bar{\gamma}_\theta(\langle x | \theta \rangle)\theta d\theta \right|^2 d\theta d\mu(x) \leq \int_{\mathbb{R}^d} \int_{\mathbb{S}^{d-1}} \bar{\gamma}_\theta^2(\langle x | \theta \rangle)d\theta d\mu(x) \tag{142}$$

$$\leq \int_{\mathbb{R}^d} \int_{\mathbb{S}^{d-1}} \int_{\mathbb{R}} v^2 d\gamma_{\theta, \langle x | \theta \rangle}(v)d\theta d\mu(x) \tag{143}$$

$$\leq \int_{\mathbb{S}^{d-1}} \int_{\mathbb{R}^d} \int_{\mathbb{R}} v^2 d\gamma_{\theta, \langle x | \theta \rangle}(v)d\mu(x)d\theta \tag{144}$$

$$\leq \int_{\mathbb{S}^{d-1}} \int_{\mathbb{R}} \int_{\mathbb{R}} v^2 d\gamma_{\theta, u}(v)d\mu_\theta(u)d\theta \tag{145}$$

$$\leq \int_{\mathbb{S}^{d-1}} \int_{\mathbb{R}^2} v^2 d\gamma_\theta(u, v)d\theta \tag{146}$$

$$\leq \int_{\mathbb{S}^{d-1}} \int_{\mathbb{R}} v^2 d\rho_\theta(v)d\theta \tag{147}$$

$$\leq \int_{\mathbb{S}^{d-1}} \int_{\mathbb{R}^d} \langle y | \theta \rangle^2 d\rho(y)d\theta = \frac{1}{d} \int_{\mathbb{R}^d} \|y\|^2 d\rho(y) < \infty \tag{148}$$

where we used Jensen's inequality in the first lines, and $\rho \in \mathcal{P}_2(\mathbb{R}^d)$. This proves that $v_\mu$ is in $L^2(\mu, \mathbb{R}^d)$.

Fix now $\xi \in L^2(\mu, \mathbb{R}^d)$. Denote $S_\xi = \mathrm{Id} + \xi$ and $\mu^\xi = S_{\xi\#}\mu$, then for every $\theta \in \mathbb{S}^{d-1}$, the plan $\hat{\gamma}_\theta^\xi := (S_\xi, \mathrm{Id})_{\#}\hat{\gamma}_\theta$ is a transport plan between $\mu^\xi$ and $\rho$, such that $(P_\theta, P_\theta)_{\#}\hat{\gamma}_\theta^\xi \in \Pi(\mu_\theta^\xi, \rho_\theta)$ is not necessarily optimal. Then, we have

$$W_2^2(\mu_\theta^\xi, \rho_\theta) \leq \int_{(\mathbb{R}^d)^2} \langle x - y | \theta \rangle^2 d\hat{\gamma}_\theta^\xi(x, y)$$

$$\leq \int_{(\mathbb{R}^d)^2} \langle S_\xi(x) - y | \theta \rangle^2 d\hat{\gamma}_\theta(x, y)$$

$$\leq \int_{(\mathbb{R}^d)^2} \langle x + \xi(x) - y | \theta \rangle^2 d\hat{\gamma}_\theta(x, y)$$

$$\leq \int_{(\mathbb{R}^d)^2} \langle x - y | \theta \rangle^2 d\hat{\gamma}_\theta(x, y) + 2 \int_{(\mathbb{R}^d)^2} \langle x - y | \theta \rangle \langle \theta | \xi(x) \rangle d\hat{\gamma}_\theta(x, y) + \int_{(\mathbb{R}^d)^2} \langle \xi(x) | \theta \rangle^2 d\hat{\gamma}_\theta(x, y)$$

$$\leq W_2^2(\mu_\theta, \rho_\theta) + 2 \int_{(\mathbb{R}^d)^2} \langle x - y | \theta \rangle \langle \theta | \xi(x) \rangle d\hat{\gamma}_\theta(x, y) + \int_{\mathbb{R}^d} \langle \xi(x) | \theta \rangle^2 d\mu(x) \tag{149}$$

The second term in the right hand side of the last inequality is

$$\int_{(\mathbb{R}^d)^2} \langle x - y | \theta \rangle \langle \theta | \xi(x) \rangle d\hat{\gamma}_\theta(x, y) = \int (u - v) \int \langle \theta | \xi(x) \rangle d\mu_{\theta, u}(x) d\gamma_\theta(u, v)$$

$$= \int \int \int (u - v) \langle \theta | \xi(x) \rangle d\mu_{\theta, u}(x) d\gamma_{\theta, u}(v) d\mu_\theta(u)$$

$$= \int \int \int (u - v) \langle \theta | \xi(x) \rangle d\gamma_{\theta, u}(v) d\mu_{\theta, u}(x) d\mu_\theta(u)$$

$$= \int_{\mathbb{R}^d} \langle \theta | \xi(x) \rangle \int (\langle x | \theta \rangle - v) d\gamma_{\theta, \langle x | \theta \rangle}(v) d\mu(x)$$

$$= \int_{\mathbb{R}^d} \langle \theta | \xi(x) \rangle (\langle x | \theta \rangle - \bar{\gamma}_\theta(\langle x | \theta \rangle)) d\mu(x) \tag{150}$$

Therefore, integrating (149) using (150), we get

$$\mathrm{SW}_2^2(\mu^\xi, \rho) \leq \mathrm{SW}_2^2(\mu, \rho) + 2 \int_{\mathbb{S}^{d-1}} \int_{\mathbb{R}^d} (\langle x | \theta \rangle - \bar{\gamma}_\theta(\langle x | \theta \rangle)) \langle \theta | \xi(x) \rangle d\mu(x) d\theta + \frac{1}{d} \|\xi\|_{L^2(\mu)}^2 \tag{151}$$

but since

$$\int_{\mathbb{S}^{d-1}} \int_{\mathbb{R}^d} (\langle x | \theta \rangle - \bar{\gamma}_\theta(\langle x | \theta \rangle)) \langle \theta | \xi(x) \rangle d\mu(x) d\theta = \int_{\mathbb{R}^d} \langle \xi(x) | \int_{\mathbb{S}^{d-1}} (\langle x | \theta \rangle - \bar{\gamma}_\theta(\langle x | \theta \rangle)) \theta d\theta \rangle d\mu(x) \tag{152}$$

$$= \int_{\mathbb{R}^d} \langle \xi(x) | \frac{1}{d} x - \int_{\mathbb{S}^{d-1}} \bar{\gamma}_\theta(\langle x | \theta \rangle) \theta d\theta \rangle d\mu(x) \tag{153}$$

$$= \langle \xi | v_\mu \rangle_{L^2(\mu)} \tag{154}$$

equation (151) rewrites as

$$\mathrm{SW}_2^2(\mu^\xi, \rho) \leq \mathrm{SW}_2^2(\mu, \rho) + 2 \langle v_\mu | \xi \rangle_{L^2(\mu)} + \frac{1}{d} \|\xi\|_{L^2(\mu)}^2 \tag{155}$$

that is

$$F_\mu(0) - 2 \langle v_\mu | \xi \rangle_{L^2(\mu)} \leq F_\mu(\xi) \tag{156}$$

and this finishes proving Proposition 4.7(b).

Finally, we prove Proposition 4.7(c). Assume that $\mu, \rho$ are supported in some compact set $\Omega \subseteq \mathbb{R}^d$ and are without atoms. Let $\xi \in L^2(\mu, \mathbb{R})$ be fixed and define $\varphi(t) := \mathrm{SW}_2^2(\mu^t, \rho)$ where $\mu^t = (\mathrm{Id} + t\xi)_{\#}\mu$. Equation (155) applied to the vector field $t\xi$ gives

$$\varphi(t) \leq \varphi(0) + 2t \langle v_\mu | \xi \rangle_{L^2(\mu)} + \frac{1}{d} t^2 \|\xi\|_{L^2(\mu)}^2 \tag{157}$$

Therefore, we immediately have the inequalities

$$\limsup_{t \mapsto 0^+} \frac{1}{t}(\varphi(t) - \varphi(0)) \leq 2 \langle \xi | v_\mu \rangle_{L^2(\mu)} \tag{158}$$

$$\liminf_{t \mapsto 0^-} \frac{1}{t}(\varphi(t) - \varphi(0)) \geq 2 \langle \xi | v_\mu \rangle_{L^2(\mu)} \tag{159}$$

Let's derive the other inequalities : let $(\varphi_\theta, \psi_\theta)$ be a pair of c-concave Kantorovich potentials for $(\mu_\theta, \rho_\theta)$ (for the cost $c(u, v) = \frac{1}{2}(u - v)^2$). For every $t > 0$, we then have

$$\frac{1}{t}(\mathrm{W}_2^2(\mu_\theta^t, \rho_\theta) - \mathrm{W}_2^2(\mu_\theta, \rho_\theta)) \geq \frac{2}{t} \int_{\mathbb{R}^2} \varphi_\theta(u)(d\mu_\theta^t(u) - d\mu_\theta(u)) \tag{160}$$

$$\geq \frac{2}{t} \left( \int_{\mathbb{R}^d} \varphi_\theta(\langle x + t\xi(x) | \theta \rangle) - \varphi_\theta(\langle x | \theta \rangle) d\mu(x) \right) \tag{161}$$

(the factor 2 comes from the factor $\frac{1}{2}$ in the cost $c$). By c-concavity, $\varphi_\theta$ is Lipschitz on $P_\theta(\Omega)$ (it has the same modulus of continuity as $c$ - note that we use here the fact that $\mu$ and $\rho$ have compact support). Thus, $t \mapsto \frac{1}{t}(\varphi_\theta(\langle x + t\xi(x)|\theta\rangle) - \varphi_\theta(\langle x|\theta\rangle))$ is bounded from below by $-L|\langle \xi(x)|\theta\rangle|$, which is integrable as $\xi \in L^2(\mu, \mathbb{R}^d)$, where $L$ is the Lipschitz constant of $\varphi_\theta$, which depends only on $\mathrm{diam}(\Omega)$. Since for the cost $c$, c-concavity means that $\frac{1}{2}|\cdot|^2 - \varphi_\theta$ is convex and lsc (see (Santambrogio, 2015, Proposition 1.21)), $\varphi_\theta$ has at every point right and left derivatives $\varphi_\theta^+$ and $\varphi_\theta^-$, therefore, applying Fatou's lemma and integrating on $\mathbb{S}^{d-1}$,

$$\liminf_{t \mapsto 0^+} \frac{1}{t}(W_2^2(\mu_\theta^t, \rho_\theta) - W_2^2(\mu_\theta, \rho_\theta)) \geq 2 \int_{\mathbb{S}^{d-1}} \int_{\mathbb{R}^d} \varphi_\theta^{\mathrm{sgn}(\langle \xi(x)|\theta\rangle)}(\langle x|\theta\rangle)\langle \xi(x)|\theta\rangle d\mu(x)d\theta \tag{162}$$

However, since $\mu$ is without atoms, by Proposition A.1, for almost every $\theta \in \mathbb{S}^{d-1}$, $\mu_\theta$ is without atoms, and for $\mu_\theta$-almost every $u$, $\varphi_\theta$ is differentiable at $u$ with $\varphi_\theta'(u) = \varphi_\theta^+(u) = \varphi_\theta^-(u)$[6]. Furthermore, we have $\varphi_\theta'(u) = (u - T_\theta(u))$, where $T_\theta$ is the optimal transport map from $\mu_\theta$ to $\rho_\theta$, and $\bar{\gamma}_\theta = T_\theta$ (as $\gamma_\theta = (\mathrm{Id}, T_\theta)_{\#}\mu_\theta)$, and therefore

$$\int_{\mathbb{R}^d} \varphi_\theta'(\langle x|\theta\rangle)\langle \xi(x)|\theta\rangle d\mu(x)) = \int_{\mathbb{R}^d} (\langle x|\theta\rangle - T_\theta(\langle x|\theta\rangle))\langle \xi(x)|\theta\rangle d\mu(x) \tag{163}$$

$$= \int_{\mathbb{R}^d} (\langle x|\theta\rangle - \bar{\gamma}_\theta(\langle x|\theta\rangle))\langle \xi(x)|\theta\rangle d\mu(x) \tag{164}$$

so

$$\liminf_{t \mapsto 0^+} \frac{1}{t}(W_2^2(\mu_\theta^t, \rho_\theta) - W_2^2(\mu_\theta, \rho_\theta)) \geq 2 \int_{\mathbb{S}^{d-1}} \int_{\mathbb{R}^d} (\langle x|\theta\rangle - \bar{\gamma}_\theta(\langle x|\theta\rangle))\langle \xi(x)|\theta\rangle d\mu(x)d\theta \tag{165}$$

Integrating this latter inequality, we obtain

$$\liminf_{t \mapsto 0^+} \frac{1}{t}(\varphi(t) - \varphi(0)) \geq 2\langle \xi|v_\mu\rangle_{L^2(\mu)} \tag{166}$$

Using a similar argument, we show that

$$\limsup_{t \to 0^-} \frac{1}{t}(\varphi(t) - \varphi(0)) \leq 2\langle \xi|v_\mu\rangle_{L^2(\mu)} \tag{167}$$

This proves that $\varphi$ is differentiable at $t = 0$, with

$$\varphi'(0) = 2\langle \mu|\xi\rangle_{L^2(\mu)} \tag{168}$$

This finishes the proof. $\square$

## B.7. Proof of Corollary 4.8

First, if $\mu$ is a Lagrangian critical point for $\mathrm{SW}_2^2(\cdot, \rho)$, then for every $\xi \in L^2(\mu, \mathbb{R}^d)$, it satisfies (14) :

$$\left. \frac{d}{dt} \mathrm{SW}_2^2((\mathrm{Id} + t\xi)_{\#}\mu, \rho) \right|_{t=0^+} = 0 \tag{169}$$

But applying Proposition 4.7(b) to the vector field $t\xi$, we have for every $t > 0$

$$\mathrm{SW}_2^2((\mathrm{Id} + t\xi)_{\#}\mu, \rho) \leq \mathrm{SW}_2^2(\mu, \rho) + 2t\langle v_\mu|\xi\rangle_{L^2(\mu)} + \frac{1}{d}t^2\|\xi\|_{L^2(\mu)}^2 \tag{170}$$

Combined with the previous equation, this yields

$$0 = \left. \frac{d}{dt} \mathrm{SW}_2^2((\mathrm{Id} + t\xi)_{\#}\mu, \rho) \right|_{t=0^+} \leq 2\langle v_\mu|\xi\rangle_{L^2(\mu)} \tag{171}$$

Therefore, we have $\langle v_\mu|\xi\rangle_{L^2(\mu)} \geq 0$ for every $\xi \in L^2(\mu, \mathbb{R}^d)$, and this implies $v_\mu = 0$ in $L^2(\mu, \mathbb{R}^d)$. Thus, $\mu$ is a barycentric Lagrangian critical point.

---

[6]Since $\frac{x^2}{2} - \varphi_\theta$ is convex, it is differentiable almost everywhere, with a nondecreasing differential. Furthermore its set of nondifferentiability is at most countable, so it has zero $\mu_\theta$-measure as $\mu_\theta$ is without atoms.

Now, assume that $\mu, \rho$ are compactly supported and without atoms. Then, by proposition 4.7(c), for every $\xi \in L^2(\mu, \mathbb{R}^d)$, we have

$$\frac{d}{dt} \mathrm{SW}_2^2((\mathrm{Id} + t\xi)_{\#}\mu, \rho)\Big|_{t=0^+} = 2\langle v_\mu | \xi \rangle_{L^2(\mu)} \tag{172}$$

Therefore $\mu$ satisfies Definition 4.1 if and only if $\langle v_\mu | \xi \rangle_{L^2(\mu)} = 0$ for every $\xi \in L^2(\mu, \mathbb{R}^d)$, which is equivalent to $v_\mu = 0$ $\mu$-a.e. Thus $\mu$ is Lagrangian critical if and only if it is barycentric Lagrangian critical. $\qquad\square$

### B.8. Proof of Theorem 4.5

First, up to extending $\Omega$, we may assume that the $\mu_n, \mu$ are supported in $\Omega$. Indeed, if $R > 0$ is such that $\Omega \subseteq B(0, R)$, then if $x \in \mathrm{spt}(\mu_n)$ is such that $v_{\mu_n}(x) = 0$, we have

$$0 = v_{\mu_n}(x) = \frac{1}{d}x - \int_{\mathbb{S}^{d-1}} \bar{\gamma}_{n,\theta}(\langle x | \theta \rangle)\theta d\theta \tag{173}$$

where for every $\theta \in \mathbb{S}^{d-1}$, $\gamma_{n,\theta}$ is the optimal transport plan between $\mu_{n,\theta}$ and $\rho_\theta$, so that

$$|x| \leq d \left| \int_{\mathbb{S}^{d-1}} \bar{\gamma}_\theta(\langle x | \theta \rangle)\theta d\theta \right| \leq d \int_{\mathbb{S}^{d-1}} |\bar{\gamma}_\theta(\langle x | \theta \rangle)| d\theta \leq dR \tag{174}$$

Since $v_{\mu_n} = 0$ $\mu_n$-almost everywhere, this implies that $\mu_n$ is supported in $\Omega' = B(0, dR)$, and so is $\mu$.

Now consider $\xi : \Omega \mapsto \mathbb{R}^d$ a continuous vector field. For every $n$ and $t \in \mathbb{R}$, applying Proposition 4.7(b) to $t\xi$, we have

$$\mathrm{SW}_2^2((\mathrm{Id} + t\xi)_{\#}\mu_n, \rho) \leq \mathrm{SW}_2^2(\mu_n, \rho) + 2t\langle v_{\mu_n} | \xi \rangle_{L^2(\mu_n)} + \frac{1}{d}t^2\|\xi\|_{L^2(\mu_n)}^2 \tag{175}$$

$$\leq \mathrm{SW}_2^2(\mu_n, \rho) + \frac{1}{d}t^2\|\xi\|_{L^2(\mu_n)}^2 \tag{176}$$

since $v_{\mu_n} = 0$. Letting $n \to \infty$, we thus find

$$\mathrm{SW}_2^2((\mathrm{Id} + t\xi)_{\#}\mu, \rho) \leq \mathrm{SW}_2^2(\mu, \rho) + \frac{1}{d}t^2\|\xi\|_{L^2(\mu)}^2 \tag{177}$$

(Recall that $\mathrm{SW}_2 \leq \mathrm{W}_2$ and that on compact spaces, weak convergence coincide with convergence in the $\mathrm{W}_2$ topology). But since $\mu$ is by assumption without atoms, by Proposition 4.7(c), $t \mapsto \mathrm{SW}_2^2((\mathrm{Id} + t\xi)_{\#}\mu, \rho)$ is differentiable at 0 with derivative $2\langle v_\mu | \xi \rangle_{L^2(\mu)}$, so this inequality implies $\langle v_\mu | \xi \rangle_{L^2(\mu)} = 0$. Since this holds for every continuous vector field $\xi : \Omega \mapsto \mathbb{R}^d$, by a density argument we conclude that $v_\mu = 0$ in $L^2(\mu, \mathbb{R}^d)$, and $\mu$ is indeed a barycentric Lagrangian critical point for $\mathrm{SW}_2^2(\cdot, \rho)$. This finishes the proof. $\qquad\square$

### B.9. Proof of Proposition 5.1

First, let $\mu = \frac{\pi}{8}\mathcal{H}_{|[-\frac{4}{\pi}, \frac{4}{\pi}]}$ and let $\rho$ be the sliced-uniform measure, of which we recall the definition below.

**Definition B.13.** The probability measure $\rho \in \mathcal{P}(\mathbb{R}^2)$ supported on the unit open ball $B(0, 1)$ of the plane with the density $f(x) = \frac{1}{2\pi} \frac{1}{\sqrt{1-|x|^2}}$ is such that in every direction $\theta \in \mathbb{S}^{d-1}$, its projection $P_{\theta\#}\rho$ is the normalized restriction of the Lebesgue measure to $[-1; 1]$. We'll call $\rho$ the (two-dimensional) *sliced-uniform measure* on $[-1; 1]$.

As explained in Definition B.13, each projection $P_{\theta\#}\rho$ is the normalized restriction of the Lebesgue measure to $[-1, 1]$. Indeed, the density of $P_{e1\#}\rho$ at $x \in [-1; 1]$ is given by

$$P_{e1\#}\rho(x) = \frac{1}{2\pi} \int_{-\sqrt{1-x^2}}^{\sqrt{1-x^2}} \frac{1}{\sqrt{1-x^2-y^2}}dy = \frac{1}{2\pi} \int_{-1}^{1} \frac{1}{\sqrt{1-t^2}}dt = \frac{1}{2\pi} \int_{-\frac{\pi}{2}}^{\frac{\pi}{2}} d\theta = \frac{1}{2} \tag{178}$$

with the changes of variables $y = \sqrt{1-x^2}t$, $t = \sin\theta$. By symmetry, the same result holds for all $\theta$.

Then, identifying $\mathbb{S}^1 \simeq (-\pi, \pi] \simeq [0, 2\pi)$, we have for every direction $\theta$, $\rho_\theta = \frac{1}{2}\mathcal{L}_{[-1,1]}^1$, and when $\theta \neq \pm\frac{\pi}{2}$, we have $\mu_\theta = \frac{\pi}{8|c_\theta|}\mathcal{L}_{[-\frac{4|c_\theta|}{\pi}, \frac{4|c_\theta|}{\pi}]}^1$, with the notation $c_\theta = \cos(\theta)$ and $s_\theta = \sin(\theta)$ (in the vertical direction, $\mu_{\pm\frac{\pi}{2}} = \delta_0$). The optimal

transport map from $\mu_\theta$ to $\rho_\theta$ is then $T_\theta(x) = \frac{\pi}{4|c_\theta|}x$. If $x = (x_1, 0) = x_1 e1 \in \mathrm{spt}(\mu) = [-1, 1] \times \{0\}$, where $(e_1, e_2)$ is the canonical basis of $\mathbb{R}^2$, we have (noting $\vec{\theta} = (c_\theta, s_\theta)^T$, with the vector notation to differentiate with the scalar angle $\theta$),

$$d \int_{-\pi}^{\pi} T_\theta(\langle x|\vec{\theta}\rangle)\vec{\theta}\frac{d\theta}{2\pi} = 2 \int_{-\pi}^{\pi} T_\theta(x_1 c_\theta) \begin{pmatrix} c_\theta \\ s_\theta \end{pmatrix} \frac{d\theta}{2\pi} \tag{179}$$

$$= \frac{\pi}{2}x_1 \int_{-\pi}^{\pi} \frac{c_\theta}{|c_\theta|} \begin{pmatrix} c_\theta \\ s_\theta \end{pmatrix} \frac{d\theta}{2\pi} \tag{180}$$

$$= \frac{1}{4}x_1 \int_{-\pi}^{\pi} |c_\theta| d\theta e_1 \tag{181}$$

(We see that the integral on the second coordinate cancels by antisymmetry). Since

$$\frac{1}{4} \int_{-\pi}^{\pi} |c_\theta| d\theta = \frac{1}{2} \int_{0}^{\pi} |c_\theta| d\theta = \int_{0}^{\pi/2} c_\theta d\theta = 1 \tag{182}$$

we thus have

$$x_1 e_1 = d \int_{-\pi}^{\pi} T_\theta(\langle x|\vec{\theta}\rangle)\vec{\theta}\frac{d\theta}{2\pi} \tag{183}$$

that is $v_\mu(x) = 0$. This proves that $\mu$ satisfies Definition 4.2 and is therefore a barycentric Lagrangian critical point for $\mathrm{SW}_2^2(\cdot, \rho)$.

Now, we consider the case where $d > 1$, $\rho = \mathcal{N}(0, I_d)$ and $\mu = (Id, 0_{d-1})\#\mathcal{N}(0, \alpha_d^2)$ with $\alpha_d = d \int_{\mathbb{S}^{d-1}} |\langle\theta|e_1\rangle| d\theta$. For every $\theta \in \mathbb{S}^{d-1}$, we have $\rho_\theta = \mathcal{N}(0, 1)$. Noting $(e_1, ..., e_d)$ the canonical basis of $\mathbb{R}^d$, when $\langle\theta|e_1\rangle \neq 0$, we have $\mu_\theta = P_{\theta\#}\mu = \mathcal{N}(0, (\alpha_d|\langle\theta|e_1\rangle|)^2)$, and when $\langle\theta|e_1\rangle = 0$, $\mu_\theta = \delta_0$. Therefore, the optimal transport map from $\mu_\theta$ to $\rho_\theta$ is given by $T_\theta : x \mapsto (\alpha_d|\langle\theta|e_1\rangle|)^{-1}x$. Let $x = x_1 e_1 \in \mathrm{spt}\mu = \mathbb{R} \times \{0\}^{d-1}$, then we have

$$d \int_{\mathbb{S}^{d-1}} T_\theta(\langle x|\theta\rangle)\theta d\theta = d \int_{\mathbb{S}^{d-1}} T_\theta(x_1 \langle\theta|e_1\rangle)\theta d\theta \tag{184}$$

$$= dx_1 \int_{\mathbb{S}^{d-1}} \frac{\langle\theta|e_1\rangle}{\alpha_d|\langle\theta|e_1\rangle|}\theta d\theta \tag{185}$$

$$\tag{186}$$

By symmetry we see that the components of this integral along $e_2, ..., e_d$ are zero, and thus

$$d \int_{\mathbb{S}^{d-1}} T_\theta(\langle x|\theta\rangle)\theta d\theta = dx_1 \int_{\mathbb{S}^{d-1}} \frac{\langle\theta|e_1\rangle^2}{\alpha_d|\langle\theta|e_1\rangle|}d\theta e_1 \tag{187}$$

$$= x_1 \frac{1}{\alpha_d}d \int_{\mathbb{S}^{d-1}} |\langle\theta|e_1\rangle| d\theta e_1 \tag{188}$$

$$= x_1 e_1 \text{ by definition of } \alpha_d \tag{189}$$

This proves that $\mu$ satisfies Definition 4.2 and is therefore a barycentric Lagrangian critical point for $\mathrm{SW}_2^2(\cdot, \rho)$. $\qquad\square$

## B.10. Proof of Proposition 5.2

*Sketch of proof.* Up to translating, rotating, and rescaling, we may decompose $\mu$ as $\mu = (1 - \lambda)\mu_0 + \lambda\mu_1$ where $\mu_1 = \frac{1}{2}\mathcal{H}^1_{|[-1,1]\times\{0\}}$. For every $\theta \in \mathbb{S}^1$, let $\hat{\gamma}_\theta \in \Pi(\mu, \rho)$ be such that $(P_\theta, P_\theta)\#\hat{\gamma}_\theta$ is optimal between $\mu_\theta$ and $\rho_\theta$. Then we can decompose $\hat{\gamma}_\theta$ and $\rho$ into

$$\hat{\gamma}_\theta = (1 - \lambda)\hat{\gamma}_{\theta,0} + \lambda\hat{\gamma}_{\theta,1} \tag{190}$$

and

$$\rho = (1 - \lambda)\rho_{\theta,0} + \lambda\rho_{\theta,1}, \tag{191}$$

where $\hat{\gamma}_{\theta,i}$ couples $\mu_i$ and $\rho_{\theta,i} \in \mathcal{P}_2(\mathbb{R}^d)$. Denoting $\rho_{\theta,i,\theta}$ the projection of $\rho_{\theta,i}$ on $\theta$ for $i = 0, 1$, these decompositions verify

$$\mathrm{SW}_2^2(\mu^t, \rho) \leq (1 - \lambda) \int_{\mathbb{S}^1} \mathrm{W}_2^2(\mu_{0,\theta}^t, \rho_{\theta,0,\theta}) d\theta + \lambda \int_{\mathbb{S}^1} \mathrm{W}_2^2(\mu_{1,\theta}^t, \rho_{\theta,1,\theta}) d\theta, \tag{192}$$

with equality at $t = 0$. We bound separately the two terms of the right hand side. The first term can be easily bounded by

$$(1 - \lambda) \int_{\mathbb{S}^1} \mathrm{W}_2^2(\mu_{0,\theta}, \rho_{\theta,0,\theta}) d\theta + O(t^2). \tag{193}$$

All that is left is then to show that the second term can be bounded for any $C > 0$, on a neighborhood of $t = 0$, by

$$\int_{\mathbb{S}^{d-1}} \mathrm{W}_2^2(\mu_{1,\theta}, \rho_{\theta,1,\theta}) d\theta - Ct^2. \tag{194}$$

We obtain such a bound by writing $\mathrm{W}_2^2(\mu_{1,\theta}^t, \rho_{\theta,1,\theta}) = \|F_{\mu_{1,\theta}^t}^{-1} - F_{\rho_{\theta,1,\theta}}^{-1}\|_{L^2([0,1])}^2$, and by making use of an explicit expression of $F_{\mu_{1,\theta}^t}^{-1}$ and of its symmetry to compute a Taylor expansion of

$$\int_{\mathbb{S}^{d-1}} \mathrm{W}_2^2(\mu_{1,\theta}^t, \rho_{\theta,1,\theta}) d\theta \tag{195}$$

and bound it from above in the desired way. $\qquad \square$

Up to translating, rotating and rescaling, we may assume that $S = [-1, 1] \times \{0, 0\}$ and $\vec{n} = e_2$. Since $a\mathcal{H}_{|S}^1 \leq \mu$, we write

$$\mu = (1 - \lambda)\mu_0 + \lambda\mu_1 \tag{196}$$

where $\lambda \in [0, 1]$ and $\mu_0, \mu_1$ are probability measures such that $\mu_1 = \frac{1}{2}\mathcal{H}_{[-1,1] \times \{0\}}^1$ and $\lambda = 2a$. For every $\theta \in \mathbb{S}^1$, let $\hat{\gamma}_\theta \in \Pi(\mu, \rho)$ be such that $(P_\theta, P_\theta)_{\#}\hat{\gamma}_\theta$ is an optimal transport plan between $\mu_\theta$ and $\rho_\theta$. Using Theorem A.2 we can disintegrate $\hat{\gamma}_\theta$ with respect to $\mu$, thus writing $d\hat{\gamma}_\theta(x, y) = d\hat{\gamma}_\theta(y|x)d\mu(x)$, and we define two probability measures $\rho_{\theta,0}, \rho_{\theta,1} \in \mathcal{P}_2(\mathbb{R}^2)$ by

$$\int \varphi(y)\rho_{\theta,i}(y) := \int \int \varphi(y)d\hat{\gamma}_\theta(y|x)d\mu_i(x), \quad i \in \{0, 1\}, \varphi \in C_b(\mathbb{R}^2) \tag{197}$$

and two transport plans $\hat{\gamma}_{\theta,i} \in \Pi(\mu_i, \rho_{\theta,i})$ by $d\hat{\gamma}_{\theta,i}(x, y) = d\hat{\gamma}_\theta(y|x)d\mu_i(x)$. By (Villani, 2008, Theorem 4.6), the $(P_\theta, P_\theta)_{\#}\hat{\gamma}_{\theta,i}$ are actually optimal between their margins. In fact, we have

$$\mathrm{W}_2^2(\mu_\theta^t, \rho_\theta) \leq (1 - \lambda)\mathrm{W}_2^2(\mu_{0,\theta}^t, \rho_{\theta,0,\theta}) + \lambda\mathrm{W}_2^2(\mu_{1,\theta}^t, \rho_{\theta,1,\theta}) \tag{198}$$

where $\nu^t := \frac{1}{2}(\tau_{te_2\#}\nu + \tau_{-te_2\#}\nu)$ for any measure $\nu$, with equality at $t = 0$. We will establish bounds separately on $\mathrm{W}_2^2(\mu_{0,\theta}^t, \rho_{\theta,0,\theta})$ and $\mathrm{W}_2^2(\mu_{1,\theta}^t, \rho_{\theta,1,\theta})$. First, we notice that

$$\int_{\mathbb{S}^1} \mathrm{W}_2^2(\mu_{0,\theta}^t, \rho_{\theta,0,\theta}) d\theta \leq \int_{\mathbb{S}^1} \mathrm{W}_2^2(\mu_{0,\theta}, \rho_{\theta,0,\theta}) d\theta + \frac{1}{d}t^2 \tag{199}$$

Indeed, if we consider the transport plan $\hat{\gamma}_{\theta,0}^t \in \Pi(\mu_0^t, \rho_{\theta,0})$ defined by

$$\hat{\gamma}_{\theta,0}^t := \frac{1}{2}((\tau_{te_2}, Id)_{\#}\hat{\gamma}_{\theta,0} + (\tau_{-te_2}, Id)_{\#}\hat{\gamma}_{\theta,0}) \tag{200}$$

we have

$$W_2^2(\mu_{0,\theta}^t, \rho_{\theta,0,\theta}) \leq \int \langle x - y | \theta \rangle^2 d\hat{\gamma}_{\theta,0}^t(x,y) \tag{201}$$

$$\leq \int \frac{1}{2}(\langle x + te_2 - y | \theta \rangle^2 + \langle x - te_2 - y | \theta \rangle^2) d\hat{\gamma}_{\theta,0}(x,y) \tag{202}$$

$$\leq \int \langle x - y | \theta \rangle^2 + t^2 \langle e_2 | \theta \rangle^2) d\hat{\gamma}_{\theta,0}(x,y) \tag{203}$$

$$\leq W_2^2(\mu_{0,\theta}, \rho_{\theta,0,\theta}) + t^2 \langle e_2 | \theta \rangle^2 \tag{204}$$

and by integrating on the sphere we get (199).

Now, all we need to prove is that for every $C > 0$, there exists a neighborhood of $t = 0$ in which

$$\int_{\mathbb{S}^1} W_2^2(\mu_{1,\theta}^t, \rho_{\theta,1,\theta}) d\theta \leq \int_{\mathbb{S}^1} W_2^2(\mu_{1,\theta}, \rho_{\theta,1,\theta}) d\theta - Ct^2 \tag{205}$$

By summing it with (199), we obtain the proposition's statement. To derive this bound, we look at the quantile functions : for every $\theta \in \mathbb{S}^1$, we have

$$W_2^2(\mu_{1,\theta}^t, \rho_{\theta,1,\theta}) d\theta = \|F_{\mu_{1,\theta}^t}^{-1} - F_{\rho_{\theta,1,\theta}}^{-1}\|_{L^2([0,1])}^2 \tag{206}$$

$$= \|F_{\mu_{1,\theta}^t}^{-1} - F_{\mu_{1,\theta}}^{-1} + F_{\mu_{1,\theta}}^{-1} - F_{\rho_{\theta,1,\theta}}^{-1}\|_{L^2([0,1])}^2 \tag{207}$$

$$= \|F_{\mu_{1,\theta}^t}^{-1} - F_{\mu_{1,\theta}}^{-1}\|_{L^2([0,1])}^2 + 2\langle F_{\mu_{1,\theta}^t}^{-1} - F_{\mu_{1,\theta}}^{-1} | F_{\mu_{1,\theta}}^{-1} - F_{\rho_{\theta,1,\theta}}^{-1}\rangle_{L^2([0,1])} \tag{208}$$

$$+ \|F_{\mu_{1,\theta}}^{-1} - F_{\rho_{\theta,1,\theta}}^{-1}\|_{L^2([0,1])}^2 \tag{209}$$

$$= W_2^2(\mu_{1,\theta}^t, \mu_{1,\theta}) + W_2^2(\mu_{1,\theta}, \rho_{\theta,1,\theta}) + 2\langle F_{\mu_{1,\theta}^t}^{-1} - F_{\mu_{1,\theta}}^{-1} | F_{\mu_{1,\theta}}^{-1} - F_{\rho_{\theta,1,\theta}}^{-1}\rangle_{L^2([0,1])} \tag{210}$$

We easily see that $W_2^2(\mu_{1,\theta}^t, \mu_{1,\theta}) \leq W_2^2(\mu_1^t, \mu_1) \leq t^2$. Therefore, we simply need to show that for every $C > 0$, there exists a neighborhood of $t = 0$ on which

$$\int_{\mathbb{S}^1} \langle F_{\mu_{1,\theta}^t}^{-1} - F_{\mu_{1,\theta}}^{-1} | F_{\mu_{1,\theta}}^{-1} - F_{\rho_{\theta,1,\theta}}^{-1}\rangle_{L^2([0,1])} d\theta \leq -Ct^2 \tag{211}$$

Since $\mu_1 = \frac{1}{2}\mathcal{H}_{|[-1,1]\times\{0\}}^1$, we have, for every $t$,

$$\mu_1^t = \frac{1}{4}(\mathcal{H}_{|[-1,1]\times\{-t\}}^1 + \mathcal{H}_{|[-1,1]\times\{t\}}^1) \tag{212}$$

Now let $\theta \in \mathbb{S}^1 \setminus \{\pm\frac{\pi}{2}\}$ (we make again the identification $\mathbb{S}^1 \simeq \mathbb{R}/2\pi\mathbb{Z}$). The projections of $\mu_1^t$ and $\mu_1$ on $\mathbb{R}\theta$ are

$$\mu_{1,\theta}^t = \frac{1}{4|c_\theta|}(\lambda_{A_{\theta,t}^-} + \lambda_{A_{\theta,t}^+}), \quad A_{\theta,t}^\pm = [\pm|ts_\theta| - |c_\theta|, \pm|ts_\theta| + |c_\theta|] \tag{213}$$

and

$$\mu_{1,\theta} = \frac{1}{2c_\theta}\lambda_{A_\theta}, \quad A_\theta = [-|c_\theta|, |c_\theta|] \tag{214}$$

Therefore the quantile function of $\mu_{1,\theta}$ is simply

$$F_{\mu_{1,\theta}}^{-1}(x) = -|c_\theta| + 2|c_\theta|x, \quad x \in [0,1] \tag{215}$$

In the following, since for any $\theta$ and any measures $\nu_1, \nu_2 \in \mathcal{P}(\mathbb{R}^2)$, $W_2^2(\nu_{1,\theta+\pi}, \nu_{2,\theta+\pi}) = W_2^2(\nu_{1,\theta}, \nu_{2,\theta})$, we can restrict ourselves to $\theta \in (-\frac{\pi}{2}, \frac{\pi}{2})$. To compute the quantile function of $\mu_{1,\theta}^t$, we then need to consider two cases.

- First, when $|\theta| \in [0, \arctan(1/|t|)]$, the two segments $A_{\theta,t}^{\pm}$ overlap. Their union can then be decomposed into three segments where the density of $\mu_{1,\theta}^t$ is constant :

$$B_- \cup B_0 \cup B_+ = [-|c_\theta| - |ts_\theta|, -|c_\theta| + |ts_\theta|] \tag{216}$$
$$\cup [-|c_\theta| + |ts_\theta|, |c_\theta| - |ts_\theta|] \tag{217}$$
$$\cup [|c_\theta| - |ts_\theta|, |c_\theta| + |ts_\theta|] \tag{218}$$

On $B_\pm$, the density is $\frac{1}{4|c_\theta|}$ while on $B_0$, it is $\frac{1}{2|c_\theta|}$. One can check that the quantile function of $\mu_{1,\theta}^t$ and $\mu_\theta$ is then (using the shorthand notation $t_\theta = \tan(\theta)$)

$$F_{\mu_{1,\theta}^t}^{-1}(x) = \begin{cases} -|c_\theta| - |ts_\theta| + 4|c_\theta|x & \text{for } x \in \left[0, \frac{|t|}{2}|t_\theta|\right] \\ -|c_\theta| + |ts_\theta| + 2|c_\theta|\left(x - \frac{|t|}{2}|t_\theta|\right) & \text{for } x \in \left[\frac{|t|}{2}|t_\theta|, 1 - \frac{|t|}{2}|t_\theta|\right] \\ |c_\theta| - |ts_\theta| + 4|c_\theta|\left(x - 1 + \frac{|t|}{2}|t_\theta|\right) & \text{for } x \in \left[1 - \frac{|t|}{2}|t_\theta|, 1\right] \end{cases} \tag{219}$$

- Second, when $|\theta| \in (\arctan(1/|t|), \pi/2)$, the two segments $A_{\theta,t}^{\pm}$ do not overlap, in which case the quantile function of $\mu_{1,\theta}^t$ is

$$F_{\mu_{1,\theta}^t}^{-1}(x) = \begin{cases} -|c_\theta| - |ts_\theta| + 4|c_\theta|x & \text{for } x \in \left[0, \frac{1}{2}\right] \\ -|c_\theta| + |ts_\theta| + 4|c_\theta|\left(x - \frac{1}{2}\right) & \text{for } x \in \left(\frac{1}{2}, 1\right] \end{cases} \tag{220}$$

Denoting $m_{t,\theta} = \frac{1}{2}\min(1, |tt_\theta|)$, we can actually condense the two previous expressions of $F_{\mu_{1,\theta}^t}^{-1}$ into a single one valid for every $\theta \in (-\pi/2, \pi/2)$ :

$$F_{\mu_{1,\theta}^t}^{-1}(x) = \begin{cases} -|c_\theta| - |ts_\theta| + 4|c_\theta|x & \text{for } x \in [0, m_{t,\theta}] \\ -|c_\theta| + 2|c_\theta|x & \text{for } x \in (m_{t,\theta}, 1 - m_{t,\theta}] \\ -|c_\theta| + |ts_\theta| + 4|c_\theta|\left(x - \frac{1}{2}\right) & \text{for } x \in (1 - m_{t,\theta}, 1] \end{cases} \tag{221}$$

We see in particular that

- $F_{\mu_{1,\theta}^t}^{-1}(x) = F_{\mu_{1,\theta}}^{-1}(x)$ for every $x \in (m_{t,\theta}, 1 - m_{t,\theta}]$

- For every $t \in \mathbb{R}$ and $x \in [0, 1]$, $F_{\mu_{1,\theta}^t}^{-1}(1 - x) = 1 - F_{\mu_{1,\theta}^t}^{-1}(x)$ (in fact, we only needed to use the symmetry of $\mu_{1,\theta}^t$ to see this)

Therefore, we have

$$\langle F_{\mu_{1,\theta}^t}^{-1} - F_{\mu_{1,\theta}}^{-1} | F_{\mu_{1,\theta}}^{-1} - F_{\rho_{\theta,1,\theta}}^{-1} \rangle_{L^2([0,1])} = \int_0^1 (F_{\mu_{1,\theta}^t}^{-1}(x) - F_{\mu_{1,\theta}}^{-1}(x))(F_{\mu_{1,\theta}}^{-1}(x) - F_{\rho_{\theta,1,\theta}}^{-1}(x))dx$$

$$= \int_0^{m_{t,\theta}} (F_{\mu_{1,\theta}^t}^{-1}(x) - F_{\mu_{1,\theta}}^{-1}(x))(F_{\mu_{1,\theta}}^{-1}(x) - F_{\rho_{\theta,1,\theta}}^{-1}(x))dx$$

$$+ \int_{1-m_{t,\theta}}^1 (F_{\mu_{1,\theta}^t}^{-1}(x) - F_{\mu_{1,\theta}}^{-1}(x))(F_{\mu_{1,\theta}}^{-1}(x) - F_{\rho_{\theta,1,\theta}}^{-1}(x))dx$$

$$= \int_0^{m_{t,\theta}} (F_{\mu_{1,\theta}^t}^{-1}(x) - F_{\mu_{1,\theta}}^{-1}(x))(F_{\mu_{1,\theta}}^{-1}(x) - F_{\rho_{\theta,1,\theta}}^{-1}(x))dx$$

$$+ \int_0^{m_{t,\theta}} (F_{\mu_{1,\theta}^t}^{-1}(1 - x) - F_{\mu_{1,\theta}}^{-1}(1 - x))(F_{\mu_{1,\theta}}^{-1}(1 - x) - F_{\rho_{\theta,1,\theta}}^{-1}(1 - x))dx$$

$$= \int_0^{m_{t,\theta}} (F_{\mu_{1,\theta}^t}^{-1}(x) - F_{\mu_{1,\theta}}^{-1}(x))(F_{\mu_{1,\theta}}^{-1}(x) - F_{\rho_{\theta,1,\theta}}^{-1}(x) - (F_{\mu_{1,\theta}}^{-1}(1 - x) - F_{\rho_{\theta,1,\theta}}^{-1}(1 - x)))dx$$

We have

$$F_{\mu_{1,\theta}^t}^{-1}(x) - F_{\mu_{1,\theta}}^{-1}(x) = 2|c_\theta||x - |ts_\theta| = 2|c_\theta|(x - \frac{1}{2}|t_\theta|) \tag{222}$$

$$F_{\mu_{1,\theta}}^{-1}(x) - F_{\mu_{1,\theta}}^{-1}(1 - x) = -|c_\theta| + 2|c_\theta|x - (-|c_\theta| + 2|c_\theta|(1 - x)) = 4|c_\theta|(x - \frac{1}{2}) \tag{223}$$

for $x \in [0, m_{t,\theta}]$. If for $x \in [0, 1] \setminus \{\frac{1}{2}\}$ we note

$$G_\theta(x) := \frac{F_{\rho_{\theta,1,\theta}}^{-1}(x) - F_{\rho_{\theta,1,\theta}}^{-1}(1 - x)}{x - \frac{1}{2}} \tag{224}$$

then we have

$$\langle F_{\mu_{1,\theta}^t}^{-1} - F_{\mu_{1,\theta}}^{-1} | F_{\mu_{1,\theta}}^{-1} - F_{\rho_{\theta,1,\theta}}^{-1} \rangle_{L^2([0,1])} = \int_0^{m_{t,\theta}} (x - \frac{1}{2})(4|c_\theta| - G_\theta(x))2|c_\theta|(x - \frac{1}{2}|tt_\theta|)dx \tag{225}$$

However, our hypothesis that for every $\theta$ the density of $\rho_\theta$ is bounded from above by $b > 0$ allows us to derive a lower bound for $G_\theta$. Indeed, since $\rho = (1 - \lambda)\rho_{\theta,0} + \lambda\rho_{\theta,1}$, we have $\rho_{\theta,1} \leq \frac{1}{\lambda}\rho$ and thus $\rho_{\theta,1,\theta} \leq \tilde{b}$ with $\tilde{b} = \frac{b}{\lambda}$. Then, using the shorthand notations $F_\theta = F_{\rho_{\theta,1,\theta}}$ and $F_\theta^{-1} = F_{\rho_{\theta,1,\theta}}^{-1}$, for almost every $x \in [0, 1]$,

$$F_\theta^{-1}(F_\theta(x)) = x \tag{226}$$

Let $x = \alpha + h$ with $h > 0$. Since

$$F_\theta(x) = F_\theta(\alpha) + \rho_{\theta,1,\theta}((\alpha, \alpha + h]) \leq F_\theta(\alpha) + \tilde{b}h \tag{227}$$

we have

$$\alpha + h = F_\theta^{-1}(F_\theta(\alpha + h)) \leq F_\theta^{-1}(F_\theta(\alpha) + \tilde{b}h) \tag{228}$$

Similarly, if $x = \alpha - h$ with $h > 0$, we have

$$F_\theta(x) = F_\theta(\alpha) - \rho_{\theta,1,\theta}((\alpha - h, \alpha]) \geq F_\theta(\alpha) - \tilde{b}h \tag{229}$$

thus

$$\alpha - h = F_\theta^{-1}(F_\theta(\alpha - h)) \geq F_\theta^{-1}(F_\theta(\alpha) - \tilde{b}h) \tag{230}$$

and thus we have

$$-2h \geq F_\theta^{-1}(F_\theta(\alpha) - \tilde{b}h) - F_\theta^{-1}(F_\theta(\alpha) + \tilde{b}h) \tag{231}$$

Now, pick $\alpha$ such that $F_\theta(\alpha) = \frac{1}{2}$. Let $x \in [0, 1/2]$, and let $h > 0$ be such that $x = \frac{1}{2} - \tilde{b}h$. Then, substituting the value of $x$ in the previous equation, we get

$$F_\theta^{-1}(x) - F_\theta^{-1}(1 - x) \leq -2h = -\frac{2}{\tilde{b}}(\frac{1}{2} - x) \tag{232}$$

$$G_\theta(x) \geq \frac{2}{\tilde{b}} > 0 \tag{233}$$

for almost every $x \in [0, 1/2]$. Thus, since by definition of $m_{t,\theta}$, $(x - \frac{1}{2})(x - \frac{1}{2}|tt_\theta|) \geq 0$ for $x \in [0, m_{t,\theta}]$, this means that for every $\theta \in (-\pi/2, \pi/2)$,

$$\langle F_{\mu_{1,\theta}^t}^{-1} - F_{\mu_{1,\theta}}^{-1} | F_{\mu_{1,\theta}}^{-1} - F_{\rho_{\theta,1,\theta}}^{-1} \rangle_{L^2([0,1])} \leq 2|c_\theta|(4|c_\theta| - \frac{2}{\tilde{b}}) \int_0^{m_{t,\theta}} (x - \frac{1}{2})(x - \frac{1}{2}|tt_\theta|)dx \tag{234}$$

Let's compute the integral on the right-hand side :

$$\int_0^{m_{t,\theta}} (x - \frac{1}{2})(x - \frac{1}{2}|tt_\theta|)dx = \int_0^{m_{t,\theta}} x^2 - \frac{1}{2}(1 + |tt_\theta|)x + \frac{1}{4}|tt_\theta|dx \tag{235}$$

$$= \frac{m_{t,\theta}^3}{3} - \frac{1}{4}(1 + |tt_\theta|)m_{t,\theta}^2 + \frac{1}{4}|tt_\theta|m_{t,\theta} \tag{236}$$

If $|\theta| \leq \arctan(1/|t|)$, then $m_{t,\theta} = \frac{1}{2}|tt_\theta|$ and

$$\int_0^{m_{t,\theta}} (x - \frac{1}{2})(x - \frac{1}{2}|tt_\theta|)dx = \frac{|tt_\theta|^3}{24} - \frac{1}{16}(1 + |tt_\theta|)|tt_\theta|^2 + \frac{1}{8}|tt_\theta|^2 \tag{237}$$

$$= \frac{|tt_\theta|^2}{16} - \frac{|tt_\theta|^3}{48} \tag{238}$$

$$= \frac{1}{16}|tt_\theta|^2(1 - \frac{1}{3}|tt_\theta|) \tag{239}$$

and in fact, since $|tt_\theta| \leq 1$ when $|\theta| \leq \arctan(1/|t|)$, we have

$$\int_0^{m_{t,\theta}} (x - \frac{1}{2})(x - \frac{1}{2}|tt_\theta|)dx = \frac{1}{16}|tt_\theta|^2(1 - \frac{1}{3}|tt_\theta|) \geq \frac{1}{24}|tt_\theta|^2 > 0 \tag{240}$$

Let $\theta_1 \in (0, \pi/2)$ be such that $4c_{\theta_1} - \frac{2}{\tilde{b}} \leq -\frac{1}{\tilde{b}}$ and let $t$ be small enough so that $\alpha_t := \arctan(1/|t|) > \theta_1$. Then :

- If $|\theta| \in (\alpha_t, \pi/2)$, then we can simply bound (234) from above by 0

$$\langle F_{\mu_{1,\theta}^t}^{-1} - F_{\mu_{1,\theta}}^{-1} | F_{\mu_{1,\theta}}^{-1} - F_{\rho_{\theta,1,\theta}}^{-1} \rangle_{L^2([0,1])} \leq 0 \tag{241}$$

as $4|c_\theta| - \frac{2}{\tilde{b}} < 0$ and the integral is positive. Thus

$$\int_{[-\pi/2,-\alpha_t] \cup [\alpha_t,\pi/2]} \langle F_{\mu_{1,\theta}^t}^{-1} - F_{\mu_{1,\theta}}^{-1} | F_{\mu_{1,\theta}}^{-1} - F_{\rho_{\theta,1,\theta}}^{-1} \rangle_{L^2([0,1])} d\theta \leq 0 \tag{242}$$

- If $|\theta| \in [0, \theta_1)$ then, combining (234) and (240) we have

$$\langle F_{\mu_{1,\theta}^t}^{-1} - F_{\mu_{1,\theta}}^{-1} | F_{\mu_{1,\theta}}^{-1} - F_{\rho_{\theta,1,\theta}}^{-1} \rangle_{L^2([0,1])} \leq 2|c_\theta|(4|c_\theta| - \frac{2}{\tilde{b}}) \int_0^{m_{t,\theta}} (x - \frac{1}{2})(x - \frac{1}{2}|tt_\theta|)dx \tag{243}$$

$$\leq 2|c_\theta|(4|c_\theta| - \frac{2}{\tilde{b}}) \frac{1}{16}|tt_\theta|^2(1 - \frac{1}{3}|tt_\theta|) \tag{244}$$

$$\leq \frac{1}{4}(2 + \frac{1}{\tilde{b}})t^2 t_{\theta_1}^2 (1 + \frac{1}{3}|tt_{\theta_1}|) \tag{245}$$

Therefore, we conclude that there exists some constant $C_0 > 0$ such that

$$\int_{[-\alpha_t,-\theta_1] \cup [\theta_1,\alpha_t]} \langle F_{\mu_{1,\theta}^t}^{-1} - F_{\mu_{1,\theta}}^{-1} | F_{\mu_{1,\theta}}^{-1} - F_{\rho_{\theta,1,\theta}}^{-1} \rangle_{L^2([0,1])} d\theta \leq C_0 t^2 + o(t^2) \tag{246}$$

- Finally, if $|\theta| \in [\theta_1, \alpha_t]$ then, again combining (234) and (240), we have

$$\langle F_{\mu_{1,\theta}^t}^{-1} - F_{\mu_{1,\theta}}^{-1} | F_{\mu_{1,\theta}}^{-1} - F_{\rho_{\theta,1,\theta}}^{-1} \rangle_{L^2([0,1])} \leq 2|c_\theta|(4|c_\theta| - \frac{2}{\tilde{b}}) \int_0^{m_{t,\theta}} (x - \frac{1}{2})(x - \frac{1}{2}|tt_\theta|)dx \tag{247}$$

$$\leq 2|c_\theta|(4|c_\theta| - \frac{2}{\tilde{b}}) \frac{1}{16}|tt_\theta|^2(1 - \frac{1}{3}|tt_\theta|) \tag{248}$$

$$\leq -\frac{1}{12\tilde{b}}|c_\theta||tt_\theta|^2 \tag{249}$$

$$\leq -\frac{1}{12\tilde{b}}t^2 \frac{|s_\theta^2|}{|c_\theta|} \leq -\frac{|s_{\theta_1}|^2}{12\tilde{b}}t^2 \frac{1}{|c_\theta|} \tag{250}$$

However, the integral $\int_{\theta_1}^{\alpha_t} \frac{d\theta}{|c_\theta|}$ diverges to infinity when $t \mapsto 0$. Indeed, using the development

$$\alpha_t := \arctan(1/|t|) = \frac{\pi}{2} - \arctan(|t|) = \frac{\pi}{2} - |t| + o(t^2) \tag{251}$$

we have

$$\int_{\theta_1}^{\theta_t} \frac{d\theta}{|c_\theta|} = \int_{\sin(\theta_1)}^{\sin(\alpha_t)} \frac{du}{1-u^2} \tag{252}$$

$$= \frac{1}{2}[\ln(1+u) - \ln(1-u)]_{\sin(\theta_1)}^{\sin(\alpha_t)} \tag{253}$$

$$= \frac{1}{2}(\ln(1+\sin(\alpha_t)) - \ln(1-\sin(\alpha_t))) + C \tag{254}$$

$$= \frac{1}{2}\left(\ln\left(1+\sin\left(\frac{\pi}{2}-|t|+o(t^2)\right)\right) - \ln\left(1-\sin\left(\frac{\pi}{2}-|t|+o(t^2)\right)\right)\right) + C \tag{255}$$

$$= \frac{1}{2}(\ln(1+\cos(|t|+o(t^2))) - \ln(1-\cos(|t|+o(t^2)))) + C \tag{256}$$

$$= \frac{1}{2}(\ln(1+\cos(|t|+o(t^2))) - \ln(1-\cos(|t|+o(t^2)))) + C \tag{257}$$

$$= \frac{1}{2}\left(\ln\left(2-\frac{1}{2}t^2+o(t^2)\right) - \ln\left(\frac{1}{2}t^2+o(t^2)\right)\right) + C \tag{258}$$

$$= \frac{1}{2}(\ln(2)+o(1) - 2\ln(t) + \ln(2) + o(1)) + C \tag{259}$$

$$= -\ln(t) + C + o(1) \xrightarrow[t\to 0]{} +\infty \tag{260}$$

Therefore, for any $C > 0$, there exists a neighborhood of $t = 0$ in which,

$$\int_{[-\alpha_t,-\theta_1]\cup[\theta_1,\alpha_t]} \langle F_{\mu_{1,\theta}^t}^{-1} - F_{\mu_{1,\theta}}^{-1} | F_{\mu_{1,\theta}}^{-1} - F_{\rho_{\theta,1,\theta}}^{-1}\rangle_{L^2([0,1])} d\theta \leq -Ct^2 \tag{261}$$

Thus, we can prove (211) by bounding the integral of $\langle F_{\mu_{1,\theta}^t}^{-1} - F_{\mu_{1,\theta}}^{-1} | F_{\mu_{1,\theta}}^{-1} - F_{\rho_{\theta,1,\theta}}^{-1}\rangle$ on $(-\pi/2, \pi/2)$ separately on the three regions $(-\pi/2, -\alpha_t] \cup [\alpha_t, \pi/2)$, $[-\theta_1, \theta_1]$ and $[-\alpha_t, -\theta_1] \cup [\theta_1, \alpha_t]$ using (246), (242) and (261), taking in (261) a constant $C > 0$ big enough to compensate the constant $C_0$ in (246). This concludes the proof. $\square$

## C. Stability and numerical approximation

In this section, we will discuss briefly the regularity properties of (practical) Monte Carlo approximations of the SW objective and what they entail for applying our theoretical understanding of $F$ to practical applications involving $F_L$. The discussion will be similar to the one found in (Tanguy et al., 2024a), although they focus on the discrete setting, where $\rho$ is also a point cloud, whereas we focus on the semi-discrete one.

In practice, the Sliced-Wasserstein distance objective (6) discussed in Section 3 is usually computed through a Monte Carlo estimator to approximate the integral. In the semi-discrete setting, this amounts to approximating the function $F(X)$ discussed in Section 3 with the function $F_L = \frac{1}{2L}\sum_{l=1}^{L} W_2^2(\mu_{P_{\theta_l}(X)}, \rho_{\theta_l})$, where $\theta_1, ..., \theta_L \in \mathbb{S}^{d-1}$ are chosen directions. The latter may vary: for example, they may be uniformly sampled on $\mathbb{S}^{d-1}$ at every step of a stochastic gradient descent (or some other optimization algorithm), or fixed once and for all.

In fact, the local behavior of $F_L$ is quite different from that of $F$, and exhibits a cell structure. Indeed, for every $\sigma \in \mathfrak{S}_n^L$, let $\mathcal{C}_\sigma = \{X \in (\mathbb{R}^d)^N \mid \forall l \in \{1, ..., L\}, \sigma_{\theta_l, X} \text{ is uniquely defined and is } \sigma_l\}$. Then, for every $X \in \mathcal{C}_\sigma$, we have

$$F_L(X) = \frac{1}{2L}\sum_{l=1}^{L}\sum_{i=1}^{N} \int_{V_{\theta_l,i}} |\langle X_{\sigma_l(i)}|\theta_l\rangle - x|^2 d\rho_{\theta_l}(x) \tag{262}$$

which simplifies to

$$F_L(X) = q_\sigma(X) + C_0 \tag{263}$$

with the quadratic function

$$q_\sigma(X) = \frac{1}{2NL}\sum_{l=1}^{L}\sum_{i=1}^{N} |\langle X_{\sigma_l(i)}|\theta_l\rangle - b_{\theta_l,i}|^2 \tag{264}$$

and the constant

$$C_0 = \frac{1}{2L} \sum_{l=1}^{L} \sum_{i=1}^{N} \int_{V_{\theta_l,i}} |x - b_{\theta_l,i}|^2 d\rho_{\theta_l}(x) \tag{265}$$

In fact, $\mathcal{C}_{\boldsymbol{\sigma}}$ can also be written as $\mathcal{C}_{\boldsymbol{\sigma}} = \{X \in (\mathbb{R}^d)^N \mid \forall \boldsymbol{\sigma}' \in \mathfrak{S}_N^L, q_{\boldsymbol{\sigma}'}(X) > q_{\boldsymbol{\sigma}}(X)\}$, from which we can deduce that $\mathcal{C}_{\boldsymbol{\sigma}}$ is an open polyhedral cone, obtained as the intersection of $L(N! - 1)$ half-open planes. Furthermore, $F_L$ is actually the infimum of the $C_0 + q_{\boldsymbol{\sigma}}$:

$$F_L(X) = \inf_{\boldsymbol{\sigma} \in \mathfrak{S}_N^L} q_{\boldsymbol{\sigma}}(X) + C_0 \tag{266}$$

As a consequence of these considerations, inside every cell $\mathcal{C}_{\boldsymbol{\sigma}}$, $F_L$ will be $C^{\infty}$ as it is equal to a quadratic function, and its gradient and Hessian at $X \in \mathcal{C}_{\boldsymbol{\sigma}}$ are respectively

$$\nabla_{X_i} F_L(X) = \frac{1}{NL} \sum_{l=1}^{L} (\langle X_i | \theta_l \rangle - b_{\theta_l, \boldsymbol{\sigma}_l^{-1}(i)}) \theta_l \tag{267}$$

and

$$\nabla_{X_i} \nabla_{X_j} F_L(X) = \frac{1}{NL} \delta_{ij} \sum_{l=1}^{L} \theta_l \theta_l^T \geq 0 \tag{268}$$

Thus, $F_L$ is convex inside every cell $\mathcal{C}_{\boldsymbol{\sigma}}$. In fact, we know by (Tanguy et al., 2024b, Theorem 2) that when $L > d$, for almost every family $\theta_1, ..., \theta_L \in \mathbb{S}^{d-1}, \bigcap_{l=1}^{L} (\mathbb{R}\theta_i)^{\perp} = \{0\}$ and $\frac{1}{L} \sum_{l=1}^{L} \theta_l \theta_l^T$ is definite positive, which makes $F_L$ strictly convex inside every cell. In these conditions, any critical point contained in a cell will be a local minimum.

This is of significance when optimizing $F_L$. Indeed, even if it were possible to derive theoretical guarantees that high energy critical points of $F$ are unstable, a numerical scheme optimizing $F_L$ could end up converging to a high energy critical point of $F_L$ because of its local convexity. Consequently, on must be chose a number of directions $L$ and of points $N$ large enough to make sure the size of the cells $\mathcal{C}_{\boldsymbol{\sigma}}$ is small enough to prevent this behavior.

**Experiments**   In another experiment, based on the discussion of Section C, we considered again the point cloud $X = (X_1, ..., X_N)$ with $X_i = -\frac{4}{\pi} + \frac{8}{\pi} \frac{i-1}{N-1}$, with $N = 100$, the perturbation $\xi$ that alternates between $e_2$ and $-e_2$, and we plotted the estimator $t \mapsto F_L(X + t\xi)$ in Figure C, where $\rho$ is the sliced-uniform measure, for different sets of test directions $\{\theta_1, ..., \theta_L\}$. We tested different values of $L$, and, for each of these values, we considered two cases :

- one set of test directions $\{\theta_1, ..., \theta_L\}$ including $e_2$, with $\theta_i = \frac{\pi}{2} + \frac{2\pi(i-1)}{L}$ for $i \in \{1, ..., L\}$

- one set of test directions $\{\theta_1, ..., \theta_L\}$ excluding $e_2$, with $\theta_i = \frac{\pi}{2} + \frac{\pi}{L} + \frac{2\pi(i-1)}{L}$ for $i \in \{1, ..., L\}$

We observe that, as expected from the discussion in Section C, when the test directions exclude $e_2$ (so that the points of $X$ have distinct projections for every test direction), the estimator $t \mapsto F_L(X + t\xi)$ is locally smooth, and we distinctively see its cell structure for the smaller values of $L$. On the other hand, when the test directions include $e_2$, we see that the estimator is not smooth at $t = 0$. This again conforms to what we theoretically expect, as

$$W_2^2(\mu_{X+t\xi,e_2}, \rho_{e_2}) = W_2^2(\frac{1}{2}(\delta_{-|t|} + \delta_{|t|}, \frac{1}{2}\mathcal{L}_{|[-1,1]}^1) = \int_0^1 (|t| - x)^2 dx = \frac{1}{3} - |t| + t^2 \tag{269}$$

so $F_L(X + t\xi) = f(t) - \frac{1}{L}|t|$ where $f(t)$ is some smooth term.

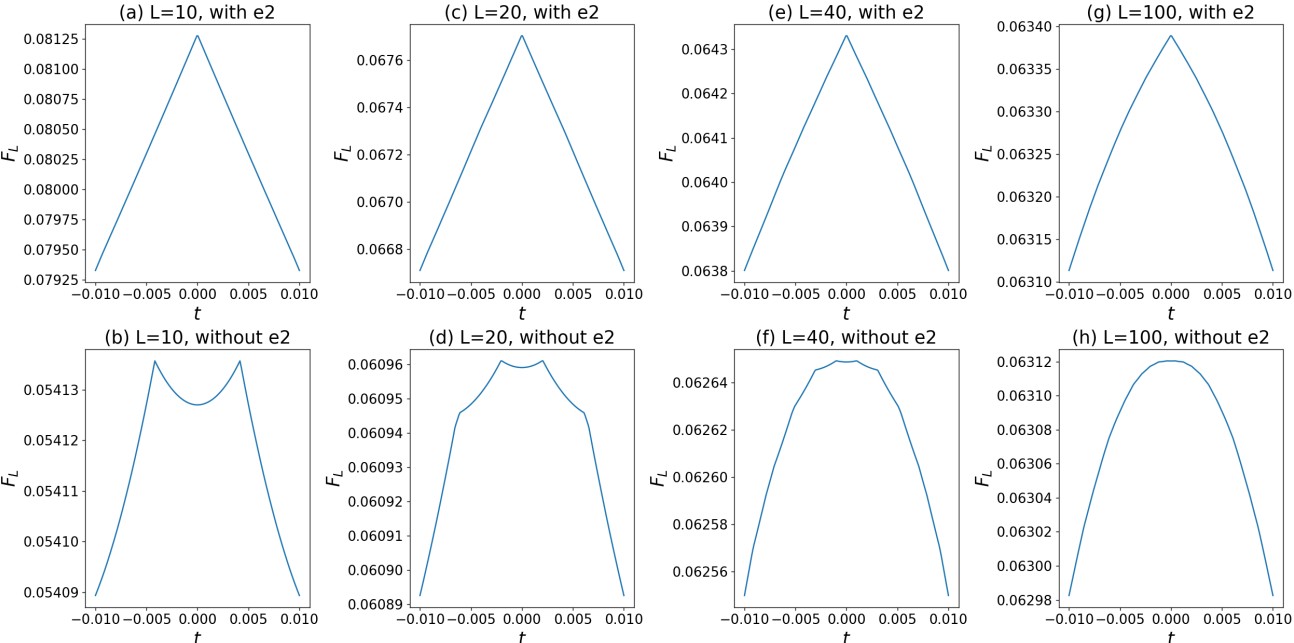

*Figure 3.* Behavior of $F_L$ for different sets of test directions. Depicts the value of $F_L(X + t\xi)$, where $X$ is a point cloud of $N = 100$ points uniformly distributed on the segment $[-4/\pi, 4/\pi] \times \{0\}$, $\xi$ alternates between $e_2$ and $-e_2$, and $\rho$ is the sliced-uniform distribution. Each column corresponds to a different number $L \in \{10, 20, 40, 100\}$ of fixed test directions ; on the top line $e_2$ is included in the test directions while on the bottom line it is excluded

