# OpenReview forum: "Towards Understanding Gradient Dynamics of the Sliced-Wasserstein Distance via Critical Point Analysis"
_ICML.cc/2025/Conference — ICML 2025 poster_

### Official Review · Reviewer_9JEB · 2025-02-28

**Overall Recommendation:** 4

**Summary:**

The paper studies the existence and stability of critical points for semi-discrete sliced Wasserstein loss functions. In particular, they prove that there exist critical points which do not coincide with the global minimum, but also that any of those critical points is unstable under small perturbations. The authors include some numerical experiments for validating the theoretical results.

## After rebuttal

The authors addressed my comments and questions adequately. I keep my rating at "accept".

**Claims And Evidence:**

The results are clear and underpinned by formal proofs.

In particular, the authors provide in Section 5 explicit examples of (Lagrangian) critical points located on lower-dimensional subspaces.
As an elementary tool, they characterize critical points of the semi-discrete sliced Wasserstein distance by the barycentric projection and prove that limits of discrete measures which are Lagrangian critical points are again Lagrangian critical point.

While I believe that the claims are interesting and the proofs are correct (even though I hadn't the opportunity to check them in detail), some limitations of the analysis could be emphasized more clearly in the introduction (please correct me if I got one of those limitations wrong):
- the examples and instability results from Section 5 are only in 2D
- the instability result only considers critical points of this specific form

**Essential References Not Discussed:**

Generally, the literature part is comprehensive and well-organized. Some additional papers should/could be discussed:

Li and Moosmueller consider in the paper "Measure transfer via stochastic slicing and matching" a stochastic gradient descent on the sliced Wasserstein distance in the continuous case (where both measures have densities). It is stochastic in the sense that in each iteration one random direction is chosen. The authors show global convergence of this scheme, which also implies that there don't exist stable critical points.

There is a paper, that proposes a numerical scheme (Altekrueger et al. "Neural Wasserstein Gradient Flows for Discrepancies with Riesz Kernels" ICML 2023) to escape critical points introduced by searching for Lagrangian instead of Wasserstein critical points (i.e. in the regular tangent space instead of the geometric one see the field "strenghts and weaknesses" below).

**Experimental Designs Or Analyses:**

Not applicable (the major claims are purely theoretical).

**Methods And Evaluation Criteria:**

The proof techniques are feasible.

**Other Comments Or Suggestions:**

see other fields

**Other Strengths And Weaknesses:**

The existence and characterization of critical points of the sliced Wasserstein distance is a problem which is of very high interest for the optimal transport community. Given that the paper makes significant progress in this direction, I definitely vote for acceptance. However, I have a couple of comments (not ordered by importance):

- From my perspective, and also in the view of machine learning applications, the most fundamental limitation of the paper is the absolute continuity assumption for the target measure. In particular, for most machine learning applications, the target measure is given by a dataset and therefore discrete. From a computational viewpoint it is usually impossible to compute the Laguerre tesselations which are required to compute the gradient in the semi-discrete case. Given the difficult nature of the problem, I would not expect the authors to consider a more general case as they have done. But I would expect them to discuss this limitation.

- Similarly as in the previous content, I would ask the authors to clarify in the abstract, that they study mostly the semi-discrete case of the SW objective.

- The authors extensively work with the different notions of critical points (Lagrangian and Wasserstein). These notions are directly related to the notions of geometric and regular tangent spaces of the Wasserstein space, which are contained in the book of Ambrosio, Gigli, Savare and were studied in more detail in the PhD thesis of Gigli. In these notations, Wasserstein critical points coincide with critical points with respect to the geometric tangent space and Lagrangian critical points coincide with critical points in the regular tangent space. For absolutely continuous measures both tangent spaces coincide. For the final version I would highly recommend the authors to work out this relations in a clean way.

- In particular, I would guess that the claim of Prop 4.3 is somehow related to the statement, that the barycentric projection is always contained in the regular tangent space (see the PhD thesis of Gigli Thm 4.15).

Reference for the PhD thesis of Gigli:
"On the geometry of the space of probability measures endowed with the quadratic optimal transport distance"

**Questions For Authors:**

- Is it clear that the Wasserstein gradient flow starting at a discrete measure with a absolutely continuous target measure remains discrete? If not, can we relate the discrete setting to the continuous one? Maybe in the sense of mean-field limits?

- Maybe some "outlook-question": Is it sufficient to choose a random direction in a gradient descent scheme of the loss to escape the critical points (like in the paper of Li, Moosmueller mentioned above)?

**Relation To Broader Scientific Literature:**

The geometry and properties of sliced Wasserstein losses was studied in several papers recently. As far as I know, the results on the non-existence of stable critical points are new and I consider them as a significant contribution.

**Theoretical Claims:**

Due to the high review load, I was not able to check the proofs in detail. Based on my intuition, the claims of the statements are realistic.

---

> ### Author Rebuttal · Authors · 2025-04-01
>
> Thank you for your positive feedback and relevant questions.
>
> - Concerning the limitations of the analysis in Section 5 : it is indeed true that most of the examples we discuss are in 2D, with the exception of our Proposition 5.1(b) which gives some examples of critical points in arbitrary dimension $d > 1$. We will make the introduction clearer about the limitations of this section.
>
> - Regarding the limitations of our assumption of absolute continuity of the target measure $\rho$, we refer to the discussion in our answer to reviewer BmZL. We will make sure to discuss this more extensively in a revised version of our article. In particular, note that one of the advantages of the Sliced-Wasserstein distance is precisely that the Laguerre tesselations are easy to compute in 1D. Indeed, if we assume that $\rho$ is discretized using $M$ points on the real line, then the Laguerre cells are computed by sorting the $M$ points, and the $i$-th Laguerre cell consists of the $i$-th chunk of $\frac{M}{N}$ points in the sorted list.
>
> - Thank you for bringing these additional references to our attention. In particular, the paper by Li and Moosmueller appears to be extremely relevant to our analysis as their assumption (A3) essentially means that the target $\mu$ is the only barycentric Lagrangian critical point in the compact set $K_{\sigma_0,\mu}$ in which the iterates of their scheme remain. This indeed ties our theoretical analysis of Lagrangian critical points to considerations of convergence of practical schemes optimizing $SW$ objectives. On the other hand, as we understand it, the paper by Altekrüger at al considers perturbations by velocity plans, which is a larger class of perturbations than in our Lagrangian framework (as a velocity plan allows to split atoms, while a perturbation by a vector field $\xi \in L^2(\mu,\mathbb{R}^d)$ cannot)
>
> - We agree that it would be fruitful to establish clear links between the different notions of critical points and the different types of tangent spaces investigated by Gigli et al. We will include a discussion on this in a revised version of the article, maybe in an appendix, since for the sake of clarity we have avoided to make use of the theory of gradient flows developed by Ambrosio, Gigli, Savaré in the main body of the article.
>
> - Whether the Wasserstein gradient flow starting at a discrete measure with an absolutely continuous target measure remains discrete is an excellent question. For $W_2$ objectives, it is known that the gradient flow diffuses (this is necessary as the flow converges to $\rho$ exponentially in the $W_2$ metric, see for instance Proposition 3.1 in [1]. For the $SW$ objective, although we have not formally shown that the flow of the $SW$ "diffuses" discrete measures, we expect it to do so. This is in fact why we chose to work with the "Lagrangian" framework and Lagrangian critical points. Indeed, what we study is the behavior of the system $dX_t^i/dt = - \nabla F(X_t^i)$ where $F$ is defined in Section 3, which corresponds to the continuous time limit of the gradient descent algorithm implemented in practice. Therefore, considering perturbations by vector fields $\xi$, which can't "split" atoms, allows for a theory better suited to analyze how algorithms relying on particle dynamics work.
>
> [1] Huang, Y. J., & Malik, Z. (2024). Generative Modeling by Minimizing the Wasserstein-2 Loss. arXiv preprint arXiv:2406.13619.
>
> - We have not investigated specifically what happens when the gradient descent is performed using a very small number of directions, such as $L = 1$. However, in our numerical experiments, we did observe that it is extremely easy to escape critical points such as those described in Proposition 5.1 or Proposition 5.2, as long as some stochasticity is introduced in the choice of directions.

---

> > ### Comment · Reviewer_9JEB · 2025-04-01
> >
> > Many thanks for your replies. As written in the original review, I vote for acceptance. One remark:
> >
> > > In particular, note that one of the advantages of the Sliced-Wasserstein distance is precisely that the Laguerre tesselations are easy to compute in 1D. Indeed, if we assume that $\rho$ is discretized using $M$ points on the real line, then the Laguerre cells are computed by sorting the $M$ points, and the $i$-th Laguerre cell consists of the $i$-th chunk of $\frac{M}{N}$ points in the sorted list.
> >
> > I would not consider your argument "we can compute the Laguerre tesselations b discretizing $\rho$" as valid here. There are some claims of the paper (such as Prop. 3.3) which are false without the assumption that the target measure has a bounded density (at least from my intuition, correct me if I am wrong). In particular, if I understand it correctly, the proposition says that the gradient descent "remains valid" (in the sense that it does not hit the diagonal) over the iterates.  From my viewpoint it remains a bitter taste if such a result is shown for the absolute continuous case, but later on used in a discretized setting. And without discretizing $\rho$, the computation involves the integration of the density over the orthogonal complement, which is most likely intractable. However, for this paper I would consider it as ok to keep it as a loose end...

---

> > > ### Author Response · Authors · 2025-04-08
> > >
> > > Thank you for your reply. Regarding Proposition 3.3, the main advantage of assuming that the $\rho_\theta$ are densities bounded by some constant $\beta > 0$, is that we can control how close the critical points and the iterates of the gradient descent can be to the diagonal by a constant of the form $\frac{C(d)}{\beta N}$ with explicit dependence on $\beta, N$. Note, however, that the fact that the iterates do not hit the diagonal as long as the step size is small enough (the second bullet point of the Proposition) does not actually require this boundedness assumption to hold. In particular, it holds for a discretized $\rho$ (the gradient descent for such a $\rho$ being well-defined thanks to the extension of Proposition 3.1 discussed in our reply to reviewer BmZL).
> > >
> > > Since we carried out our analysis in the semi-discrete setting, the additional assumption on $\rho$ didn't seem to be too costly an assumption so we wrote the entire Proposition under it, although we see now that in the light of the possible extensions of our other results to larger classes of $\rho$, our Proposition 3.3 now appears somewhat limited. We will thus also clarify these points in a revised version of our article.

---

### Official Review · Reviewer_cZYb · 2025-03-03

**Overall Recommendation:** 4

**Summary:**

This paper proposes a systematic study of the Sliced-Wasserstein functional in an optimization context where the target measure is continuous and the measure to optimize is discrete. The authors use the notion of ‘Lagrangian critical point’ of a functional defined over probability measures, since it aligns well with a particle based discretization of the objective functional. The authors then show a number of properties of this functional related to its well-behavedness when a gradient descent is used to optimize it. They also provide a characterization of the critical points of the objective (namely good behavior when the number of points in the optimized measure grows), and show that unstable critical points may exist both theoretically and experimentally. They finally show experimentally the convergence behavior of gradient descent on this functional.

## Update after rebuttal

Thanks to the authors for the answers to some of my questions. I would have appreciated discussion about the other few points raised in the rest of my review. Nevertheless, my opinion on the paper is largely positive and I maintain my score to 4.

**Claims And Evidence:**

The paper is mostly theoretical. In terms of technical contributions, the authors characterize the points of non-differentiability of the objective, compute the gradient explicitly for Lp-norm ground costs on the subset where the functional is actually differentiable. They also show that a standard gradient descent typically remain away from the problematic points. They proceed to characterize critical points and show practical examples of unstable critical points in relatively simple settings. The experiments support the theoretical analysis of the critical points and are consistent with the expected behavior of gradient descent from the theoretical analysis.

**Essential References Not Discussed:**

I am not aware of such missing references for this paper.

**Experimental Designs Or Analyses:**

I did check the soudness of the experiments, and have no particular complaints about them. It would have been interesting, experimentally at least, to give an idea about the behavior of other algorithms that are more likely to be used in practice, e.g. SGD or ADAM.

**Methods And Evaluation Criteria:**

This point does not really apply since the paper is mostly theoretical and the experiments are mostly confirming qualitatively some of the theoretical developments.

**Other Comments Or Suggestions:**

It is a bit awkward to have the introductory material about OT and SW distances presented after an introduction that is clearly not understandable if one is not already familiar with that material. I suggest moving the last part of the introdution which already gets quite technical and discusses results that are not broadly known about critical points and the behavior of W2 in a specific section after the introduction. Maybe details could be provided in an additional appendix (citing relevant theorems in the literature for instance) to make the introductory material more self contained.

**Other Strengths And Weaknesses:**

Though one could find that the setting of the paper could be restrictive, the approach is original and the results interesting. Namely it shows the theoretical difficulties of the optimization of SW distance, and offers a few guarantees the GD is likely to be relatively well behaved and to avoid non differentiabilities, while converging in practice to a stable critical point. The theoretical results generally provide new insight on important questions about the optimization dynamics of SW.
However, I found the rest of the introcution to be particularly well written and provides a good introduction to optimization over measures in ML.

**Questions For Authors:**

1) Which parts of the theoretical analyses also hold in more general settings in terms of discretization choices for the measure to be optimized, or with other types of target measures (atomless, discrete, mixed…) ? This would make the contributions more impactful and more broadly applicable in practical scenarios.
2) Is there any hope in a second order analysis of the critical points, or somehow accessing the quality of the (stable) critical points towards which GD would converge ? An experiment showing possible bad behavior in practice would be interesting.

**Relation To Broader Scientific Literature:**

The paper raises interesting points about the optimization of optimal transport distances to a reference measure. A choice that is deliberate is that the authors choose to consider cases where the target measure is absolutely continuous but the measure to optimise is discrete, and discrete in the sense that it corresponds to a Lagrangian view on the measure, i.e. a collection of particles. This brings the paper close to the setting of semi-discrete optimal transport. This choice is different from what has already been treated in the literature, which adressed the cases where both the target and the variable admit densities, or are both discrete. The case chosen by the authors is indeed of practical interest, but corresponds to one possibility among several to optimize the SW distance. The choice of an absolutely continuous target is not suited to generative modeling (where the target is only known by discrete samples) but is suited for variational inference, which is of broad interest. The choice of a Lagrangian discretization of the measure is also of interest, but other choices such as parameterizing the density of the objective via a generative model could also have been discussed (which is something that is actually done in practice). More generally, this positioning within the existing literature could have been made more explicit. However, the findings in the considered case are interesting and insightful.

**Theoretical Claims:**

I did not check thoroughly the proofs or the supplementary material. I find that a few claims in the introduction about the analysis of W_2^21 as the objective are a bit too depth in the cited works and would benefit a bit more exposure and recontextualization (I couldn’t find some of them in the related references), maybe as an additional appendix to avoid that the reader has to check many references to find the relevant theorems , or rederive things on their own.

---

> ### Author Rebuttal · Authors · 2025-04-01
>
> Thank you for your relevant remarks and questions.
>
> - Regarding the possible generalizations of our results to other types of optimized or target measures (notably discrete ones), we refer to our answer to reviewer BmZL which addresses these points extensively. Note in particular that even though we assumed throughout the article that the target measure $\rho$ is a density, this is mostly because we started our investigation from the study of the semi-discrete setting, and most of our results can actually be proven when $\rho$ is simply assumed to be without atoms. This covers many types of measures often encountered in machine learning, such as measures supported on lower dimensional manifolds of $\mathbb{R}^d$.
>
> - We did make some attempts at a second order analysis of the $SW$ distance, but it turned out to be a significantly difficult problem. Most of what we know concerns cases such as the ones in Proposition 5.2, where we have explicit expressions of the quantile functions and we can compute explicit Taylor expansions of the $SW$. Even the discrete case is difficult : while formally differentiating under the integral sign does indeed give the expression of $\nabla F$ (Proposition 3.1), we cannot obtain the Hessian of $F$ by this method, as formally differentiating again $\nabla F$ under the integral sign gives $\frac{1}{Nd} I_d$, which cannot be the actual Hessian of $F$ as $F$ is not convex (it is semiconcave by Proposition A.2). Furthermore, attempts to numerically approximate the Hessian were inconclusive, as the computed Hessian would converge to $\frac{1}{Nd} I_d$ when we increased precision or added more directions (which is to be expected from the analysis of Appendix B, which shows that the approximation of $SW$ using a fixed number $L$ of directions $\theta_1,\ldots,\theta_L$ has Hessian $\frac{1}{NL} \sum_l \theta_l \theta_l^T$ everywhere where it is defined, and this converges to $\frac{1}{Nd} I_d$ when $L \to \infty$ and the $\theta_l$ are choosed randomly).

---

### Official Review · Reviewer_F449 · 2025-03-12

**Overall Recommendation:** 3

**Summary:**

The paper under consideration presents theoretical results regarding (Section 3) properties of discrete gradient descent w.r.t. SW functional (with absolute continuous target measure) and (Section 4) properties of (lagrangian) critical points of SW functional; (Section 5) provides some examples of lower-then-data dimension critical points different from the target functional and states that they (seems to be) unstable. The paper has an experimental section which illustrates some of the theoretical findings.

### Update after the rebuttal

I thank the authors for their response. I think, my current score is solid

**Claims And Evidence:**

Ok.
But I have a question: (lines 369-371): “We can find $\xi$ such that …”. I do not understand how to find such $\xi$? May you give some examples of such $\xi$?

**Essential References Not Discussed:**

Everything seems to be ok.

**Experimental Designs Or Analyses:**

N/A: The paper is theoretical

**Methods And Evaluation Criteria:**

N/A: The paper is theoretical

**Other Comments Or Suggestions:**

No

**Other Strengths And Weaknesses:**

On the one hand (without delving into proofs), I find the manuscript more-or-less understandable and easy-to-read. It is definitely a strength for such a theoretical work with a lot of statements.

On the other hand, the main weakness of this paper is that it is almost fully theoretical. There is an experimental section, but it just presents some illustrations of some theoretical claims.

So, my opinion is as follows: as the theoretical work, the paper is good, but it has rather limited practicality. Maybe Proposition 3.2. is interesting to some extent.

**Questions For Authors:**

1. Line 154, second line: “descent lemma” for this objective. What is “descent lemma”?

2. Minor, lines 367 an 371 - conflict of notations, $a$ used for different properties.

3. What does it mean: “alternating vector field $\xi$”? - lines 369-370 (second column)

**Relation To Broader Scientific Literature:**

Some related but not cited papers devoted to modelling Wasserstein gradient flows w.r.t KL-divergence:

[1] Large-scale Wasserstein Gradient Flows, NeurIPS’21

[2] Optimizing Functionals on the Space of Probabilities with Input Convex Neural Networks, TMLR

[3] Proximal Optimal Transport Modeling of Population Dynamics, AISTATS’22

Also, not that directly related, but also a good reference. This work study gradient flows in Sliced-Wasserstein space:

[1] Efficient Gradient Flows in Sliced-Wasserstein Space, TMLR

**Theoretical Claims:**

I checked only the general flow of the statements of the theorems/propositions in the main text. I didn’t check the proofs.

---

> ### Author Rebuttal · Authors · 2025-04-01
>
> Thank you for your positive remarks and comments. Regarding the questions you raised :
>
> - We meant by "descent lemma" a result that guarantees that, provided some conditions on the initial point and the step size are satisfied, one step of the gradient descent will decrease the loss. In particular, it gives the maximal step-size for which descent is maximal and still guaranteed - this size is related to the inverse of the smoothness constant of the functional. Above this limit, descent is not guaranteed for an explicit time-discretization, as corroborated by our experiments (Figure 2) where step-size below $2Nd$  yields descent while step-size above $2Nd$ diverges. Hence, while this is a theoretical result, it gives practical hints for optimization of the $SW$ distance in practice since below $2Nd$ (which is a known quantity) the descent is guaranteed. The term "descent lemma" is a standard term in optimization, cf for instance [1]. We will revise the article to clarify this point.
>
> [1] Bauschke, H. H., Bolte, J., & Teboulle, M. (2017). A descent lemma beyond Lipschitz gradient continuity: first-order methods revisited and applications. Mathematics of Operations Research.
>
> - Regarding the meaning of "a suitable alternating vector field $\xi$" at lines 369-371 (second column), we meant was that (using the notations of Proposition 5.2) by approximating the perturbation $\mu^t$ by $(Id+t\xi)\mu$ where $\xi$ rapidly alternates between $\vec{n}$ and $-\vec{n}$ on the segment $S$, we may hope that $SW^2_2((Id+t\xi)\mu,\rho)$ will also have a maximum at $t = 0$. For example, for $S = [-1,1] \times \{0\}$, we may consider $\xi$ such that $\xi(x,0) = \vec{e_2}$ for $x \in [i/n,(i+1)/n)$ with $i$ even, and $\xi(x,0) = -\vec{e_2}$ for $x \in [i/n, (i+1)/n)$ with $i$ odd, with $n$ large. In the experiments shown in Figure 1, we use such alternating perturbations. We will make our formulation clearer in a revised version of the article.
>
> We thank you for the additional references, which we will accordingly cite in a revised version of our article.

---

### Official Review · Reviewer_BmZL · 2025-03-16

**Overall Recommendation:** 4

**Summary:**

This paper investigates the properties of gradient flows for the Sliced-Wasserstein (SW) distance when used as an objective functional. It rigorously develops different notions of critical points—Eulerian, Wasserstein, and Lagrangian (including a barycentric variant)—and studies the convergence and stability properties of discrete gradient descent schemes that approximate the continuous Wasserstein gradient flow. The theoretical contributions include proving that, under suitable assumptions, the discrete (particle-based) critical points converge to a continuous critical point and that “bad” critical points (e.g., those supported on lower-dimensional structures) are unstable. Numerical experiments are provided to validate the theoretical findings and illustrate the behavior of gradient descent dynamics with various step sizes.

**Claims And Evidence:**

The paper’s claims are well supported by a combination of rigorous theoretical analysis and simple synthetic numerical experiments. The main claims -- such as the equivalence of discrete and continuous notions of criticality (Proposition 4.3 and Theorem 4.4) and the instability of lower-dimensional critical points (Proposition 5.2) -- are backed by detailed proofs in the supplementary material and supported by illustrative experiments. However, some claims rely on technical assumptions (e.g., compact support, absence of atoms) that may limit generality, and additional discussion on these assumptions would be beneficial.

**Essential References Not Discussed:**

n/a

**Experimental Designs Or Analyses:**

The experiments are designed to illustrate both the convergence behavior of gradient descent (with respect to various step sizes) and the instability of undesired critical points. The numerical analyses (e.g., the plots in Figures 1 and 2) support the theoretical insights. One concern is the limited range of experiments only on synthetic data, but given the theoretical nature of this paper, this is not a major issue.

**Methods And Evaluation Criteria:**

The methodology is sound: the paper formulates the SW objective within the Wasserstein space $P_2(\mathbb R^d)$  and develops discrete gradient descent dynamics on an empirical measure. This approach is appropriate for bridging continuous optimal transport theory with practical particle methods. Empirical evaluation is conducted through of numerical experiments that give the readers a better sense of validation of the theoretical analysis.

**Other Comments Or Suggestions:**

n/a

**Other Strengths And Weaknesses:**

The paper provides a deep and rigorous theoretical analysis of the SW gradient flow, filling a gap in the literature regarding the convergence properties and stability of critical points.

**Questions For Authors:**

- How do the results extend (or fail to extend) if the target measure $\rho$ or the approximating measures $\mu$ are not atomless? Could you comment on potential generalizations or necessary modifications of your framework in such cases?

**Relation To Broader Scientific Literature:**

The paper is well situated within the literature on optimal transport, Wasserstein gradient flows, and generative modeling using SW distances. It builds on foundational works in the field (e.g., by Ambrosio et al., Villani, and Bonnotte) and connects with recent studies on particle methods and generative models (e.g., Merigot et al., 2021; Liutkus et al., 2019).

**Theoretical Claims:**

I carefully checked several key proofs (notably those for Proposition 3.1, Proposition 4.3, and Theorem 4.4). The derivations appear to be mathematically rigorous, and the use of techniques from optimal transport theory (e.g., Wasserstein geometry, barycentric projections) is correct.

---

> ### Author Rebuttal · Authors · 2025-04-01
>
> Thank you for your positive remarks and relevant questions. Even though we restricted our theoretical analysis of the $SW$ distance to absolutely continuous target measures (and in Section 3 to discrete approximating candidates), this was mostly for the sake of simplicity, and it is indeed possible to extend many of our results to larger classes of measures, often (but not always) without requiring significant adaptation of the proofs. For example :
>
> - Direct extensions :
>
>     - In section 4, the notion of barycentric Lagrangian critical point for $SW^2_2(\cdot,\rho)$ can be defined for arbitrary measures $\mu, \rho \in P_2(R^d)$, as the 1D optimal transport plans $\gamma_\theta$ are always uniquely defined.
>
>     - If we replace the assumption that $\rho$ is absolutely continuous by the assumption that it has no atoms, then Propositions 3.1, 3.2, 4.3, 4.6, Theorem 4.4 and Corollary 4.7 remain true.
>
> - Minor extensions :
>
>     - When $\rho$ is an uniform point cloud $\rho = \frac 1N \sum_{i=1}^N \delta_{Y_i}$ (with the same $N$ as in $\mu$), it should be possible to prove analogues of Propositions 3.1 and 3.2, replacing the barycenters $b_{\theta,i}$ by the reordered projections of the points $Y_i$ in the statements of the propositions (in fact, Theorem 1 in Bonneel et al. 2015 is the analogue of our Proposition 3.1 with $p=2$).
>
>     - Proposition 3.3 remains true under the weaker assumption (A1) on $\rho$ that there exists $\beta > 0$ and $\Theta \subset \mathbb{S}^{d-1}$ with $\mathcal{H}^{d-1}(\Theta) > 0$ such that $\rho_\theta$ is a density bounded from above by $\beta$ for every $\theta \in \Theta$
>
>     - The proof of Proposition 5.2 should be adaptable to the case where we only assume of $\rho$ that it has no atoms, and $\exists b > 0$ and a neighborhood $U \subset \mathbb{S}^1$ of $\vec{n}$ such that $\forall \theta \in U$, $\rho_\theta$ is a density bounded from above by $b$ - let's call this assumption (A2).
>
> - Significant extensions :
>
>      - The proofs of Propositions 3.1 and 3.2 should be adaptable to arbitrary $\rho$. The main difficulty would be that the Power cells $V_{\theta,i}$ are not well-defined. We would instead have to work with a decomposition $\rho_\theta = \sum_i \rho_{\theta,i}$ where $\rho_{\theta,i}$ is coupled to $\langle X_{\sigma_\theta(i)}, \theta \rangle$ by the optimal 1D transport plan (when $\rho_\theta$ has no atoms, we have $\rho_{\theta,i} = \rho_{\theta|V_{\theta,i}}$)
>
>     - Theorem 4.4 also holds under the assumption that $\rho$ has no atoms and that the set of atoms of $\mu$ is closed, but proving this requires more sophisticated methods. The sketch of the proof is roughly the following : the beginning of the proof is the same as in the article. We cannot use Proposition 4.6(c) as $\mu$ may have atoms, but we can prove that Equation (152) still holds is $\xi$ is assumed to cancel at the atoms of $\mu$, and this allows to prove $v_\mu = 0$ $\mu$-ae on the complementary of the set of atoms of $\mu$. Then, by considering perturbations of each individual atom of $\mu$, we show that $v_\mu$ also cancels at the atoms. Since this assumption on $\mu$ seems unnatural and its proof is more complex, we chose to state the theorem in the article under the stronger but more natural assumption that $\mu$ is without atoms.
>
> Moreover, regarding the limitations of our analysis :
>
> - While our original assumption of absolute continuity of $\rho$ does limit the applicability of our results, the extensions we discussed to weaker assumptions on $\rho$ such as atomlessness, (A1) or (A2), allow us to cover a much wider range of target measures, including many types of singular measures which arise in machine learning, such as densities supported on a lower dimensional manifold (for example, $\rho = \frac 12 \mathcal{H}^1_{[-1,1]\times\{0\}} \in \mathcal{P}_2(\mathbb{R}^2)$ has no atoms, and satisfies (A1) and (A2) for $\vec{n} = \vec{e}_2$)
>
> - The assumption of compact support of $\mu, \rho$, which we some of our results require, seem harder to relax. Indeed, it is needed in Proposition 4.6(c) to obtain regularity properties of the Kantorovich potentials, which we use to prove the differentiability, while the proof of Theorem 4.4 makes use of the fact that in a compact space, $SW_2$ and $W_2$ are equivalent distances metrizing the topology of weak convergence.
>
> - Finally, the fact that our numerical experiments, in which the target measures were discretized, exhibit the behaviors of convergence and instability that our theoretical analysis highlighted, suggests that our results should still be relevant in the cases where the target measure $\rho$ is approximated by a discrete measure.
>
> We will add in a revised version of our article a discussion of the generalizability of our results and of their limitations.

---

### Decision · Program_Chairs · 2025-05-01

**Decision:**

Accept (poster)

**Comment:**

The authors study the properties of gradient flows with Sliced-Wasserstein (SW) distance as an objective functional. In particular, the authors consider the semi-discrete SW loss functions, and study the existence and stability of critical points via the notion of Lagrangian critical point. The authors derive important theoretical results including the existence (in lower-dimensional subspaces) and instability of critical points (which are not global minimum) but in relatively simple settings; properties of SW functional for well-behavedness of gradient descent. The authors also characterize critical points by the barycentric projection and prove the limits of critical points are critical. The authors provide simple empirical simulations to illustrate their theoretical findings. Additionally, during the rebuttal, the authors also discuss extensions for the semi-discrete SW. We urge the authors to incorporate these extensions and also discuss the limitations (e.g., compact support assumption, relatively simple settings for theoretical instability analysis) in the update version. Overall, we think that it is a good submission, and provides interesting theoretical results for gradient flows with SW as an objective functional.